# The environmentally-regulated interplay between local three-dimensional chromatin organisation and transcription of *proVWX* in *E. coli*

Fatema-Zahra M. Rashid [1,2,3], Frédéric G. E. Crémazy [1,4], Andreas Hofmann [5], David Forrest [6], David C. Grainger [6], Dieter W. Heermann [5] & Remus T. Dame [1,2,3] ✉

Nucleoid associated proteins (NAPs) maintain the architecture of bacterial chromosomes and regulate gene expression. Thus, their role as transcription factors may involve three-dimensional chromosome re-organisation. While this model is supported by in vitro studies, direct in vivo evidence is lacking. Here, we use RT-qPCR and 3C-qPCR to study the transcriptional and architectural profiles of the H-NS (histone-like nucleoid structuring protein)-regulated, osmoresponsive *proVWX* operon of *Escherichia coli* at different osmolarities and provide in vivo evidence for transcription regulation by NAP-mediated chromosome re-modelling in bacteria. By consolidating our in vivo investigations with earlier in vitro and in silico studies that provide mechanistic details of how H-NS re-models DNA in response to osmolarity, we report that activation of *proVWX* in response to a hyperosmotic shock involves the destabilization of H-NS-mediated bridges anchored between the *proVWX* downstream and upstream regulatory elements (DRE and URE), and between the DRE and *ygaY* that lies immediately downstream of *proVWX*. The re-establishment of these bridges upon adaptation to hyperosmolarity represses the operon. Our results also reveal additional structural features associated with changes in *proVWX* transcript levels such as the decompaction of local chromatin upstream of the operon, highlighting that further complexity underlies the regulation of this model operon. H-NS and H-NS-like proteins are wide-spread amongst bacteria, suggesting that chromosome re-modelling may be a typical feature of transcriptional control in bacteria.

Nucleoid Associated Proteins (NAPs) are architectural proteins that bind along the bacterial chromosome maintaining its compaction and organisation. They do so by DNA bending, lateral filament formation along the DNA, or DNA bridging, the latter of which results in the formation of long- and short-range DNA loops that contribute to the global and local structural organisation of the genome. NAP binding is sensitive to environmental changes such as fluctuations in temperature, pH, and osmolarity. Hence, NAPs organise the bacterial chromosome into a dynamic structure that is re-modelled in response to changes in the cell environment. NAPs also function as transcription factors that coordinate global gene regulation in response to environmental stimuli. Collectively, these characteristics entail a coupled

sensor-effector model of NAPs whereby NAPs regulate gene expression by reorganising the local structure of the chromosome at the level of individual operons in response to stimuli that modulate their architectural properties[1–11].

Histone-like Nucleoid Structuring protein (H-NS) of *Escherichia coli* is a 137 amino acid long NAP that functions as a global regulator of gene expression[12] and a xenogeneic silencer[13]. It exists as a dimer in solution[14], formed via interaction between a pair of N-terminal dimerisation domains[15–17]. The dimer binds to the minor groove of AT-rich DNA with an AT-hook-like motif within its C-terminal DNA binding domain[18]. The interaction serves as a nucleation point for the co-operative multimerization of H-NS along the DNA to form a protein-DNA filament. Structurally, the H-NS dimers are held together via interactions between central dimer-dimer interaction domains[19,20]. H-NS−DNA filaments can repress transcription by occluding the binding of RNA polymerase to promoter and promoter-like elements trapped within the structures[21–23], but the filaments do not exert a detectable roadblocking effect on elongating RNA polymerase in vitro[24]. The presence of two DNA binding domains per dimer allows the H-NS multimer to recruit a second DNA molecule to form a DNA−H-NS−DNA bridge[2,25–27]. DNA−H-NS−DNA bridges occlude the binding of RNA polymerase (RNAP) to promoters[28], block transcription initiation at the promoter clearance step by trapping RNAP in a loop[29], and impede transcription elongation in vitro[24].

H-NS is a coupled sensor-effector[11]. Its structural conformation and, by extension, its global gene regulation, is sensitive to physicochemical cues. Its response to osmolarity, in particular, is modulated by a $K^+$- and $Mg^{2+}$-sensitive α-helix, helix α3[27,30], that extends between the N-terminal dimerisation domain and central dimer-dimer interaction domain[19]. In vitro and in silico studies indicate that the stabilisation of the helix α3 by $Mg^{2+}$ favours the open conformation of H-NS dimers in which both C-terminal DNA-binding domains are available for DNA binding. This promotes the formation of DNA−H-NS−DNA bridges that interfere with transcription initiation and elongation. Destabilization of helix α3 by $K^+$ shifts the equilibrium of the conformation assumed by H-NS dimers towards the closed state where one DNA-binding domain of the dimer is folded onto the body of the protein[27]. This favours the formation of H-NS−DNA filaments[27] – structures that hinder transcription initiation but are conducive to elongation[24]. In contrast to eukaryotic systems where a causal link between chromatin architecture and transcription has been established[31], in vivo evidence highlighting an interplay between local three-dimensional chromatin organisation and gene expression in prokaryotes is lacking. The dual role of H-NS as a transcription factor and an osmoresponsive architectural protein makes it an attractive choice for addressing this knowledge gap. Hence, we examined the H-NS-mediated structural regulation of the *proVWX* operon in *Escherichia coli*.

*ProVWX* (*proU*)[32,33] is an H-NS-regulated, osmoresponsive operon activated by the rapid cytoplasmic accumulation of $K^+$ (counter-ion: glutamate) that ensues as a primary response to hyper-osmotic stress[34–41]. The acute increase in cytoplasmic $K^+$ prevents cell dehydration by drawing water back from the environment[42]. This, however, occurs at the expense of protein stability and function[36]. $K^+$ also behaves as a second messenger that activates the cell's secondary long-term responses to osmotic stress, which includes the activation of *proVWX*. The products of the expression of the three structural genes – *proV*, *proW*, and *proX* – assemble into the ProU transporter – a transmembrane protein complex of the ATP-binding cassette (ABC) superfamily[43]. ProU imports osmoprotectants such as glycine-betaine, proline-betaine, and proline, among others, into the cell with high affinity[33,37,38,44,45]. The osmoprotectants maintain a low osmotic potential in the cytoplasm without an adverse effect on cellular physiology, allowing $K^+$ to be expelled from the cell[36,46].

The osmosensitivity of the *E. coli proU* operon is inherent in the sequences of its $σ^{70}$-dependent promoter (P2), and cryptic[47,48] $σ^S$-dependent promoter (P1), positioned 60 and 250 bp upstream of the *proV* open reading frame (ORF), respectively[39,49,50] (Supplementary Data 1A). The osmosensitivity is also inherent in the *cis* regulatory elements that extend across the promoters[50]. Indeed, *proU* is activated by $K^+$ in the absence of a trans-acting factor. The addition of increasing concentrations of K-glutamate, but not of L-proline and glycine betaine, to a purified in vitro transcription system stimulates the expression of *proU* cloned on a plasmid DNA template from both, the P1 and P2 promoters[39,51,52]. Non-osmoregulated genes such as *bla*, *lac*, and *pepN* are repressed under identical conditions[52]. The inherent osmosensitivity of *proU* may arise from the non-consensus −10 sequence of P2. P2 carries three GC base pairs that interfere with dsDNA melting and open complex formation. Increasing concentrations of K-glutamate may cause microstructure changes to the promoter that either increase its accessibility to RNAP or favour the isomerisation of the RNAP-bound promoter to form the open promoter complex[52].

Owing to Rho-dependent termination of transcription from P1 under ordinary growth conditions, *proVWX* expression is primarily driven by P2[47,48]. Expression therefrom is regulated by the negative regulatory element (NRE) that extends from ~300 bp upstream of the *proV* ORF to -1100 bp into the *proV* ORF[50] (Supplementary Data 1A). *ProV* is 1203 bp in length, hence, truncates of the gene double as truncates of the NRE. β-galactosidase assays performed by generating in-frame *lacZ* fusions to the 5′ end of *proV* truncates show that the osmoresponse of *proVWX* weakens as the NRE is shortened from its downstream end[50]. The NRE exerts its role by means of H-NS, and, as of yet, has been shown to confer osmosensitivity only to *proU* P2. Other promoters cloned in place of *proU* P2 do not exhibit an osmolarity-dependent response[49,50,53,54]. A pair of *cis* regulatory elements occur within the broad region designated as NRE: the upstream regulatory element (URE) positioned at −229 to −47 of the P2 transcription start site (TSS), and the downstream regulatory element (DRE) that stretches across P2 from −40 to +177[39,49,50,55,56] (Supplementary Data 1A). An intrinsic curvature in the structure of the URE (Supplementary Data 1A) contributes to the osmotic inducibility of P2. The insertion of spacer sequences between P2 and the upstream curved DNA sequence shows that the two elements must be positioned stereospecifically, that is, the inserted spacer must comprise (multiples of) a full turn of DNA for the promoter to be fully activated at high osmolarity[57]. The URE exerts its role by means of H-NS − a NAP that preferentially binds curved DNA[58–61]. The URE acts cooperatively with the DRE in an *hns*+ strain to strengthen the repression of *proU* at low osmolarity and enhance its activation at higher osmolarity[49]. The DRE consists of a pair of high-affinity H-NS binding sites that function as nucleation sites for the formation of an H-NS−DNA nucleoprotein complex[56,62] (Supplementary Data 1A). In vitro studies support a hypothesis where at low osmolarity (i.e. low intracellular $K^+$), the H-NS−DNA complex organises into a transcriptionally-repressive bridged conformation to silence *proVWX*[24,27,49,55,56], reminiscent of the role of H-NS at the *rrnB* P1, *hdeAB*, and *bgl* promoters[29,63,64], and that the cellular influx of $K^+$ upon a switch to a higher osmolarity may drive a $K^+$-mediated reorganisation of the H-NS−DNA complex to relieve *proVWX* repression[24,27,39,51].

In addition to H-NS, the NAPs IHF and HU also regulate *proU*[65–68]. IHF increases the expression of *proU* but does not affect the osmo-inducibility of the operon[67]. IHF mediates its effect via a putative binding site positioned ~450 bp upstream of P2[67], and the specific binding site of the protein in the −33 to +25 region around the P2 TSS[69]. Due to limited study, it is unclear whether *proVWX* activation as a result of upstream IHF binding is affected through P1 or P2[67,68], however, the specific binding of IHF at P2 structurally modifies the promoter to partially relieve H-NS-mediated repression at low osmolarity[69]. An effect of *ihf* deletion on the expression of the *proVWX* operon was not observed in genome-wide studies[65], likely due to the modest decrease in the expression of the operon at low osmolarity conditions in the absence of IHF[67]. HU regulates expression from the *proU* P2 promoter

and its osmoinducibility[68]. Δ*hupA*, Δ*hupB*, and Δ*hupAhupB* mutants show reduced *proU-lacZ* expression compared to the corresponding wild-type strain, and a reduced fold-change in expression from P2 between low and high osmolarity conditions[68]. HU plays its role independently of H-NS as evidenced from Δ*hupB* Δ*hns* double mutants, where expression from P2 is realised as sum of the individual mutations rather than an epistatic output[68]. The *proU-lacZ* expression results presented for Δ*hupA*, Δ*hupB*, and Δ*hupAhupB* strains[68] conflict with genome-wide transcriptome data[66] which indicate that a Δ*hupA* mutation derepresses *proU* during the exponential phase of growth, Δ*hupB* derepresses the operon during transition-to-stationary phase, and Δ*hupAhupB* in stationary phase[66]. However, the detection of *proU* expression at the transcriptional level from its native locus in the chromosome in genome-wide experiments[66] and post-translationally from a plasmid-localised *proU-lacZ* gene in reporter gene expression assays[68] may account for the discrepancy. The ectopic placement of a non-native *proU-lacZ* construct eliminates the effects of relevant regulatory elements and perturbs local topological states that impact *proU* expression either directly, or indirectly by affecting the binding of regulatory NAPs. StpA, an H-NS paralogue (58% sequence identity)[70,71] that has been proposed to function as a molecular back-up for H-NS[72] represses *proV-lacZ* 9-fold when expressed from multi-copy plasmids in *E. coli* Δ*hns*[73]. Chromatin Immunoprecipitation (ChIP) of FLAG-tagged StpA shows that in wild-type *E. coli* and in *E. coli* Δ*hns*, StpA binds the *proVWX* regulatory region and its occupancy spans over P1 and P2[74]. StpA also binds the *proV* ORF, however, only as a heterooligomeric complex with H-NS[74]. A ChIP signal of FLAG-tagged StpA is not observed at *proV* in *E. coli* Δ*hns*[74]. Conversely, FLAG-tagged H-NS is detected at *proV* in *E. coli* Δ*stpA*[74].

The transcription of the *proVWX* operon is also regulated by genetic and chemical factors that modify the supercoiling density of the genome[50,51,75,76]. Mutations to DNA gyrase, and growth in media supplemented with non-lethal concentrations of novobiocin, a DNA gyrase inhibitor, reduce the levels of negative supercoiling in the cell and the expression of *proVWX*[75,77]. Mutations in *topA*, and growth in high osmolarity media increase negative supercoiling density and *proVWX* expression[75]. However, extremely high levels of negative supercoiling observed in Δ*topA* mutants repress the operon[75]. Furthermore, in a purified in vitro transcription system, transcription from the P2 promoter increases with increasing negative supercoiling density of the template[51]. Increased negative supercoiling may favour transcription by facilitating the melting of the relatively GC-rich P2 promoter and promoting a structural reorganisation of the H-NS−NRE nucleoprotein complex[46].

Collectively, the coordinated role of an inherently osmoresponsive promoter[39,49,50], supercoiling density, and architectural proteins – HU[68], IHF[67–69], StpA[73,78], and the osmoresponsive H-NS[27,39,49,55,56] – in the activation and repression of the *proU* operon from the P2 promoter implies that the regulation of *proVWX* may involve local three-dimensional reorganisation of chromatin structure, analogous to eukaryotic chromatin remodelling. Here, we use RT-qPCR and 3C-qPCR to study the interplay between transcription and local three-dimensional chromatin structure at the *proVWX* operon in low and high salt conditions, and in response to a hyperosmotic shock.

## Results and discussion
### Transcription of the *proVWX* operon in *E. coli* K-12 strain MG1655 Δ*endA* (NT331)
Early studies examining the osmosensitivity of the *proU* operon involved transcriptional and translational fusions of *lacZ* to truncates of *proV* and the subsequent detection of the specific activity of β-galactosidase (*lacZ*) in a variety of genetic backgrounds and osmolarity conditions as a measure of *proU* expression[34,37,39,47,49,50,53–55,67,68,78,79]. The technique is limited in that it relies on post-translational detection. It

requires transcription of the reporter gene construct and translation of the mRNA for a read-out. As such, it is more reflective of intracellular protein levels than transcription. RT-qPCR (reverse transcriptase quantitative PCR) directly measures transcript levels. Therefore, we used RT-qPCR to study the osmolarity-dependent response of *proVWX*. We designed eleven primer pairs to evaluate the transcriptional profile across the operon and its flanking genes (Supplementary Data 1A and 1B).

The transcriptional profile across *proU* shows that the central *proW* gene has lower transcript levels than the flanking *proV* and *proX* genes and that the transcript level of the terminal *proX* gene is higher than that of *proV*, in agreement with genome-wide RNA-seq studies[80]. This trend was observed at 0.08 M NaCl, 0.3 M NaCl, and upon a hyperosmotic shock (Fig. 1a−c and Supplementary Fig. 1a−c; Supplementary Tables 1 and 2). Mechanistically, the decline in relative expression between *proV* and *proW* may be accounted for by transcription termination at the junction of the two genes, while the sharp increase in expression between *proW* and *proX* may arise because of the presence of an internal promoter between the two genes. Indeed, with Term-seq – a technique used to identify transcription termination sites[81] – the presence of a termination site downstream of *proW*1 at chr:2806395-2806398 was detected in our study and at chr:2806397-2806401 in an earlier report[82] (Accession number: NC_000913.3 – https://www.ncbi.nlm.nih.gov/nuccore/556503834) (Supplementary Fig. 2, Supplementary Data 1A). Despite its positioning downstream of *proW*1 (Supplementary Data 1A), the terminator may have had an impact on the *proW*1 RT-qPCR signal. Our Term-seq studies also identify a termination site between *proW* and *proX* downstream of a putative hairpin structure (Supplementary Fig. 2, Supplementary Data 1A), however, this site was not detected in Term-Seq studies performed by others[82]. TSS mapping using differential RNA sequencing[83] identifies a TSS within *proW* at chr:2806878 (NC_000913.3 – https://www.ncbi.nlm.nih.gov/nuccore/556503834)[84] positioned 23 basepairs downstream of the *proW*2 amplicon. Transcription initiating from this site may account for the increase in the transcript levels of *proX*. The promoter regulating expression from this site is likely a non-canonical promoter rather than a classical H-NS-repressed spurious promoter since H-NS ChIP does not show a reproducible H-NS signal at *proW* (Supplementary Fig. 3)[85], and spurious transcripts have not been detected from *proW* in a Δ*hns* background[86].

The increased transcript level in the downstream region of *proW* (amplicon *proW*2) compared to its upstream end (amplicon *proW*1) (Fig. 1a−c and Supplementary Fig. 1a−c) implies the presence of an additional TSS within the *proW* ORF or increased stability of the 3′ region of the *proW* transcript. TSS mapping with differential RNA sequencing shows no such a feature between the *proW*1 and *proW*2 amplicons[84]. However, a genome-wide study aimed at mapping RNA G-quadruplexes (rG4) in the *E. coli* transcriptome detected the presence of rG4 structures within the *proW* coding sequence[87], occurring between the *proW*1 and *proW*2 amplicons (Supplementary Data 1A). Using the *hemL* transcript that encodes glutamate-1-semialdehyde aminotransferase (aminomutase) as a model, rG4s were shown to play a role in the stabilisation of transcripts[87]. $K^+$ stabilises rG4s when the ion occupies the centre of the quartet[88]. Extrapolation of these findings to rG4s at *proW* suggests that at higher intracellular concentrations of $K^+$, the increase in the relative transcript level of *proW*2 may be detected due to the stabilisation of rG4s at the 5′ region of the transcript.

The *proVWX* operon is repressed in NT331 cells growing exponentially in a medium with an NaCl concentration of 0.08 M. Upon a hyper-osmotic shock that raises [NaCl] from 0.08 M to 0.3 M, the transcript levels of *proVWX* increase by up to ~8-fold (Fig. 1d and Supplementary Fig. 1d). The spike in expression may be mediated by the relief of H-NS-mediated repression owing to the cytoplasmic influx

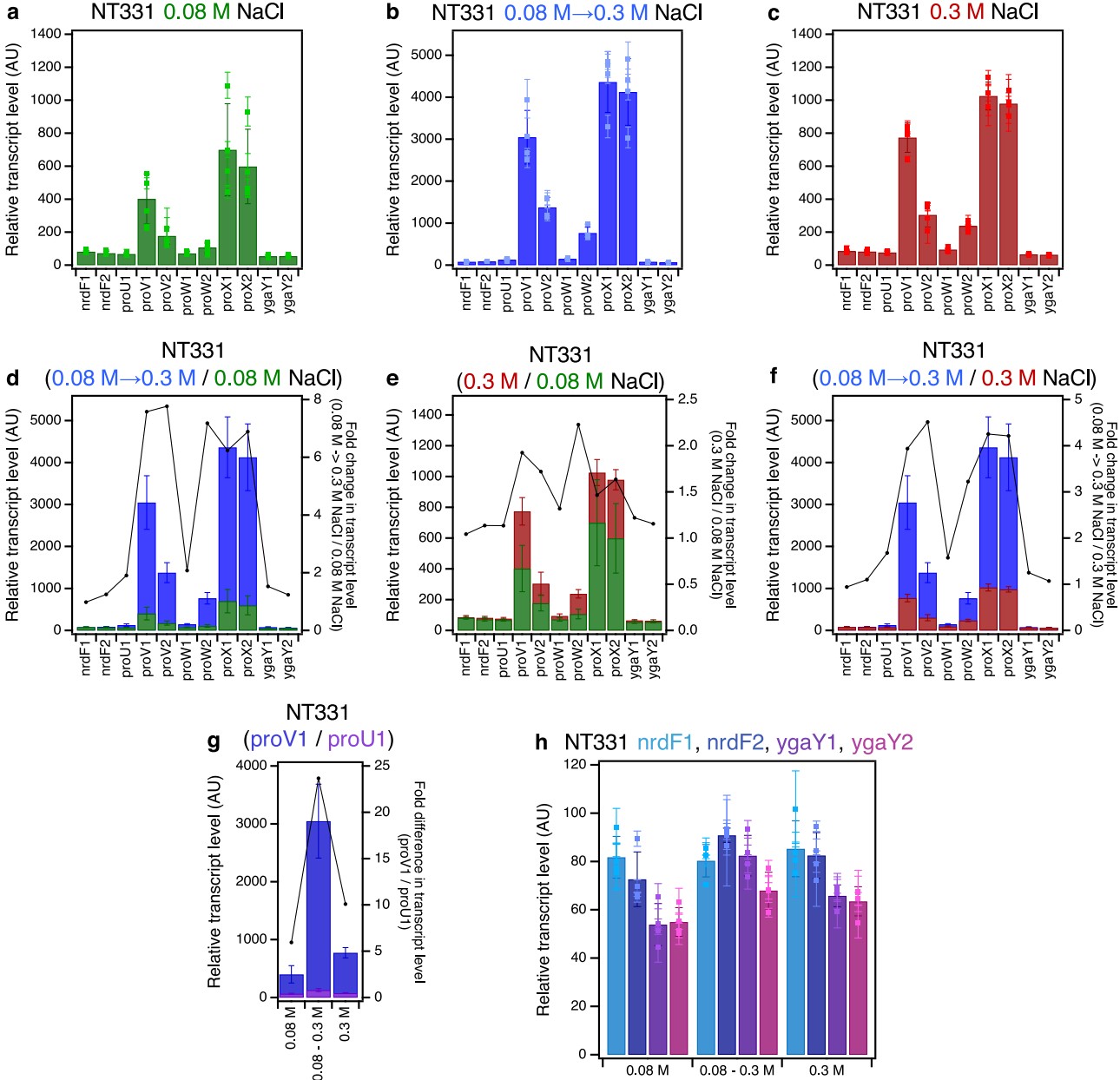

**Fig. 1 | ProU transcription in NT331 (MG1655 ΔendA).** The RT-qPCR profile of *proVWX* and its flanking regions in NT331 during **a** exponential growth in M9 medium with 0.08 M NaCl, **b** hyperosmotic shock in M9 medium from 0.08 M to 0.3 M NaCl, and **c** exponential growth in M9 medium with 0.3 M NaCl. The fold changes in transcript levels of *proVWX* and its flanking regions between **d** a hyperosmotic shock and exponential growth at 0.08 M NaCl, **e** exponential growth at 0.3 M NaCl and 0.08 M NaCl, and **f** a hyperosmotic shock and exponential growth at 0.3 M NaCl. **g** The difference in the transcript level of the *proV*1 amplicon compared to the *proU*1 amplicon during exponential growth at 0.08 M NaCl, following a hyperosmotic shock, and during exponential growth at 0.3 M NaCl. **h** The transcript levels of amplicons flanking *proVWX* during exponential growth at 0.08 M NaCl, following a hyperosmotic shock, and during exponential growth at 0.3 M NaCl. Y-axes: All bar graphs and data points with error bars show relative transcript levels in arbitrary units and are plotted on the left y-axis. Plots without error bars show fold changes in transcript levels and correspond to the right y-axis. Internal control: *rpoD*. See also Supplementary Fig. 1. Data (**1a–h**) are presented as mean values +/- standard deviation. Dot plots (**1a–c**, **1h**): *n* = 3 technical replicates of a biologically independent culture. Bar graphs (**1a–h**): *n* = 4 biologically independent cultures. Source data are provided as a Source Data file.

of K[+] that occurs as an initial response to increased extracellular osmolarity[34,36–41]. In vitro studies support a model where the influx of K[+] drives a conformational change in the H-NS−DNA nucleoprotein structure at the NRE of *proVWX* from a transcriptionally repressive DNA−H-NS−DNA bridge to a transcriptionally-conducive H-NS−DNA filament[24,27].

In an NT331 culture growing exponentially in a high salt environment of 0.3 M NaCl, the transcript levels of *proVWX* are ~1.5- to 2.5-fold higher than during exponential growth at 0.08 M NaCl, and up to 4.5-fold lower than transcript levels following a hyperosmotic shock

(Fig. 1e, f and Supplementary Fig. 1e, f). Exponential growth at 0.3 M NaCl reflects the adaptation of *E. coli* to increased extracellular osmolarity brought about by the import of osmoprotectants, the synthesis of trehalose, and the export of K[+] [36,46]. The latter of these, in particular, re-instates repression[34].

The dynamics of the transcript levels of *proVWX* during osmotic stress suggests that an increase of *proVWX* transcript levels upon a hyper-osmotic shock may rapidly establish the required number of ProU transporters. The decreased transcript levels of *proVWX* upon adaptation to hyper-osmotic stress maintain the necessary levels of

ProU. A higher transcript level of *proVWX* during exponential growth at 0.3 M NaCl compared to 0.08 M NaCl (Fig. 1e and Supplementary Fig. 1e) may reflect the increased requirement of the ProU transporter at higher osmolarity.

The fold changes in the transcript levels of the constituent genes of *proVWX* differ upon changes to ambient osmolarity. Following a hyperosmotic shock, the transcript level of *proV* increases ~7.5-fold, the P2 proximal amplicon of *proW* (amplicon *proW*1) increases ~2-fold, the P2 distal amplicon of *proW* (amplicon *proW*2) is ~7-fold higher, and *proX* transcripts increase by a factor of ~6.5. During exponential growth at 0.3 M NaCl, in comparison to 0.08 M NaCl, the transcript levels of *proV*, *proW*1, *proW*2, and *proX* increase by factors of ~1.8, ~1.3, ~2.2, and ~1.5, respectively (Fig. 1d–e, and Supplementary Fig. 1d–e). Differences in the fold-increase or fold-decrease of *proV*, *proW*, and *proX* transcripts in *E. coli* upon changes to growth conditions have also been observed in genome-wide studies[80]. The results show that the *proV*, *proW*, and *proX* genes that occur within the same operon and are co-regulated by the mechanisms operating on P2 exhibit a degree of independent regulation. The presence of additional regulatory mechanisms operating on individual genes of an operon has been documented previously, for instance, at the *gal*[89] and *glmUS*[90] operons.

Transcription of *proVWX* primarily initiates from its σ70-dependent (P2) promoter positioned 60 bp upstream of the *proV* ORF. The *proVWX* σ$^S$-dependent (P1) promoter located at −250 bp of *proV* is cryptic and is activated by *rho* and *hns* mutations, and by cold stress[47]. To record the contribution, if any, of P1 to the expression of *proVWX* in a wild-type NT331 background, the relative transcript level of amplicon *proU*1 positioned between the P1 and P2 promoters was determined. At 0.08 M NaCl, transcript level of *proU*1 was ~6-fold lower than that of amplicon *proV*1 positioned 345 bp downstream of P2. Upon a hyperosmotic shock, the transcript level of *proU*1 increased ~2-fold, but was ~25-fold lower than that of *proV*1. During exponential growth at 0.3 M NaCl, the transcript level of *proU*1 was ~10-fold lower than *proV*1 (Fig. 1g and Supplementary Fig. 1g). This indicates that while the osmo-response of *proVWX* primarily arises from initiation at P2[67,68,91], a contribution of P1 may ensue from the dismantling of the H-NS−DNA nucleoprotein during the K$^+$ influx, to form a nucleoprotein structure that mimics an *hns* knock-down phenotype. Such a structure would alleviate Rho-dependent termination of transcripts initiated from P1[47,48,80]. P1 can contribute to the increased expression of the operon during a hyper-osmotic shock. Transcription from P1 across P2 may favour the relief of H-NS repression at the NRE[79,92].

To detect cross-talk between the expression of *proU* and its flanking genes, the relative transcript levels of *nrdF* and *ygaY* positioned upstream and downstream of *proU*, respectively, were determined. In NT331, the transcript levels of *ygaY* and *nrdF* were comparable for growth at 0.08 M NaCl, 0.3 M NaCl, and following a hyper-osmotic shock (Fig. 1h and Supplementary Fig. 1h; Supplementary Tables 1 and 2), indicating insulation from flanking chromatin. It is important to note, however, that the expression of the *nrdF* amplicon proximal to *proU* (amplicon *nrdF*2) and both *ygaY* amplicons (amplicons *ygaY*1 and *ygaY*2) were detectably higher after a hyperosmotic shock. This may occur due to an increased local density of RNAP at this osmotic condition or the decompaction of the local chromatin following the shock providing a more conducive environment for transcription.

## The transcriptional profile of *proVWX* upon alleviation of H-NS-mediated repression at the DRE

H-NS-mediated *proU* repression was alleviated by introducing a series of point mutations in the AT-rich H-NS-binding regions that were experimentally detected in the *proU* DRE with an in vitro DNase-I protection assay[55]. This approach was chosen over *hns* deletion to prevent the pleiotropic effects associated with the latter. In this design, the −35 and −10 promoter elements of P2, the P2 TSS, and the Shine-Dalgarno sequence of *proVWX* were not edited. However, four mutations were introduced in the high-affinity H-NS binding site extending between −7 and +15 of the P2 TSS. AA > GC and TA > GC mutations were carried out at −3 to −2 and at +7 to +8 positions, respectively. AT-rich codons in H-NS binding sites in the *proV* ORF were switched out for GC-rich variants with the most similar codon usage ratio in the *E. coli* chromosome. Identical mutations decided upon with the aforementioned criteria, were incorporated into both high-affinity H-NS binding sites occurring in the DRE[56]. Following mutation, the AT-content of the DRE was lowered from 66.4% to 51.6% (Supplementary Fig. 4; Supplementary Data 1A). The strain was labelled NT644. The decrease in affinity of the mutated DRE for H-NS was validated in vitro using an electrophoretic mobility shift assay (Supplementary Fig. 5).

Alleviating H-NS mediated repression – a concept first introduced by van Ulsen et al.[93] – at the *proU* operon raises the transcript levels of its constituent genes at 0.08 M NaCl by up to ~6.5-fold (Fig. 2a, b and Supplementary Fig. 6a, b; Supplementary Tables 3 and 4). This is in line with observations by Nagarajavel et al., 2007, where, in a Δ*hns* background at 0.05 M, and 0.1 M NaCl, the β-galactosidase activity of a *proV-lacZ* fusion expressed from a construct cloned downstream of the P2 promoter, URE, and DRE on a plasmid was ~9- and ~4-fold higher than in the wild-type background[49].

The relative transcript level of *proVWX* in NT644 at low osmolarity (0.08 M NaCl) is comparable to the relative transcript level in the wild-type background (NT331) after a hyper-osmotic shock (from 0.08 M to 0.3 M NaCl) (Fig. 2c and Supplementary Fig. 6c; Supplementary Tables 1–4) and not NT331 at high osmolarity (Fig. 2d and Supplementary Fig. 6d; Supplementary Tables 1–4). This appears to contradict the *proVWX* osmoregulation studies reported earlier[49] which show that the specific activity of β-galactosidase of a *proV-lacZ* fusion in a Δ*hns* background at low osmolarity (0.05 M, and 0.1 M NaCl) is similar to the specific activity of proV-β-galactosidase in a wild-type background at high osmolarity (growing exponentially at 0.3 M NaCl). The differences between these two sets of results may be accounted for by the post-translational evaluation of osmoresponse in ref. 49. β-galactosidase is a stable protein. Hence, expression analyses using *lacZ* reporter fusions[49] mask down-shifts of expression upon adaptation to high osmolarity.

A hyper-osmotic shock to NT644, increases the transcript levels of *proVWX* by ~2-fold (Fig. 2e, f and Supplementary Fig. 6e, f), and upon adaptation to the hyperosmotic stress, represented by exponential growth at 0.3 M NaCl, *proVWX* transcript levels decline to as low as up to ~25% of those at 0.08 M NaCl (Fig. 2g, h and Supplementary Fig. 6g, h). Our results also show that NT644 (H-NS-deficient *proU* regulatory element) and NT331 (wild-type) have similar transcript levels of *proU* at 0.3 M NaCl (Fig. 2i and Supplementary Fig. 6i; Supplementary Tables 1–4). This suggests that in *E. coli* cells adapted to higher osmolarities, the repression of *proU* may be mediated by factors other than H-NS occupancy at the *proU* promoter, for instance, RNaseIII-mediated processing and subsequent degradation (see below).

Expression from P1 is repressed in a Δ*hns* background[39,50]. The mutations at the *proVWX* regulatory element in NT644 are designed to disrupt H-NS binding to the DRE (Supplementary Fig. 4; Supplementary Data 1A). H-NS occupancy at the URE, and effectively, at P1, (Supplementary Fig. 3) should remain unaffected. Indeed, the relative transcript level of the *proU*1 amplicon was similar between NT331 and NT644 at the osmolarity conditions tested here (Fig. 2j and Supplementary Fig. 6j; Supplementary Tables 1–4).

Disruption of H-NS binding to the DRE did not affect the relative transcript levels of the flanking *nrdF* and *ygaY* genes. The transcript levels were similar between NT331 and NT644 during growth at 0.08 M NaCl, 0.3 M NaCl, and following a hyperosmotic shock (Figs. 1h, 2k and Supplementary Fig. 1h, 6k; Supplementary Tables 1–4).

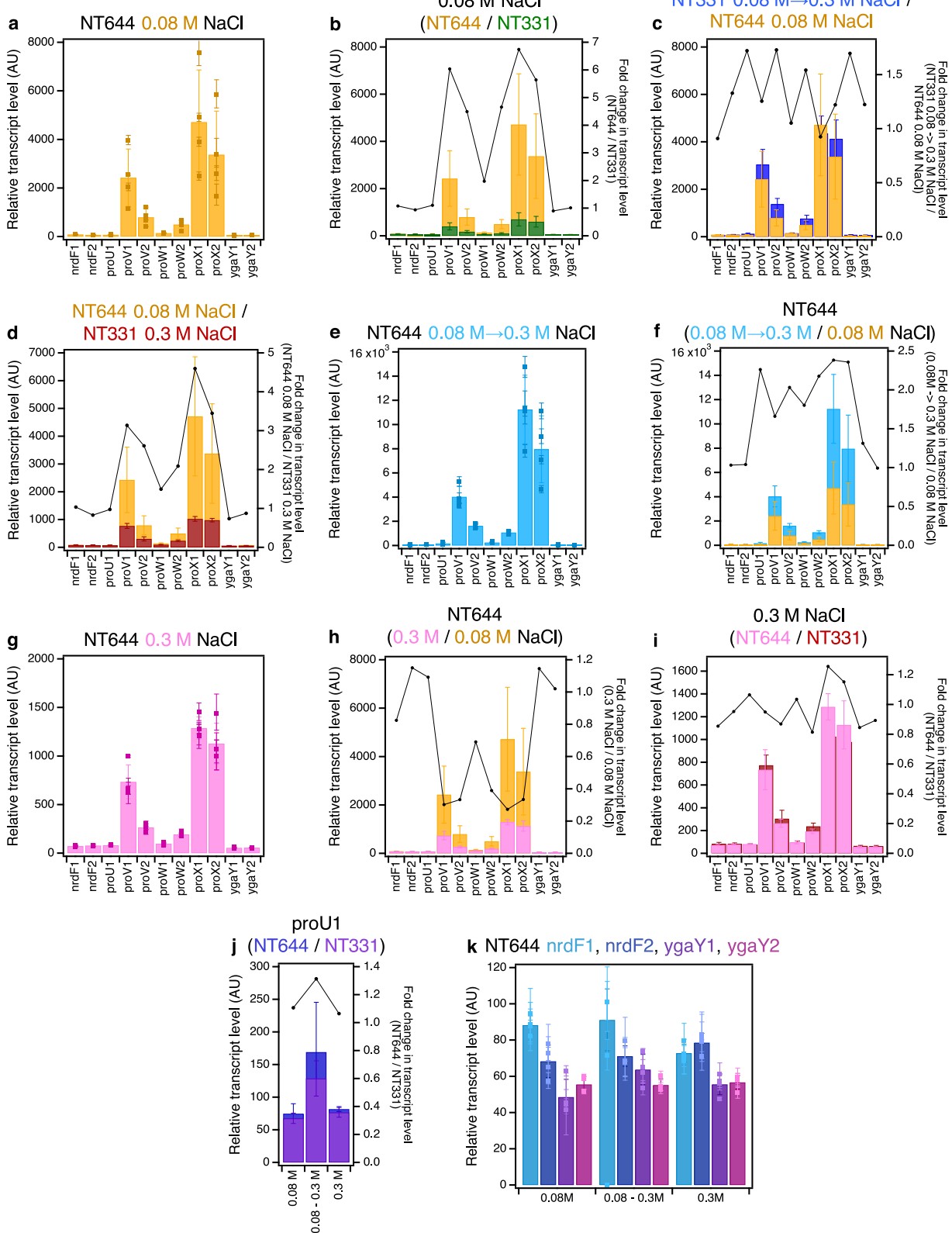

## The role of StpA in the regulation of *proVWX*

StpA is a paralogue of H-NS that complements the protein in Δ*hns* strains[70-72]. StpA stimulates the formation of DNA−H-NS−DNA bridges in vitro and stabilises the structure against changes in temperature and Mg²⁺ or K⁺ concentration[94]. ChIP studies reveal that H-NS-bound regions of the chromosome contain StpA[74]. At the *proVWX* operon, StpA occupies two sites within the NRE that overlap with H-NS-bound

regions. The first site encompasses the P1 promoter and the second lies downstream of P2 and spans across the *proV* structural gene. For ease of communication, these sites are henceforth referred to as Site P1 and Site *proV*. H-NS occupancy at *proVWX* appears to be unaffected in a Δ*stpA* background. In Δ*hns* strains, StpA only occupies Site P1 at *proVWX* – the nucleoprotein structure at Site *proV* is lost (Supplementary Fig. 3)[74]. This binding pattern may explain why the *proVWX*

**Fig. 2 | *ProU* transcription upon alleviation of H-NS-mediated repression.** The RT-qPCR profile of *proVWX* and its flanking regions in NT644 **a** during exponential growth in M9 medium with 0.08 M NaCl. A comparison of transcript levels of *proVWX* and its flanking regions **b** between NT644 and NT331 during exponential growth in M9 medium with 0.08 M NaCl, **c** between NT644 growing exponentially at 0.08 M NaCl and NT331 subjected to a hyperosmotic shock from 0.08 M to 0.3 M NaCl, and **d** between NT644 at 0.08 M NaCl and NT331 at 0.3 M NaCl. The RT-qPCR profile of *proVWX* and its flanking regions in NT644 **e** after a hyperosmotic shock from 0.08 M to 0.3 M NaCl, and **f** the fold change in transcript levels compared to exponential growth at 0.08 M NaCl. The RT-qPCR profile of *proVWX* and its flanking regions **g** during exponential growth in M9 medium with 0.3 M NaCl, and a comparison of this profile with that of **h** exponential growth of NT644 in M9 medium with 0.08 M NaCl, and **i** exponential growth of NT331 in M9 medium with 0.3 M

NaCl. **j** The fold change in transcript level of the *proU*1 amplicon between NT644 and NT331 during exponential growth at 0.08 M NaCl, following a hyperosmotic shock, and during exponential growth at 0.3 M NaCl. **k** The relative transcript level in NT644 at amplicons flanking *proVWX* during exponential growth at 0.08 M NaCl, following a hyperosmotic shock, and during exponential growth at 0.3 M NaCl. Y-axes: All bar graphs and data points with error bars show relative expression levels in arbitrary units and are plotted on the left y-axis. Plots without error bars show fold-change in expression level and correspond to the right y-axis. Internal control: *rpoD*. See also Supplementary Fig. 6. Data (**2a–k**) are presented as mean values +/- standard deviation. Dot plots (**2a, 2e, 2g, 2k**): n = 3 technical replicates of a biologically independent culture. Bar graphs (**2a–k**): n = 4 biologically independent cultures. Source data are provided as a Source Data file.

operon is only repressed by StpA when StpA is over-expressed in a Δ*hns* background[73]. However, the role of StpA in *proVWX* regulation in an otherwise wild-type background is unclear. Therefore, we investigated the effect of *stpA* deletion on the osmoresponse of *proVWX* in NT331.

At 0.08 M NaCl, *stpA* deletion elevates the relative transcript levels of all amplicons within the *proU* structural genes – except *proV*2 and *proW*1 – by ~4-fold (Fig. 3a, b and Supplementary Fig. 7a, b; Supplementary Tables 1, 2, 5 and 6). In vitro observations of the architectural properties of StpA-deficient and StpA-supplemented H-NS−DNA structures[94] suggest that this rise in expression may be driven by an increased sensitivity of the repressive H-NS−DNA nucleoprotein at *proVWX* to intracellular K⁺ concentrations. The relative transcript level of amplicon *proV*2, positioned on the *proV* ORF distal to the P1 and P2 promoters was ~7-fold higher upon *stpA* deletion, compared to the ~4-fold increase for amplicon *proV*1 that is positioned closer to the promoters (Fig. 3a, b and Supplementary Fig. 7a, b). From a different perspective, the average transcript level of *proV*2 at 0.08 M NaCl in NT331 was 44% of that of amplicon *proV*1. In NT331 Δ*stpA*, this value rose to 69%. This reflects an increased processivity of RNAP on the *proV* ORF in the absence of StpA. This finding, in context of in vitro studies on the role of StpA in transcription regulation[94], suggests that StpA-deficient H-NS−DNA nucleoprotein complexes that form over the *proV* ORF, and by extension, on other sites throughout the chromosome, function as weaker transcription roadblocks than StpA-supplemented H-NS−DNA structures. The relative transcript level of the *proW*1 amplicon was similar in NT331 and NT331 Δ*stpA* (Fig. 3a, b and Supplementary Fig. 7a, b; Supplementary Tables 1, 2, 5, and 6). This may be attributed to the strength of the terminator that lies between *proV* and *proW* (Supplementary Fig. 2)[82].

A hyperosmotic shock to NT331 Δ*stpA* from 0.08 M to 0.3 M NaCl increased the relative transcript level of *proV* and *proW* by ~14- to ~20-fold, and *proX* by ~10-fold (Fig. 3c, d and Supplementary Fig. 7c, d; Supplementary Tables 5 and 6). In NT331, the increase stands at less than 8-fold (Fig. 1b, d and Supplementary Fig. 1b, d; Supplementary Tables 1 and 2). NT331 Δ*stpA* cells adapted to growth at 0.3 M NaCl show *proU* transcript levels that are ~3- to ~4-fold higher than cells growing at 0.08 M NaCl and ~2.5- to ~4-fold lower than cells subjected to a hyperosmotic shock (Fig. 3e–g and Supplementary Fig. 7e–g; Supplementary Tables 5 and 6). These data support in vitro observations that StpA-deficient H-NS−DNA nucleoprotein filaments have an increased osmosensitivity and lowered osmostability such that the structure is much more efficiently dismantled by K⁺[94].

Transcription initiation from P1 was lower in Δ*stpA* mutants compared to wild-type cells during exponential growth at 0.08 M and 0.3 M NaCl (Fig. 3h and Supplementary Fig. 7h). This matches observations that expression from P1 is decreased in Δ*hns* strains[39,50]. How a generally repressive nucleoprotein complex of H-NS and StpA enhances expression from a promoter encapsulated within its structure is unclear. The puzzle is further complicated with the paradoxical observation that upon a hyperosmotic shock, a condition that

dismantles H-NS−DNA complexes, expression from P1 increases and is similar between NT331, and NT331 Δ*stpA*. This may be accounted for by the inherent osmosensitivity of P1 – expression from P1 increases with increasing osmolarity in vitro[39,50].

The transcript levels of *nrdF* and *ygaY* in NT331 Δ*stpA* were affected by osmolarity. At 0.08 M NaCl, transcripts of both genes were reduced to ~50% that of wild-type cells (Fig. 3i, j, and Supplementary Fig. 7i, j). A perceptible increase in transcript levels at 0.3 M NaCl was observed, which was still maintained at ~50% of that in the wild-type background when expression was normalised relative to *rpoD* (Fig. 3i, j; Supplementary Tables 1 and 5). When normalisation was performed using *hcaT* as an internal control, transcript levels were comparable to the wild-type background (Supplementary Fig. 7i, j; Supplementary Tables 2 and 6). Upon a hyperosmotic shock to NT331 Δ*stpA*, the transcript levels of *nrdF*, positioned upstream of *proU*, increased ~2-fold to a level comparable to that of NT331 under the same condition. Moreover, the transcript levels of *ygaY*, positioned downstream of *proU*, increased 11-fold at the *ygaY*1 amplicon positioned proximal to the terminus of *proX* and 4-fold at the *ygaY*2 amplicon positioned further downstream, corresponding to an increase of 3.8-fold and 1.6-fold, respectively, compared to the wild-type strain (Fig. 3k and Supplementary Fig. 7k). This implies that StpA insulates *proVWX* from its flanking operons, perhaps by constraining supercoils[72]. Indeed, a weak ChIP signal for FLAG-tagged StpA is observed at the terminus of the *proVWX* operon (Supplementary Fig. 3)[74]. The decline in transcript levels between *ygaY*1 and *ygaY*2 may be a consequence of the crypticity of *ygaY*[95,96]. Nine in-frame stop codons occur between the two amplicons (Supplementary Data 1A).

## The role of RNaseIII in regulating the expression of *proVWX*

RNaseIII, encoded by the *rnc* gene, regulates the expression of *proVWX* at a post-transcriptional level. In hypoosmotic conditions and upon a hypoosmotic shock, RNaseIII downregulates *proVWX* by cleaving a conserved secondary structure that forms at position +203 to +293 of the *proU* mRNA transcribed from P2. High osmolarity conditions inhibit the action of RNaseIII on the *proU* mRNA[54]. The regulatory role of RNaseIII on *proVWX* expression was evaluated using a construct carrying the −315 to +1260 region centred around the *proU* P2 TSS – a region was expected to contain all essential regulatory elements of *proU*[54]. The construct was cloned into a plasmid or inserted at an ectopic *attB* locus on the chromosome in a Δ*proVWX* background for study[54]. We built on this report and determined the effect of a Δ*rnc* mutation on the regulation of the entire *proVWX* operon in its native chromosome context.

Transcript levels of the *proVWX* operon were expressed perceptibly higher in NT331 Δ*rnc* compared to NT331 during exponential growth at 0.08 M NaCl (Fig. 4a, b and Supplementary Fig. 8a, b; Supplementary Tables 1, 2, 7 and 8). The fold increase in the relative transcript level of amplicon *proV*2 – positioned distal to the P1 and P2 promoters – was higher than that of the *proV*1 amplicon positioned closer to P1 and P2 (Fig. 4b and Supplementary Fig. 8b). This indicates

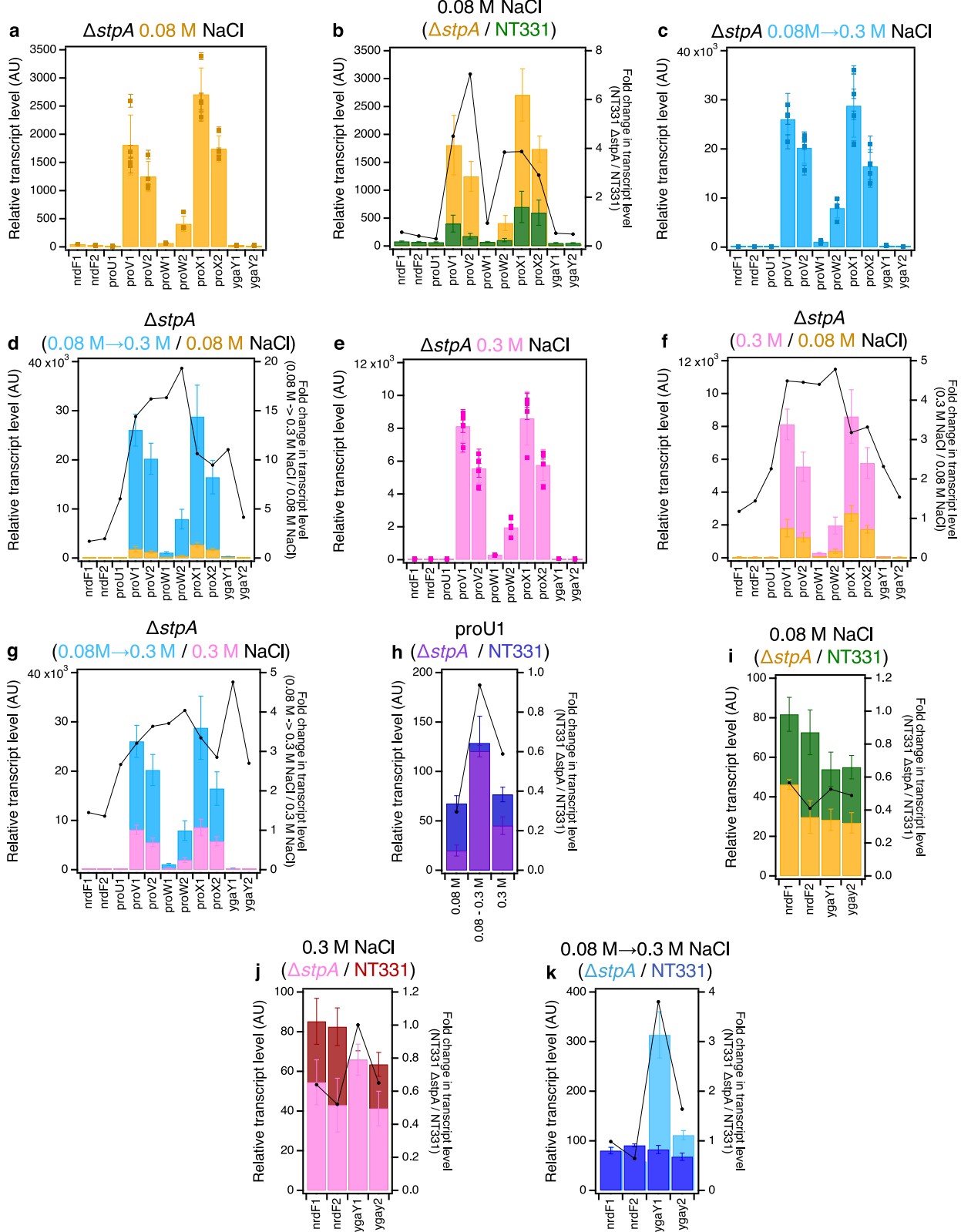

an increased stability of the 3' region of *proV* transcripts in the absence of *rnc*, suggesting that RNaseIII contributes to processing of the 3' end of the transcript. This can be a secondary event involving other RNases such as RNaseE. Processing of the conserved hairpin located at +203 to +293 of the *proU* transcript generates a monophosphorylated 5' end. Such termini can be recognised and bound by RNaseE, promoting degradation[97]. However, it is also possible that an RNaseIII processing

site may be present proximal to the 3' end of *proV* transcripts. Interestingly, the transcript level of *proX* in NT331 Δ*rnc* at 0.08 M NaCl was reduced to ~50% of that in NT331 (Fig. 4b and Supplementary Fig. 8b). How the knock-out of an RNase could trigger such an effect is unclear.

A hyperosmotic shock from 0.08 M to 0.3 M NaCl, increased the transcript levels of *proV* and *proX* by ~15-fold, *proW*1 by ~5-fold, and *proW*2 by ~11-fold (Fig. 4c, d and Supplementary Fig. 8c, d;

**Fig. 3 | Transcription of *proU* in the absence of StpA.** The RT-qPCR profile of the *proVWX* operon and its flanking regions in NT331 Δ*stpA* **a** during exponential growth at 0.08 M NaCl and **b** the fold-change in the transcript levels of the amplicons compared to NT331. **c** The RT-qPCR profile of the *proVWX* operon and its flanking regions in NT331 Δ*stpA* upon a hyperosmotic shock from 0.08 M to 0.3 M NaCl, and **d** the fold-change in transcript levels of the amplicons in comparison to exponential growth at 0.08 M NaCl. **e** The RT-qPCR profile of the *proVWX* operon and its flanking regions in NT331 Δ*stpA* during exponential growth at 0.3 M NaCl, and the fold-change in transcript levels of the amplicons with respect to **f** exponential growth at 0.08 M NaCl, and **g** a hyperosmotic shock. **h** The fold difference in transcript level of amplicon *proU*1 between NT331 Δ*stpA* and NT331.

The fold change in transcript levels of the *nrdF* and *ygaY* amplicons between NT331 Δ*stpA* and NT331 **i** during exponential growth at 0.08 M NaCl, **j** exponential growth at 0.3 M NaCl, and **k** following a hyperosmotic shock. Y-axes: All bar graphs and data points with error bars show relative transcript levels in arbitrary units and are plotted on the left y-axis. Plots without error bars show fold-change in transcript level and correspond to the right y-axis. Internal control: *rpoD*. See also Supplementary Fig. 7. Data (3**a**–**k**) are presented as mean values +/- standard deviation. Dot plots (3**a**, 3**c**, 3**e**): *n* = 3 technical replicates of a biologically independent culture. Bar graphs (3**a**–**k**): *n* = 4 biologically independent cultures. Source data are provided as a Source Data file.

Supplementary Tables 7 and 8). In comparison to NT331, the relative transcript levels of *proV* and *proW* were >2.4-fold higher in NT331 Δ*rnc* (Fig. 4e and Supplementary Fig. 8e). In contrast, the relative expression of *proX* upon a hyperosmotic shock was similar for NT331 and the Δ*rnc* mutant (Fig. 4e and Supplementary Fig. 8e; Supplementary Tables 1, 2, 7 and 8). These results emphasise that the co-regulated genes of *proVWX* have some degree of independent regulation, and that this also occurs at the post-transcriptional level. The results indicate that during a hyper-osmotic shock RNaseIII still processes *proU* mRNA, and that RNaseIII processes transcripts initiating from P1 and P2 but does not appear to interfere with transcripts that potentially originate from the internal promoter within the *proW* ORF. The data support earlier findings[54] that implicate a hairpin at the 5' end of *proU* transcripts in signalling RNaseIII-mediated processing. The fold change in transcript levels between NT331 and NT331 Δ*rnc* for the *proV*1 and *proV*2 amplicons, is ~2.5- and ~5-fold, respectively (Fig. 4e and Supplementary Fig. 8e), reinforcing the proposal that RNaseIII-mediated post-transcriptional regulation of *proU* expression may involve processing at a site proximal to the 3' ends of *proV* transcripts.

During exponential growth at 0.3 M NaCl, the transcript levels of *proV* and *proX* are 7.5- to 8.5-fold higher in comparison to growth at 0.08 M NaCl; *proW*1 is increased by ~3-fold and *proW*2, ~5.5-fold (Fig. 4f, g and Supplementary Fig. 8f, g). The increase is in line with expectations of *proU* induction at higher osmolarities. The relative transcript levels of *proW* and *proX* in the Δ*rnc* mutant at 0.3 M NaCl are ~2-fold higher than in NT331. *proV*1 is ~5-fold more abundant and proV2, ~11-fold (Fig. 4h and Supplementary Fig. 8h). This indicates that RNaseIII contributes to downregulating the expression of *proU* even upon adaptation to higher osmolarity. The function of RNaseIII upon adaptation to high osmolarity compared to growth at low osmolarity may explain why the *proVWX* transcript levels in NT644 during exponential growth at 0.3 M NaCl are as low as up to ~25% that at 0.08 M NaCl (Fig. 2g, h and Supplementary Fig. 6g, h). The increase in the relative transcript level of *proX* in NT331 Δ*rnc* at 0.3 M NaCl with respect to NT331 (Fig. 4h and Supplementary Fig. 8h), implicates RNaseIII in *proX* processing. This is in contrast of *proX* expression at 0.08 M NaCl where RNaseIII knock-out represses the gene.

In NT331 Δ*rnc*, transcription from P1 is comparable to the wild-type background (Fig. 4i and Supplementary Fig. 8i). *nrdF* and *ygaY* are insulated from *proU* and exhibit similar transcript levels at high and low osmolarity that are comparable to the wild-type (Fig. 4j and Supplementary Fig. 8j; Supplementary Tables 1, 2, 7 and 8).

## The local three-dimensional structure of *proVWX* can be studied using 3C-qPCR

The transcription of *proVWX* is regulated by an osmoresponsive architectural protein (H-NS). Indeed, chromosome contact maps of NT331 generated with Hi-C show chromosome rearrangements in the vicinity of the *proVWX* operon (Supplementary Fig. 9). However, Hi-C in bacteria is limited by its resolution. Chromosomal gene density, the scarcity of intergenic DNA, and the short length of individual genes and operons means that an *E. coli* Hi-C map with, in our hands, a 10 kb resolution cannot be used to study the structures of individual operons, identify the underlying genetic features that

encode global and finer structural changes to the chromosome, or the significance of these changes. To study the interplay between the expression of *proVWX* and local chromatin architecture, we adapted the 3C-qPCR technique[98] to a bacterial system[99], and used it to examine the organisation of the operon and a part of its flanking regions at low salt, hyper-osmotic shock, and high salt conditions in NT331, NT644, and rifampicin-treated NT331 cells. These represent wild-type, H-NS-deficient, and transcription-deficient systems, respectively.

3C-qPCR is a one-to-one technique that probes the relative interaction frequency between two loci, one of which is an anchor fragment, and the other a variable test fragment[98,99]. 3C-qPCR affords a resolution of individual restriction digestion fragments and is, therefore, suitable for high resolution studies of local chromatin remodelling. 3C libraries prepared using NlaIII as a restriction enzyme were ideal for our study since NlaIII digests *proVWX* in a manner that separates the URE from the DRE, and the high-affinity H-NS binding sites in the DRE from each other, allowing the possibility of interrogating the three-dimensional organisation of the operon using individual regulatory elements as anchor sites. We used fragment proU3_NlaIII that contains the *proVWX* P2 promoter and TSS as an anchor and determined its relative interaction frequency with other restriction digestion fragments within a 7 kb region centred around *proVWX* (Supplementary Data 1A).

To account for differences in the restriction digestion and ligation efficiencies of 3C libraries, and the differences in the starting quantity of 3C libraries in a qPCR reaction, interaction frequencies calculated from 3C-qPCR experiments are normalised to an Internal control[98]. Due to a lack of 3C-qPCR studies in bacteria, we tested the applicability of several interactions as internal controls. We considered an interaction to be a reliable internal control if normalisation resulted in the data points of all the test interactions (from proU3-proU1 up to proU3-proU17) for corresponding biological replicates (R1-R4) overlapping with each other within error. We tested four interactions that form in the *rpoD* and *hcaT* genes that have been used as internal controls in the RT-qPCR experiments reported in this article, and two additional interactions in *rpoB* that was not a reliable RT-qPCR internal control for our experimental conditions, but has been used previously. Each of these resulted in the misalignment of one or more of the biological replicates of a set of 3C libraries. This may arise since several factors contribute to regulating gene expression, and while the transcription output of *rpoD* and *hcaT*, or *rpoB* in other cases, may remain unchanged, chromatin architecture may differ. The *Escherichia coli* chromosome is organised into four structured, and two unstructured macrodomains. Since the ligation efficiency between two fragments is not only influenced by their proximity to each other, but also by their proximity to other loci, we inferred that chromatin interactions in one macrodomain may not be a reliable internal control for chromatin structure studies in another macrodomain, or perhaps even another sub-domain in the same macrodomain. Thus, we checked if a test interaction within our region of observation could serve as an internal control. The proU3-proU6 interaction best-fulfilled the criteria we set for a reliable internal control.

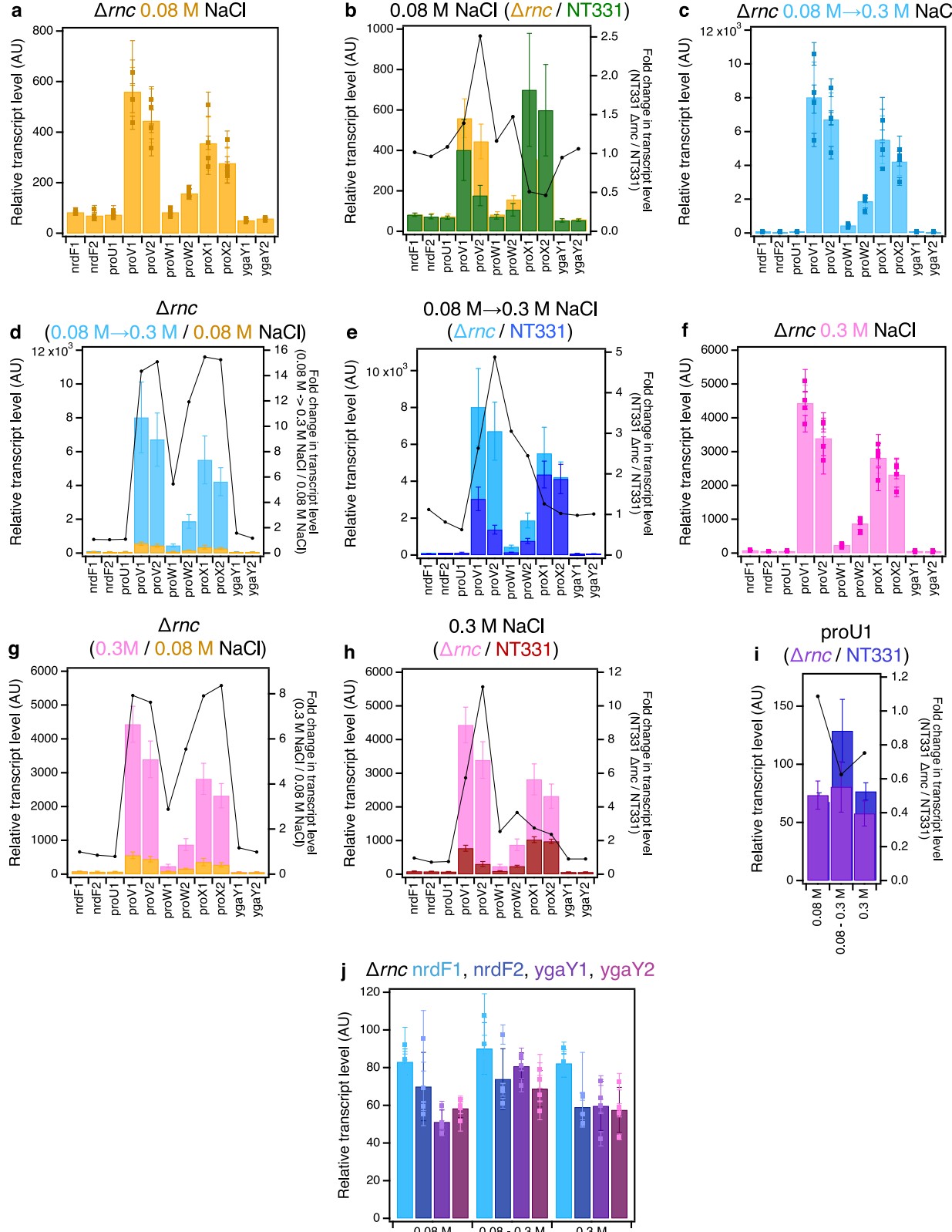

## Structural re-modelling of *proVWX* in response to osmolarity

The transcription profile of *proVWX* in NT331 differs in cultures growing exponentially in M9 medium with 0.08 M NaCl (low salt), cultures subjected to a hyperosmotic shock (salt shock), and cultures growing exponentially in a medium with 0.3 M NaCl (high salt) (Fig. 1a–c and Supplementary Fig. 1a–c). 3C-qPCR studies show that each of these conditions is also associated with a different structural

profile of the operon (Fig. 5a–c; Supplementary Table 9), indicating that a change in the expression of *proVWX* in response to osmolarity is associated with a change in its three-dimensional organisation.

A hyperosmotic shock from 0.08 M NaCl to 0.3 M NaCl increases the relative transcript level of *proVWX* in NT331 by ~2–8-fold (Fig. 1d and Supplementary Fig. 1d). The change is associated with a decrease in the relative interaction frequency between proU3_NlaIII and the local

**Fig. 4 | Transcription of *proU* in an RNaseIII deletion mutant.** The RT-qPCR profile of *proVWX* and its flanking regions in NT331 Δ*rnc* during **a** exponential growth at 0.08 M NaCl, and the fold change in transcript levels of the amplicons **b** compared to NT331 growing exponentially at 0.08 M NaCl. The RT-qPCR profile of *proVWX* and its flanking regions in NT331 Δ*rnc* upon **c** a hyperosmotic shock from 0.08 M NaCl to 0.3 M NaCl, and the fold change in transcript levels of the amplicons in comparison to **d** NT331 Δ*rnc* growing exponentially at 0.08 M NaCl, and **e** NT331 following a hyperosmotic shock. The RT-qPCR profile of *proVWX* and its flanking regions in NT331 Δ*rnc* during **f** exponential growth at 0.3 M NaCl, and the fold change in transcript levels of the amplicons relative to **g** NT331 Δ*rnc* growing exponentially at 0.08 M NaCl, and **h** NT331 growing exponentially at 0.3 M

NaCl. **i** The fold change in the relative transcript levels of the *proU*1 amplicon between NT331 Δ*rnc* and NT331. **j** The relative transcript level in NT331 Δ*rnc* at amplicons flanking *proVWX* during exponential growth at 0.08 M NaCl, following a hyperosmotic shock, and during exponential growth at 0.3 M NaCl. Y-axes: All bar graphs and data points with error bars show relative transcript levels in arbitrary units and are plotted on the left y-axis. Plots without error bars show fold-change in transcript level and correspond to the right y-axis. Internal control: *rpoD*. See also Supplementary Fig. 8. Data (**4a–j**) are presented as mean values +/- standard deviation. Dot plots (**4a, 4c, 4f, 4j**): *n* = 3 technical replicates of a biologically independent culture. Bar graphs (**4a–j**): *n* = 4 biologically independent cultures. Source data are provided as a Source Data file.

chromatin positioned upstream of *proVWX* (shaded region, Fig. 5d), suggesting that the activation of the operon involves decompaction of local chromatin. Specific decreases in relative interaction frequency between proU3_NlaIII and proU2_NlaIII (arrow I, Fig. 5d), and between proU3_NlaIII and proU11_NlaIII (arrow II, Fig. 5d) are also observed. This implies that an increase in the expression of *proVWX* also involves, firstly, a decrease in the physical interaction between the high affinity H-NS binding site of the DRE contained in fragment proU3_NlaIII and the URE contained in proU2_NlaIII that also exhibits an H-NS binding preference. Since 3C-qPCR is an ensemble technique, a decrease in relative interaction frequency may also imply a reduction in the population of cells in which the DRE and URE physically interact. This observation complements earlier reports that the URE and DRE function cooperatively to regulate *proVWX*[49,57]. Secondly, the increase in *proVWX* expression is associated with a decrease in the interaction between the DRE and *ygaY* contained in proU11_NlaIII. This regulatory feature was previously unknown.

During exponential growth of NT331 at 0.3 M NaCl, a condition that resembles adaptation to hyperosmotic stress, the expression of *proVWX* decreases by ~4-fold (Fig. 1f and Supplementary Fig. 1f) to a level that is 1.2- to 2.2-fold higher than exponential growth at 0.08 M NaCl (Fig. 1e and Supplementary Fig. 1e). The 3C-qPCR profile of *proVWX* in NT331 during exponential phase at 0.3 M NaCl shows a high relative interaction frequency between proU3_NlaIII and the chromatin positioned upstream of the operon, showing a greater resemblance to the 3C-qPCR profile of *proVWX* in NT331 at 0.08 M NaCl (shaded region, Fig. 5e) than the profile following a hyperosmotic shock (shaded region, Fig. 5f). The interaction between proU3_NlaIII and proU2_NlaIII increases upon adaptation to hyperosmotic stress (arrow I, Fig. 5f) but is detectably lower during exponential growth at 0.3 M NaCl than at 0.08 M NaCl (arrow I, Fig. 5e). The interaction between proU3_NlaIII and proU11_NlaIII in cells adapted to high osmolarity is higher than that of cells subjected to a hyperosmotic shock (arrow II, Fig. 5f), and comparable within error to cells growing at low osmolarity (arrow II, Fig. 5e).

These results show that the proU3_NlaIII fragment that carries a high-affinity H-NS binding site of the DRE interacts with the URE and *ygaY*. The interactions are disrupted during a hyperosmotic shock (arrows I and II, Fig. 5d) – a condition that involves a rapid intracellular import of K⁺ – and recovered in cells adapted to hyperosmotic stress (arrows I and II, Fig. 5e) that is associated with the replacement of K⁺ with inert osmoprotectants. These observations, in combination with in vitro experiments on the mechanistic role of H-NS[24,27], sketch a model where K⁺-dependent conformational changes of the H-NS multimer at the DRE regulate *proVWX* by modifying the three-dimensional organisation of the operon. The prevalence of a road-blocking, bridging conformation favoured under conditions of low intracellular K⁺ during exponential growth at 0.08 M NaCl, and upon adaptation to hyperosmotic stress represses the operon, while, the formation of transcriptionally-conducive H-NS−DNA filaments in response to high intracellular K⁺ concentrations following a hyperosmotic shock de-represses it[24,27,39,49,51,55,56].

## The role of H-NS in the regulation of *proVWX* by chromatin re-modelling

The URE and DRE are known to bind H-NS in in vitro EMSA[56] and in vivo ChIP assays[74,85]. However, presence of a detectable H-NS signal at *ygaY* in some ChIP experiments (Supplementary Fig. 3)[74] and its absence in others[85] makes the presence and regulatory role of an H-NS-mediated bridge between the DRE and *ygaY* uncertain. Fragment proU3_NlaIII is also bound by IHF at -33 to +25 around P2 (Supplementary Data 1A)[69]. IHF is primarily a DNA bending protein[100] with some propensity for DNA bridging[101]. Yet, it is not expected to directly mediate the interaction between proU3_NlaIII and proU11_NlaIII since, in contrast to H-NS-mediated bridges that anchor a pair of AT-rich loci[102], the recruitment of a second DNA molecule by IHF occurs non-specifically[101].

To address these discrepancies, we studied the organisation of *proVWX* in NT644, a strain in which a series of point mutations were employed to reduce the H-NS binding affinity of the DRE (Supplementary Figs. 4 and 5). Six of the thirty-two point mutations introduced in the DRE overlapped with the IHF binding site positioned around P2. The mutations, particularly the alteration of the poly-A tract of the IHF binding site from 5'-AAAAAA-3' to 5'-AAGCAA-3' (Supplementary Fig. 4; Supplementary Data 1A), are expected to weaken IHF binding affinity[103,104]. 3C-qPCR profiles of *proVWX* in this strain at all osmolarity conditions show a sharp decrease in the relative interaction frequency between proU3_NlaIII (DRE) and proU11_NlaIII (*ygaY*) (arrow I, Fig. 6a–c; Supplementary Table 10), verifying that a NAP-mediated bridge forms between the *proVWX* DRE and *ygaY*. The bridge is a repressive regulatory structure as evidenced by the increased transcript level of *proVWX* at low osmolarity and upon a hyperosmotic shock in NT644 compared to NT331 (Fig. 2b, c and Supplementary Fig. 6b, c). Both IHF and H-NS can form DNA−NAP−DNA bridges[25,101]. But, while IHF forms a bridge between its consensus binding site and a locus that is recruited non-specifically[101], H-NS preferentially bridges AT-rich loci[102]. The 3C-qPCR profiles of *proVWX* in NT644 show that point mutations to the DRE disrupt the interaction between fragments proU3_NlaIII and proU11_NlaIII (arrow I, Fig. 6a–c), rather than causing non-specific decreases in relative interaction frequency. This observation indicates that the interaction is mediated by H-NS rather than IHF. The osmosensitivity of this interaction observed in the wild-type strain (Fig. 5d) further strengthens this conclusion.

A comparison of the *proVWX* 3C-qPCR profile of NT331 at 0.08 M NaCl where H-NS-mediated interactions of proU3_NlaIII are maintained with that of NT644 where H-NS-mediated interactions of the DRE are disrupted, shows a detectable decrease in the relative interaction frequency between the DRE (proU3_NlaIII) and the URE (proU2_NlaIII) (arrow II, Fig. 6d), and a general decrease in the relative interaction frequency between proU3_NlaIII and the *nrd* genes that lie upstream of *proVWX* (shaded area, Fig. 6d). This indicates that in addition to the DRE−H-NS−*ygaY* bridge, the repressive complex formed by H-NS involves an interaction between the URE and the DRE, and the compaction of local chromatin upstream of the *proVWX* promoter. The reduced interaction between the URE and DRE may be reinforced by the disturbances in IHF-mediated bending dynamics[105].

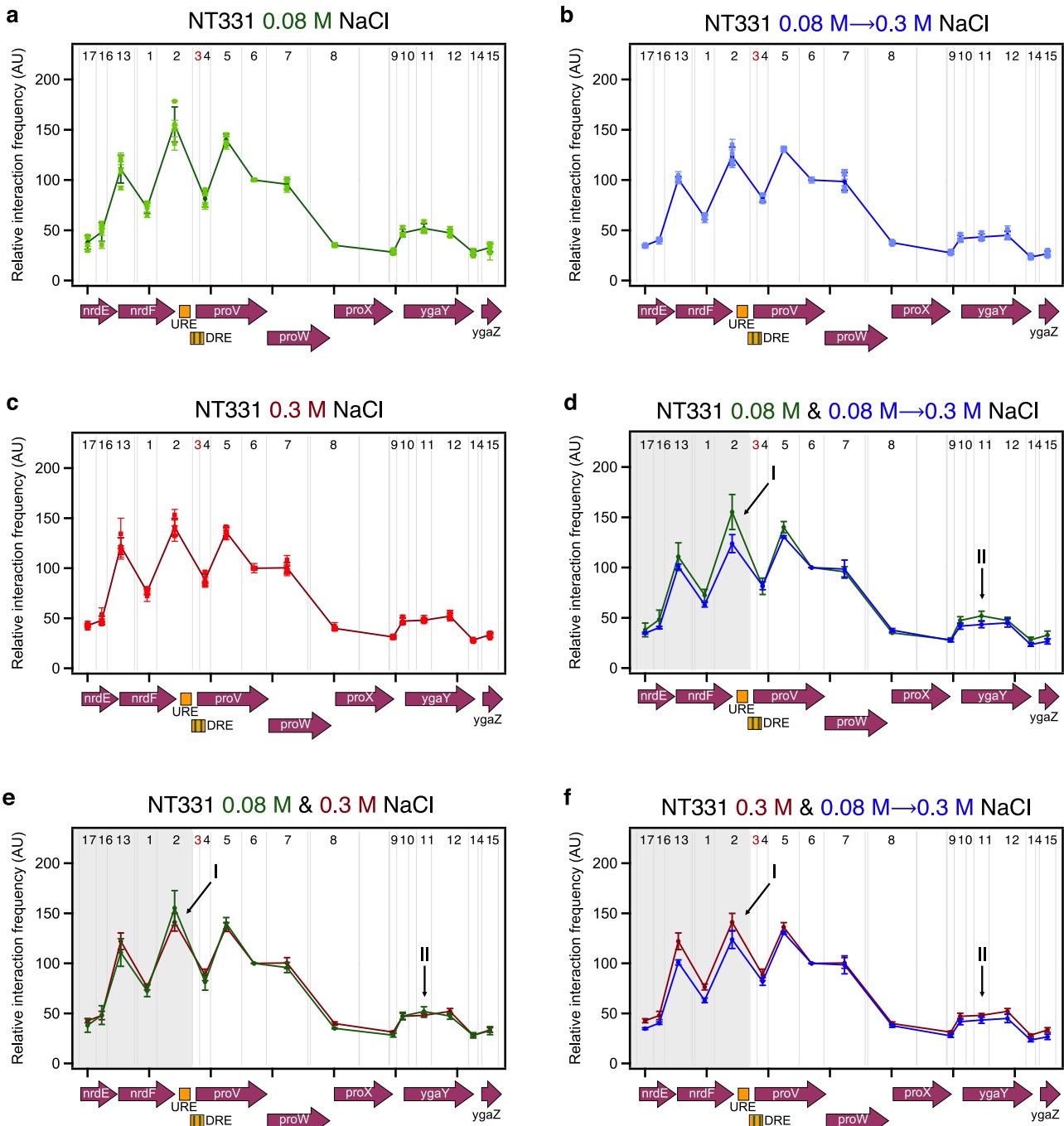

**Fig. 5 | Structural re-modelling of *proU* in response to osmolarity in a wild-type strain (NT331).** The interaction profile of fragment proU3_NlaIII across the *proVWX* operon during **a** exponential growth at 0.08 M NaCl, **b** following a hyperosmotic shock from 0.08 M NaCl to 0.3 M NaCl, and **c** during exponential growth at 0.3 M NaCl representing adaptation to hyperosmotic stress. Comparisons of the proU3_NlaIII 3C-qPCR profiles between **d** exponential growth at 0.08 M NaCl and after a hyperosmotic shock, **e** exponential growth at 0.08 M and 0.3 M NaCl, and **f** after a hyperosmotic shock and upon adaptation to hyperosmotic stress. Description of the x-axes: The x-axes show the organisation of the genetic elements in the chr:2802861-2809929 locus of *E. coli* str. K-12 substr. MG1655 (NC_000913.3 –

https://www.ncbi.nlm.nih.gov/nuccore/556503834). Restriction enzyme: NlaIII. The gray lines show the positions of NlaIII digestion sites at chr:2802861-2809929. 3C-qPCR test and anchor fragments: The numbers between the gray lines show the fragments selected for the 3C-qPCR study. Fragment 3 – proU3_NlaIII – is the anchor fragment, and is marked in red. Test fragments are labelled in black. Internal control: Interaction proU3_NlaIII-proU6_NlaIII. Data (5a–f) are presented as mean values +/- standard deviation. Dot plots (5a–c): *n* = 3 technical replicates of a biologically independent culture. Line graphs (5a–f): *n* = 4 biologically independent cultures. Source data are provided as a Source Data file.

The decompaction at the *nrd* operon is not expected to be dependent on IHF, despite the non-specificity of IHF-mediated DNA compaction by bridging[101], since the occurrence of only one IHF binding site in proU3_NlaIII cannot account for the impact observed here. The results also show that the relative interaction frequency between the

proU3_NlaIII and proU4_NlaIII fragments that constitute the two parts of the DRE (Supplementary Data 1A) is higher in NT644 than NT331 (arrow III, Fig. 6d–f). This suggests that the deficiency of H-NS and/or altered IHF bending dynamics at proU3_NlaIII favour an interaction between the high-affinity H-NS binding sites of the DRE.

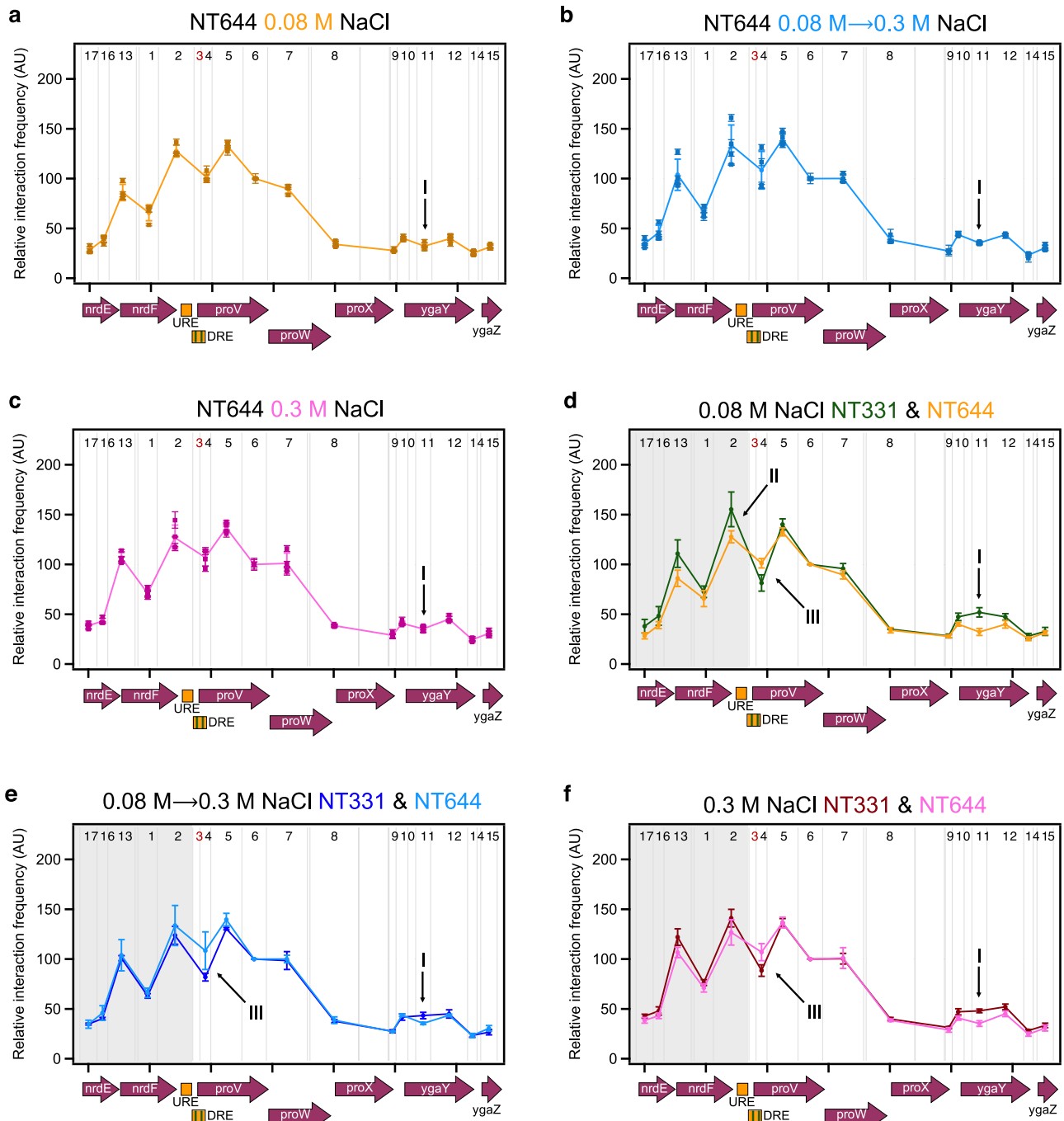

**Fig. 6 | Structural re-modelling of *proU* with an H-NS-deficient DRE in response to osmolarity.** The 3C-qPCR profiles of fragment proU3_NlaIII with reduced H-NS-binding affinity across the *proVWX* operon during **a** exponential growth at 0.08 M NaCl, **b** following a hyperosmotic shock from 0.08 M NaCl to 0.3 M NaCl, and **c** during exponential growth at 0.3 M NaCl. Comparisons of the proU3_NlaIII 3C-qPCR profiles between NT331 and NT644 **d** during exponential growth at 0.08 M NaCl, **e** following a hyperosmotic shock, and **f** during exponential growth at 0.3 M NaCl. X-axes: chr:2802861-2809929 of *E. coli* str. K-12 substr. MG1655 (NC_000913.3

– https://www.ncbi.nlm.nih.gov/nuccore/556503834). Restriction enzyme: NlaIII. Digestion sites of the enzyme within the locus of interest are marked with gray lines. 3C-qPCR anchor fragment: proU3_NlaIII (Fragment 3), labelled in red. 3C-qPCR test fragments: Fragments 1, 2, and 4–17, labelled in back. Internal control: Interaction proU3_NlaIII-proU6_NlaIII. Data (6a–f) are presented as mean values +/- standard deviation. Dot plots (6a–c): $n = 3$ technical replicates of a biologically independent culture. Line graphs (6a–f): $n = 4$ biologically independent cultures. Source data are provided as a Source Data file.

The comparable expression of *ygaY*1 and *ygaY*2 in NT331 and NT644 (Figs. 1h, 2k and Supplementary Figs. 1h, 6k; Supplementary Tables 1–4) suggests that the DRE−H-NS−*ygaY* bridge does not mediate transcriptional insulation between *proVWX* and *ygaY*. Other factors such as the presence of a terminator at the end of *proVWX*, or structural features in the vicinity of *proX* and *ygaY* that are not architecturally linked to the DRE may prevent the spill-over of transcription

into the *yga* operon. Nevertheless, the presence of StpA in H-NS nucleoprotein complexes in vivo[106], and the insulatory role of StpA at *proVWX* suggests that the DRE−H-NS−*ygaY* bridge may play a minor role in transcription termination.

Mutations to the DRE that reduce its H-NS binding affinity lock the conformation of *proVWX* in an H-NS-deficient structure, such that the 3C-qPCR profiles of the operon in strain NT644 at different osmotic

conditions are almost indistinguishable (Fig. 6a–c; Supplementary Table 10). Functionally, the locking is particularly evident in NT644 subjected to a hyperosmotic shock where a shift in osmolarity from 0.08 M NaCl to 0.3 M NaCl increases *proVWX* expression by 1.2-2.7 fold (Fig. 2f and Supplementary Fig. 6f). Such a range of activation of *proU* during hyperosmotic stress has previously been observed for expression from P2 in a Δ*hns* strain[49]. This indicates that as a result of the disruption of the repressive H-NS complex at the DRE in NT644, the increase in expression of *proVWX* in response to a hyperosmotic shock occurs primarily due to the osmosensitivity of P2. Adaptation to hyperosmotic stress represses *proVWX*. This is also observed in NT644, however, the transcript level drops to as low as 30% of that during exponential growth in low osmolarity (Fig. 2h and Supplementary Fig. 6h). The decline in expression occurs despite the operon remaining locked in its H-NS deficient structure, supporting our earlier hypothesis that in NT644 at high osmolarity, a factor other than H-NS represses *proVWX*, such as RNaseIII-mediated transcript processing and subsequent RNA degradation.

### The role of active transcription in structural re-modelling of *proVWX*

The observations discussed above provide compelling evidence that the expression of *proVWX* is regulated by its three-dimensional organisation, and that H-NS mediates the structural reorganisation in response to osmolarity. Several lines of evidence suggest that transcription, in turn, impacts chromosome structure[107–111]. Therefore, we used 3C-qPCR to study the three-dimensional organisation of *proVWX* in NT331 treated with the transcription inhibitor rifampicin at different osmotic conditions. RT-qPCR was not performed for this set of cultures since the global impact of rifampicin disrupts internal control transcripts.

At all osmolarity conditions, the 3C-qPCR profiles of the silenced *proVWX* operon in rifampicin-treated NT331 (Fig. 7a–c; Supplementary Table 11) are strikingly different from the profiles of the operon in untreated NT331 (Fig. 7d–f) and H-NS deficient NT644 (Fig. 7g–i). The silencing of *proVWX* using rifampicin is associated with an increase in the relative interaction frequency between the DRE (proU3_NlaIII) and the *nrdEF* genes (shaded area, Fig. 7d–i), a sharp decrease in the relative interaction frequency between proU3_NlaIII and proU4_NlaIII that comprise the two halves of the DRE (arrow I, Fig. 7d–i), and the maintenance of the interaction between the DRE (proU3_NlaIII) and *ygaY* (proU11_NlaIII) (arrow II, Fig. 7d–j). The structural features of *proVWX* that has been repressed by inhibiting transcription are reciprocal to the structural features of *proVWX* that has been activated by alleviating the binding of the architectural repressor H-NS (Fig. 7g–i). This signifies that local chromatin organisation is a coordinated output of the architectural properties of NAPs, and active transcription.

The 3C-qPCR profiles of *proVWX* in rifampicin-treated NT331 differ from each other only in the relative interaction frequency between proU3_NlaIII and proU2_NlaIII that denotes the interaction between the DRE and the URE (arrow III, Fig. 7j). The large error in the measurement of this interaction at low osmolarity precludes drawing reliable conclusions for this condition. Nevertheless, the *proVWX* 3C-qPCR profile in cells subjected to a hyperosmotic shock shows that the interaction between the DRE and the URE is weaker than in cells adapted to hyperosmotic stress. This is in line with predictions for an H-NS-mediated interaction at low concentrations of K⁺. The structure of the DRE−H-NS−*ygaY* bridge, however, is maintained following a rifampicin treatment regardless of the osmolarity condition (arrow II, Fig. 7j). Considering the mechanistic models of H-NS as an architectural protein[11,24,27], and the structural response of *proVWX* to osmotic stress in transcriptionally-active cells, the intracellular influx of K⁺ following a hyperosmotic shock might only weaken the DRE−H-NS−*ygaY* bridge while active transcription would be required to dismantle it. DNA−H-NS−DNA bridges are formed by a multimer of H-NS bound side-by-side

to two DNA molecules[25]. The H-NS DNA binding domains contained within the multimer exist in a dynamic equilibrium between the DNA-bound and -unbound states that can be shifted by a change in osmolarity[11]. At low [K⁺], both the DNA binding domains of an H-NS dimer – the smallest functional unit of the H-NS multimer – are available for DNA binding allowing the formation of a stable bridge[27]. At higher [K⁺], one of the two domains folds into the body of the protein and becomes unavailable[27]. Owing to the stochasticity of which DNA binding domain of a dimer dissociates from the DNA during the conformational switch, the DRE−H-NS−*ygaY* bridge may still be maintained, but weakly[11]. The weaker bridge might then be dismantled by chromatin remodelling during active transcription. The difference in the response of the DRE−H-NS−URE and DRE−H-NS−*ygaY* bridges to osmotic stress may stem from the specific binding of IHF at P2. IHF is a Y-shaped NAP consisting of an α-helical body and a pair of β-sheet arms extending outwards. Proline residues positioned at the tip of the β-arms intercalate between bases generating sharp bends in the DNA[100]. In its fully-bound state, the proline residues of IHF insert at two sites in the DNA, 9 bp apart, producing a > 160° DNA bend with flanking DNA folding onto the body of IHF[100]. At P2 such a bend can fold the URE onto the DRE, facilitating the formation of an H-NS-mediated bridge between the two elements. DNA bends formed by IHF are flexible and fluctuate between the fully-bound (bend angle 157° ± 31°) and partially-bound (bend angle 115° ± 30°) conformations[101,105]. In the partially-bound state, the DNA on one side of the IHF-mediated bend is not folded onto the body of the protein[101,105]. In vitro fluorescence lifetime measurements of the conformations sampled by IHF bound to its high affinity H′ site show that the formation of a partially bent IHF-H′ complex increases from 22% to 32% upon an increase in the concentration of KCl from 100 mM to 200 mM[105]. Thus, the intracellular influx of K⁺ upon a hyperosmotic shock may shift the equilibrium of IHF binding to favour the formation of a partially-bent IHF-P2 conformation, promoting the disassembly of the DRE−H-NS−URE bridge that is already weakened by K⁺.

The model above suggests that IHF favours repression of *proVWX* at low osmolarity. This contradicts an earlier report that shows the expression of *proV*-GFP cloned on a plasmid downstream of the P2 promoter is activated by four-fold in the presence of IHF at an osmolarity of 0.17 M NaCl in LB due to the distortion to the structure of the promoter upon IHF binding[69]. We propose that the impact of IHF on the expression of *proVWX* may be a consolidation of two opposing effects: repression mediated by facilitating the formation of the DRE−H-NS−URE bridge, and activation of P2 mediated by direct binding of IHF to the promoter, facilitating the formation of the open promoter complex.

### *ProVWX* – a model for the interplay between three-dimensional chromosome organisation and gene expression

The results presented above sketch a model of the structural regulation of *proVWX* in *Escherichia coli* (Fig. 8). H-NS dimers nucleate at high-affinity H-NS binding sites and multimerise via the N-terminal dimer-dimer interaction domains to form an H-NS−DNA filament. H-NS−DNA filaments can, in response to changes in osmolarity, temperature, and pH, reversibly recruit an additional DNA molecule to form DNA−H-NS−DNA bridges. Conditions of low osmolarity favour an open conformational state of H-NS in which both the DNA binding domains of an H-NS dimer are available for interacting with DNA. Under these conditions, an H-NS multimer preferentially bridges DNA loci. At *proVWX*, H-NS bridges the DRE to the URE, and to *ygaY*. The structures interfere with transcription initiation by occluding RNAP from the promoter and preventing promoter escape for promoter-bound RNAP. The bridges also behave as supercoil diffusion barriers and trap positive supercoils generated by elongating RNAP on the downstream template, hence, stimulating RNAP pausing and repressing *proVWX*.

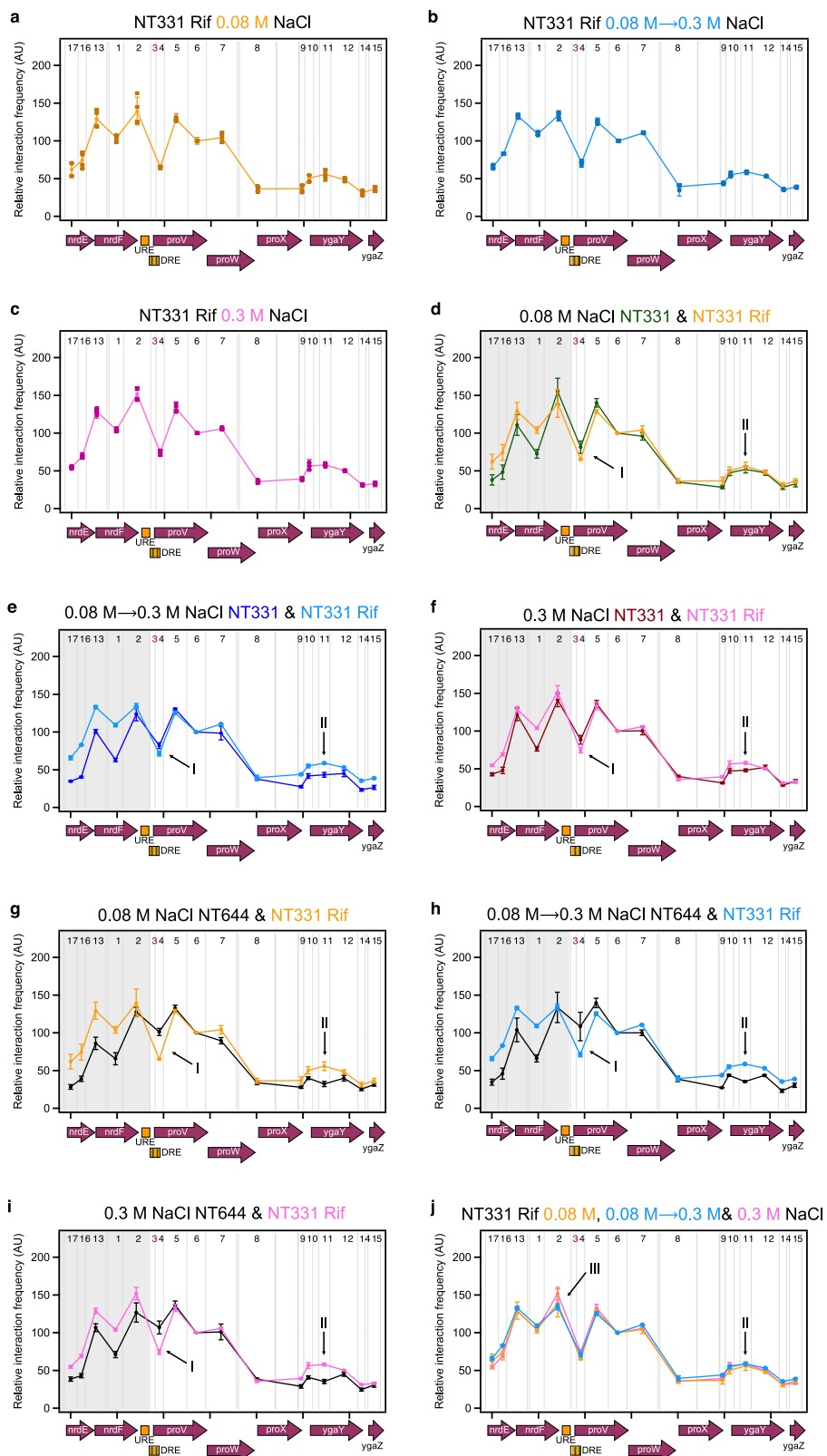

Upon a hyperosmotic shock, *E. coli* cells import K+ to rapidly reduce the osmotic potential of the cytoplasm and prevent the loss of water and dehydration. High intracellular [K+] is toxic and interferes with protein stability and function, and is, therefore, only a short-term response. The influx of K+ directly activates the osmoresponsive P2 promoter of the *proVWX* operon that encodes a transport system for inert, non-toxic osmoprotectants. The influx of K+ also shifts the

equilibrium of the conformational states assumed by H-NS from the open to the half-sequestered state where one DNA binding domain per dimer folds onto the body of the protein and becomes unavailable for interacting with DNA. Due to the stochasticity of which DNA binding domain per dimer is sequestered, DNA−H-NS−DNA bridges may either be disrupted or weakened. At *proVWX*, the DRE−H-NS−URE bridge is disrupted while the bridge between the DRE and *ygaY* weakens. The

**Fig. 7 | Structural re-modelling of *proU* in the absence of active transcription.** The 3C-qPCR profiles of fragment proU3_NlaIII across the *proVWX* operon in rifampicin-treated cells during **a** exponential growth at 0.08 M NaCl, **b** following a hyperosmotic shock from 0.08 M NaCl to 0.3 M NaCl, and **c** during exponential growth at 0.3 M NaCl. Comparisons of the proU3_NlaIII 3C-qPCR profiles between NT331 and rifampicin-treated NT331 **d** during exponential growth at 0.08 M NaCl, **e** following a hyperosmotic shock, and **f** during exponential growth at 0.3 M NaCl. Comparisons of the proU3_NlaIII 3C-qPCR profiles between NT644 and rifampicin-treated NT331 **g** during exponential growth at 0.08 M NaCl, **h** following a hyper-osmotic shock, and **i** during exponential growth at 0.3 M NaCl. **j** An overlay of the proU3_NlaIII 3C-qPCR profiles in cells subjected to a hyperosmotic shock, and during growth at 0.3 M NaCl. X-axes: chr:2802861-2809929 of *E. coli* str. K-12 substr. MG1655 (NC_000913.3 – https://www.ncbi.nlm.nih.gov/nuccore/556503834). Restriction enzyme: NlaIII. Digestion sites of the enzyme within the locus of interest are marked with gray lines. 3C-qPCR anchor fragment: proU3_NlaIII (Fragment 3), labelled in red. 3C-qPCR test fragments: Fragments 1, 2, and 4–17, labelled in back. Internal control: Interaction proU3_NlaIII-proU6_NlaIII. Data (7**a**–**j**) are presented as mean values +/- standard deviation. Dot plots (7**a**–**c**): n = 3 technical replicates of a biologically independent culture. Line graphs (7**a**–**j**): n = 4 biologically independent cultures. Source data are provided as a Source Data file.

difference in the structural response of the two bridges may stem from other architectural elements in their vicinity. The remodelling of the H-NS−DNA nucleoprotein around P2 activates *proVWX* by exposing its inherently osmoresponsive promoter(s) for RNAP binding and by exerting a weaker roadblocking effect. Active transcription from P2 then remodels the local chromatin structure of the *proVWX* template causing the disassembly of the weakened bridge between the DRE and *ygaY*. The expression of *proVWX* and the subsequent synthesis of the ProU transporter allows *E. coli* to import osmoprotectants such as glycine-betaine, proline-betaine, and proline into the cell[33,37,38,44,45]. These maintain a low osmotic potential in the cytoplasm without adverse effects on cellular physiology. Hence, $K^+$ is exported from the cell. The decrease in cytoplasmic $K^+$ stabilises the open conformation of H-NS favouring the formation of bridges and re-establishing a repressed operon.

Notably, the full range of *proVWX* osmotic regulation has not been sampled in our report, for instance, there is at least an order of magnitude of repression of *proVWX* between 0.08 M NaCl and 0.01 M NaCl[34,37,49,54]. Other regulatory mechanisms may function in these ranges, adding to the complexity of *proVWX* osmoregulation.

### The interplay between chromatin organisation and gene expression may be a general feature of transcriptional control in prokaryotes

Bacterial chromosomes are organised by active transcription and by architectural nucleoid-associated proteins (NAPs) that also function as transcription factors. NAPs are sensitive to physicochemical signals. Hence, the proteins organise the chromosome as a dynamic structure that is remodelled in response to physicochemical environmental stimuli, and they coordinate gene transcription with the extracellular environment. These characteristics have been studied separately in the past and direct evidence of the environmentally-regulated interplay between local three-dimensional chromatin organisation and gene expression in prokaryotes has been lacking. Using ensemble RT-qPCR and 3C-qPCR approaches, and the H-NS-regulated osmoresponsive *proVWX* operon of *Escherichia coli* as a model system, we provide the evidence that, in vivo, the nucleoid-associated protein H-NS regulates gene expression in response to environmental stimuli by local chromatin remodelling. The tuning of the architectural properties of H-NS by changes in osmolarity either directly remodels the chromosome to regulate gene expression, or renders the chromosome susceptible to remodelling by, for instance, the transcriptional machinery.

H-NS is a global gene regulator in, among others, *Escherichia* sp., *Shigella* sp., and *Salmonella* sp. The model of transcription regulation by chromatin reorganisation presented here provides clues of how H-NS may operate at other genes and operons within its regulon that encompasses pathogenicity-related loci. The model may also be extrapolated to explain how H-NS-like proteins such as MvaT in *Pseudomonas sp.*, and Lsr2 in *Mycobacterium sp.*, may function in response to physicochemical environmental signals that can be 'sensed' by the protein[112], and, how the regulons of atypical H-NS-like proteins such as Rok in *Bacillus sp.* may be regulated by environment-sensing protein partners like sRok[113].

The *proU* operon encodes an ABC transporter – a member of the largest group of paralogous protein complexes[114]. The components of ABC transporters are contained within operons, the earliest of which is predicted to have arisen before the divergence of bacteria and archaea[115,116]. The organisation of ABC operons is evolutionarily conserved and is specific for each orthologous group[117,118]. This ubiquity permits a systematic study of the evolution of the environmentally-regulated interplay between local chromatin structure and gene expression.

## Methods

### Strains and plasmids

Bacterial strains used in this study were derived from *Escherichia coli* MG1655 using a two-step genome editing method based on the λ-red recombinase system[119] as described in[120]. Chemically competent bacteria were transformed with pKD46[119], a temperature-sensitive plasmid encoding the λ-red proteins under an arabinose-inducible promoter. Electrocompetent pKD46+ cells were prepared from a culture grown at 30 °C in LB medium (1.0% bactotryptone (BD), 0.5% yeast extract (Alfa Aesar), 170 mM NaCl, pH 7.5) supplemented with 10 mM arabinose, for plasmid maintenance and to induce the expression of the λ-red proteins, respectively. The cells were transformed with a *kanR-ccdB* cassette[120] encoding kanamycin resistance and the ccdB toxin under a rhamnose-inducible promoter[119]. The cassette was designed to carry ~1.5 kb extensions on either side homologous to the flanks of the genomic region to be edited. *KanR-ccdB* was amplified from plasmid pKD45[119] and the homology regions from genomic DNA. The complete construct was assembled using overlap extension PCR (oePCR). Recombinants were selected for on LB agar (1.0% bactotryptone (BD), 0.5% yeast extract (Alfa Aesar), 170 mM NaCl, 1.5% bacteriological agar (Oxoid), pH 7.5) supplemented with 40 μg/mL kanamycin at 37 °C and verified using colony PCR and Sanger sequencing (BaseClear B.V.). The mutant allele construct with ~1.5 kb homology regions was assembled into a plasmid using Gibson assembly[121]. The allele was amplified and used to replace the genomic *kanR-ccdB* cassette via the λ-red recombinase system. Recombinants were selected for on M9 agar plates (42 mM $Na_2HPO_4$, 22 mM $KH_2PO_4$, 19 mM $NH_4Cl$, 8.5 mM NaCl, 2 mM $MgSO_4$, 0.1 mM $CaCl_2$, 1.5% bacteriological agar (Oxoid)) supplemented with 1% rhamnose and verified with colony PCR and Sanger sequencing (BaseClear B.V.). All bacterial strains and intermediates were stored at -80 °C as glycerol stocks. Strains and plasmids used in this study are listed in Tables 1 and 2.

### Media and growth conditions

For low salt and hyper-osmotic shock conditions, a single bacterial colony from a freshly streaked plate was grown overnight at 37 °C in low salt LB medium (LS-LB: 1.0% bactotryptone (BD), 0.5% yeast extract (Alfa Aesar), 80 mM NaCl, pH 7.5) or low salt M9 medium (LS-M9: 42 mM $Na_2HPO_4$, 22 mM $KH_2PO_4$, 19 mM $NH_4Cl$, 2.0 mM $MgSO_4$, 0.1 mM $CaCl_2$, 80 mM NaCl, 1X trace elements, 1% Bacto™ casamino acids (BD), 10 μg/mL thiamine (Sigma-Aldrich), 0.4% glycerol (PanReac Applichem)). The overnight culture was used to inoculate fresh low salt LB or M9 medium to a starting $OD_{600}$ of 0.05 for up to four

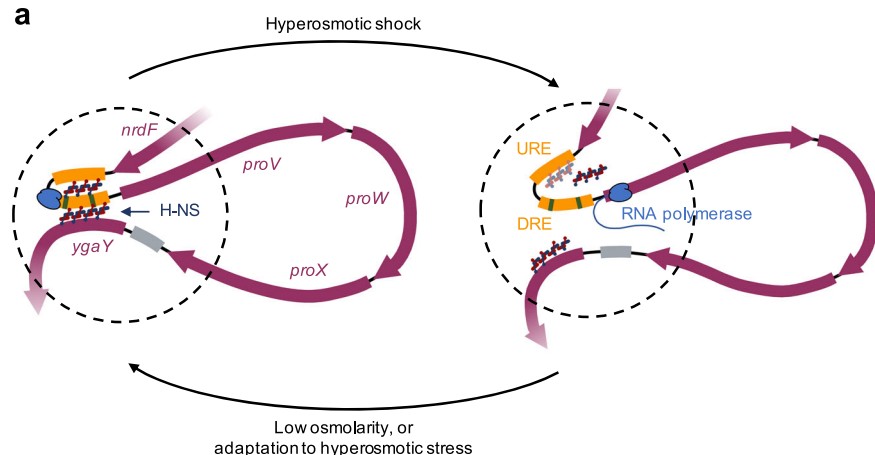

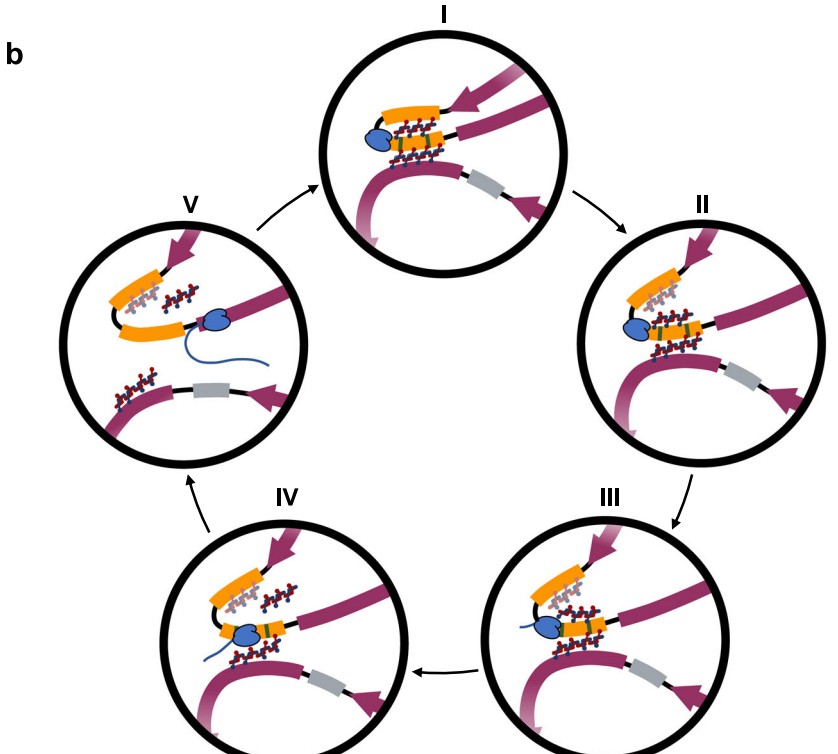

**Fig. 8 | The environmentally-regulated interplay between local three-dimensional chromatin organisation and transcription of the osmoregulated *proVWX* operon in *Escherichia coli*. a** H-NS bridges the downstream regulatory element (DRE) to the upstream regulatory element (URE), and to *ygaY*. The structures interfere with transcription[24] and repress *proVWX*. Upon a hyperosmotic shock, cells rapidly import K⁺ to reduce the osmotic potential of the cytoplasm to prevent dehydration[42]. However, a high intracellular concentration of K⁺ is cytotoxic. It causes the precipitation of proteins and interferes with enzyme activity[36]. The import of K⁺ also disrupts H-NS-mediated bridges[27], and consequently, activates the expression[24] of *proVWX*. ProVWX encodes ProU − an ABC transporter that embeds in the inner membrane of *E. coli* and actively imports osmoprotectants such as glycine- and proline-betaine[43]. These molecules reduce the osmotic potential of the cytoplasm without destabilising cellular proteins[33,37,38,44,45]. The associated decrease of intracellular K⁺[36,46] re-establishes H-NS-mediated bridges[27] that repress[24] *proVWX*. **b** A mechanistic model of the activation of *proVWX*

expression. (I) At low osmolarity, *proVWX* is repressed by DRE−H-NS−URE and DRE−H-NS−*ygaY* bridges. RNA polymerase is poised at the promoter in the open complex[49]. (II) The influx of K⁺ following a hyperosmotic shock[42] destabilises helix α3 in H-NS promoting the folding of one DNA binding domain of an H-NS dimer onto the body of the protein[27]. In the figure, this is represented as the circular DNA binding domains of H-NS interacting with the oval-shaped H-NS body. Sequestration of the DNA binding domains results in the opening or weakening of H-NS-mediated DNA bridges due to the stochasticity with which H-NS DNA binding domains are sequestered, and the influence of structural elements in the vicinity of the bridge. The DRE−H-NS−URE bridge opens. The H-NS multimer that formed the bridge may either remain bound to the high-affinity H-NS binding sites in the DRE or be cleared away by the URE. The DRE−H-NS−*ygaY* bridge is weakened. (III and IV) Promoter clearance and transcription elongation re-structures and dismantles the weakened DRE−H-NS−*ygaY* bridge. (V) RNA polymerase transcribes *proU*.

biological replicates and grown at 37 °C to an $OD_{600}$ of ~1.0. The culture was split into two aliquots. For hyper-osmotic shock, 5.0 M NaCl was added to one of the aliquots at a ratio of 46 µL per mL of culture. An equivalent volume of milliQ water was added to the second aliquot for the low salt condition. The cultures were grown for 10 minutes at 37 °C, and then immediately harvested for RNA isolation, and cell fixation for 3C-based experiments. For high salt studies, a single bacterial colony was grown overnight in high salt LB medium (LS-LB: 1.0% bactotryptone (BD), 0.5% yeast extract (Alfa Aesar), 300 mM NaCl, pH 7.5) or high salt M9 medium (HS-M9: 42 mM $Na_2HPO_4$, 22 mM $KH_2PO_4$, 19 mM $NH_4Cl$, 2.0 mM $MgSO_4$, 0.1 mM $CaCl_2$, 300 mM NaCl, 1X trace elements, 1% Bacto™ casamino acids (BD), 10 µg/mL thiamine (Sigma-Aldrich), 0.4% glycerol (Pan-Reac Applichem)). The overnight culture was used to inoculate fresh high salt LB or M9 medium to an $OD_{600}$ of 0.05 for up to four biological replicates. The cells were cultured at 37 °C to an $OD_{600}$ of ~1.0, and for an additional 10 minutes. The cells were immediately harvested for RNA isolation, and cell fixation. For experiments requiring rifampicin treatment, cultures at an $OD_{600}$ of ~1.0 in were incubated with 100 µg/mL rifampicin for two hours at 37 °C. The addition of NaCl for a hyperosmotic shock and the respective treatments for low and high osmolarity conditions were performed after incubation with rifampicin.

Trace elements were prepared as a 100X stock solution of the following composition per 100 mL: 0.1 g $FeSO_4 \cdot 7H_2O$, 0.6 g $CaCl_2 \cdot 2H_2O$, 0.12 g $MnCl_2 \cdot 4H_2O$, 0.08 g $CoCl_2 \cdot 6H_2O$, 0.07 g $ZnSO_4 \cdot 7H_2O$, 0.03 g $CuCl_2 \cdot 2H_2O$, 2 mg $H_3BO_3$, and 0.5 g $EDTA \cdot Na_2$.

### RNA isolation and handling
Bacterial cells in 1 mL of culture were collected by centrifugation at 13,000 x $g$ for 2 minutes. The supernatant was removed and the pellet was resuspended in 200 µL of Max Bacterial Enhancement Reagent (TRIzol® Max™ Bacterial RNA Isolation Kit, Catalogue number: 16096020, ThermoFisher Scientific) pre-heated to 95 °C. The lysate was incubated at 95 °C for 5 minutes. The preparation was treated with 1 mL of TRIzol® reagent (TRIzol® Max™ Bacterial RNA Isolation Kit, Catalogue number: 16096020, ThermoFisher Scientific) and incubated at room temperature for 5 minutes. RNA isolation was paused at this step to accommodate cell fixation for 3C and Hi-C library preparation by flash-freezing the TRIzol®-lysate mix in liquid nitrogen and storing at -80 °C for up to 2 weeks. The TRIzol®-lysate mix was thawed at room temperature and RNA isolation was continued using the TRIzol® Max™ Bacterial RNA Isolation Kit (Catalogue number: 16096020, Thermo-Fisher Scientific) according to the manufacturer's instructions. The concentration and purity ($A_{260}/A_{280}$) of RNA was measured with a NanoDrop™ 2000 spectrophotometer (Catalogue number: ND-2000, Thermo Scientific™) and NanoDrop 2000/2000c Software Version 1.6. Accordingly, RNA samples were diluted to a final concentration of 20 ng/µL with UltraPure™ DNase/RNase-Free Distilled Water (Catalogue number: 10977035, ThermoFisher Scientific). DNA contamination in the RNA samples was checked with RNase A (Catalogue number: 19101, Qiagen) treatment and agarose gel electrophoresis. 50 µL of the RNA samples were transferred into wells of a 96-well RNase/DNase-free plate in triplicate to facilitate multi-channel pipetting for RT-qPCR experiments. RNA samples in the multi-well plate were stored at 4 °C and placed on ice during reaction set-up to avoid freeze-thaw cycles. Stock RNA samples were stored at -20 °C.

In compliance with MIQE guidelines[122], the details of RNA yield and purity ($A_{260}/A_{280}$ measurements), and the results of RNA inhibition testing are provided in Supplementary Data 1B. Additionally, the result of a DNA contamination assessment of the RNA preparations showing no detectable contamination is provided in Supplementary Fig. 12.

### RT-qPCR
**Primer design.** Primers for RT-qPCR experiments were designed using the *Escherichia coli* K-12 MG1655 sequence (Accession number: NC_000913.3 – https://www.ncbi.nlm.nih.gov/nuccore/556503834). In silico specificity, screening was performed on SnapGene® Viewer 5.2. RT-qPCR primers were ordered as dried pellets from Sigma-Aldrich. The oligonucleotides were dissolved in 1X TE pH 8.0 to a final concentration of 100 µM and stored at 4 °C. The list of primers used for the RT-qPCR assay and corresponding amplicon details are provided in Supplementary Data 1B. Primer annealing sites and amplicon positions in the *proVWX* operon are provided in Supplementary Data 1A and 1B. The specificity of primer pairs was experimentally determined with Sanger sequencing (BaseClear B.V.) of the amplified PCR product (Supplementary Data 1B and 2) and with melting curves (Supplementary Fig. 13).

**Reaction set-up.** RT-qPCR experiments were performed with the Luna® Universal One-Step RT-qPCR Kit (Catalogue number: E3005E, New England Biolabs) with a final reaction volume of 10 µL per well. The reactions were prepared according to the manufacturer's instructions with modifications. Reaction composition is provided in Supplementary Table 12. The reactions were set-up manually in Hard-Shell® 96-well PCR plates (Catalogue number HSP9635, Bio-Rad) and run on a C1000 Touch™ thermal cycler heating block (Catalogue number: 1841100, Bio-Rad) with a CFX96™ Real-time system (Catalogue number: 1845097, Bio-Rad). The thermocycling parameters were set up according to the manufacturer's instructions and are provided in Supplementary Table 13. NT331 genomic DNA that was serially diluted in triplicate to 10X, 100X, 1000X, and 10000X dilution factors was used for the standard samples. Standard samples, test samples, and No Template Controls were processed in the same manner.

**Choice of internal control.** Three internal controls, *rpoD*, *hcaT*, and *rrsA*, were selected for this study. *rpoD* encoding the RNA polymerase $\sigma^{70}$ factor, is a house-keeping gene that has been validated as a stably-expressed mRNA suitable as an internal control in a diversity of bacterial species including *Klebsiella pneumoniae*[123], *Pseudomonas aeruginosa*[124], *Pseudomonas brassicacearum* GS20[125], and *Gluconacetobacter diazotrophicus*[126]. *hcaT* encodes a predicted 3-phenylpropionic transporter and has been validated as a

### Table 1 | List of strains

| Strain ID | Description | Reference |
|---|---|---|
| MG1655 | *Escherichia coli* K-12 | |
| NT331 | MG1655 Δ*endA* (Supplementary Figs. 10 and 11) | This study |
| NT606 | MG1655 Δ*endA*; pKD46 (AmpR) | This study |
| NT632 | MG1655 Δ*endA* rnc::kanRccdB | This study |
| NT633 | MG1655 Δ*endA* stpA::kanRccdB | This study |
| NT642 | MG1655 Δ*endA* pProU::kanRccdB; pKD46 (AmpR) | This study |
| NT644 | MG1655 Δ*endA* pProU::pProU_GC | This study |

### Table 2 | List of plasmids

| Plasmid ID/ Strain ID | Backbone | Insert | Resistance | Storage strain | Reference |
|---|---|---|---|---|---|
| pKD45/ XT198 | pKD45 | kanR $P_{rhaB}$-ccdB | Kanamycin | HCB1666 | 119 |
| pKD46/ XT146 | pKD46 | araC $P_{araB}$-λ Red | Ampicillin | MG1655 Δ*thyA* | 119 |

suitable internal control in *Escherichia coli*[127,128]. *rrsA* codes for 16 S rRNA and occurs in six copies in the *E. coli* MG1655 chromosome. It is accepted as a stably expressed house-keeping gene and was accepted as a validated internal control for *E. coli* RT-qPCR experiments[128]. *rrsA* has been used to normalise relative transcript levels in RT-qPCR studies of the *proU* operon[54]. We tested the amplicon as a potential internal control, but, the low Cycle of quantification (Cq) values of *rrsA* in comparison to the Cq values of amplicons of the *proU* operon (Supplementary Data 1C, 1D, 3 and 4) dissuaded the classification of *rrsA* as a reliable internal control. The ~2-fold higher expression of *rpoD* to *hcaT* (Supplementary Data 1C, 1D, 3 and 4) motivated the use of only one internal control at a time for normalisation since normalisation with a geometric average of the two amplicons biases the normalisation towards the higher expressed amplicon. The RT-qPCR results described in the Results and Discussion section have been normalised using *rpoD* as the internal control. Equivalent data normalised with *hcaT* expression have been provided in the supplementary files for comparison.

## Cell fixation

*Escherichia coli* cells were fixed as reported earlier[99] with modifications (Supplementary Figs. 14 and 15). 6.0 mL of the bacterial culture in M9 medium was collected by centrifugation at 3000 x *g* for 5 minutes at 4 °C. The supernatant was discarded, and the pellet was resuspended in 2 mL of LS-M9 medium for the low salt cultures or 2 mL of HS-M9 medium for the high salt and salt shock cultures. The cell suspension was treated with 8 mL of ice-cold methanol (Catalogue number: 676780, Sigma-Aldrich) and incubated at 4 °C for 10 minutes. The cells were collected by centrifugation at 3000 x *g* for 5 minutes at 4 °C and washed with 20 mL of LS/HS-M9 medium. The pellet of washed, methanol-treated cells was resuspended in 3% formaldehyde (Catalogue number: F8775, Sigma-Aldrich) in LS/HS-M9 medium. The reaction was incubated at 4 °C for 1 hour and subsequently quenched with a final concentration of 0.375 M glycine for 15 minutes at 4 °C. The fixed cells were collected by centrifugation and washed twice with 1X TE pH 8.0 (Catalogue number: 574793, Sigma-Aldrich). The cell suspension was split into three aliquots and pelleted at 10000 x *g* at room temperature for 2 minutes. The supernatant was removed, and the cell pellet was flash-frozen in liquid nitrogen for storage at -80 °C until use (up to 1 month).

Bacterial cultures grown in LB were fixed in the same manner, except that all wash steps and 3% formaldehyde preparation were performed with 1X LS-PBS (2.7 mM KCl, 10 mM $Na_2HPO_4$, 1.8 mM $KH_2PO_4$), 1X HS-PBS (280 mM NaCl, 2.7 mM KCl, 10 mM mM $Na_2HPO_4$, 1.8 mM $KH_2PO_4$), or 1X PBS (137 mM NaCl, 2.7 mM KCl, 10 mM mM $Na_2HPO_4$, 1.8 mM $KH_2PO_4$) as required.

## Hi-C and 3C library preparation

Hi-C and 3C libraries were prepared as described previously[99] with modifications. Briefly, a pellet of fixed *E. coli* cells was resuspended in 50.0 µL of 1X TE pH 8.0 (Catalogue number: 574793, Sigma-Aldrich). The cells were lysed and solubilised with 2000 U of Ready-Lyse Lysozyme (Catalogue number: R1802M, Epicentre) and 0.5% SDS (Catalogue number: L3771, Sigma-Aldrich). The extracted chromatin was digested for 3 hours at 37 °C in a volume of 250.0 µL with 100 U of NlaIII (Catalogue number: R0125L, NEB) for 3C library preparation, or 100 U of PsuI (Catalogue number: ER1551, ThermoFisher Scientific) for preparing a Hi-C library. The reaction was performed in the presence of 0.5% Triton X-100 (Catalogue number: T8787, Sigma-Aldrich) to sequester SDS in the cell lysate. 3C libraries were treated with 0.5% SDS (Catalogue number: L3771, Sigma-Aldrich) for 20 minutes at 37 °C to terminate restriction digestion. For Hi-C libraries, the overhangs in the restriction digested chromatin were filled in with biotin-14-dATP (Catalogue number: 19524016, ThermoFisher Scientific) using 30 U of

DNA Polymerase I, Large (Klenow) fragment (Catalogue number: M0210M, NEB) in a volume of 300.0 µL. The reaction was performed for 45 minutes at 25 °C, and later terminated with 0.5% SDS (Catalogue number: L3771, Sigma-Aldrich) for 20 minutes at 25 °C.

The chromatin for both 3C and Hi-C libraries was centrifuged at 25000 x *g* for 1 hour at 4 °C. The supernatant was discarded, and the gel-like pellet was resuspended in 200.0 µL of nuclease-free water (Catalogue number: AM9932, ThermoFisher Scientific). The DNA concentration of the preparation was measured with the Qubit® dsDNA HS Assay Kit (Catalogue number: Q32854, ThermoFisher Scientific). 1.0-3.0 µg of DNA was used for a ligation reaction in a volume of 1 mL using 1.0 µL of 2000 U/µL T4 DNA Ligase (Catalogue number: M0202M, NEB) for 3C experiments, and 2.0 µL of the enzyme for Hi-C experiments. The reaction was incubated at 16 °C for 16 hours, and then at 25 °C for 1 hour. The reaction was terminated with 20.5 µL of 0.5 M EDTA (Catalogue number: 15575020, ThermoFisher Scientific).

The libraries were purified with a 30-minute incubation at 37 °C in the presence of 160 µg/mL RNase A (Catalogue number: 19107, Qiagen), a treatment with 200 µg/mL Proteinase K (Catalogue number: 19157, Qiagen) for 16 hours at 65 °C in the presence of 500 mM NaCl, two extractions with 25:24:1 phenol:chloroform:isoamyl alcohol (Catalogue number: P3803, Sigma-Aldrich), and one extraction with chloroform (Catalogue number: 319988, Sigma-Aldrich). The DNA was precipitated at -20 °C overnight with 0.025 V of 5 mg/mL glycogen (Catalogue number: AM9510, ThermoFisher Scientific), 0.1 V of 1.0 M NaOAc (Catalogue number: S2889, Sigma-Aldrich) pH 8.0, and 2.5 V of 100% ethanol. Precipitated DNA was pelleted by centrifugation, washed twice with 70% ethanol, and air-dried. The pellet was dissolved in 30.0 µL nuclease-free water (Catalogue number: AM9932, ThermoFisher Scientific). At this step, 3C libraries were stored at -20 °C until use.

Hi-C libraries were treated with 3 U of T4 DNA polymerase (Catalogue number: M0203L, NEB) in the presence of 0.1 mM dGTP (Catalogue number: N0442S, NEB) for 3 hours at 16 °C to remove biotin from unligated restriction fragment ends. The library was then extracted once with 25:24:1 phenol:chloroform:isoamyl alcohol (Catalogue number: P3803, Sigma-Aldrich), and once with chloroform (Catalogue number: 319988, Sigma-Aldrich). The DNA was precipitated overnight at -20 °C with 0.025 V of 5 mg/mL glycogen (Catalogue number: AM9510, ThermoFisher Scientific), 0.1 V of 3.0 M NaOAc (Catalogue number: S2889, Sigma-Aldrich) pH 5.6, and 2.5 V of 100% ethanol. Precipitated DNA was collected by centrifugation. The DNA pellet was washed once with 70% ethanol, air-dried, and dissolved in 20.0 µL nuclease-free water (Catalogue number: AM9932, ThermoFisher Scientific). Hi-C next-generation sequencing (NGS) libraries were prepared using the KAPA HyperPlus Kit (Catalogue number: KK8512, KAPA Biosystems) according to instructions[129].

## 3C library handling

The concentration of the 3C libraries was measured with the Qubit™ dsDNA HS Assay Kit (Catalogue number: Q32854, ThermoFisher Scientific) using the Qubit 2.0 fluorometer (Catalogue number: Q32866, ThermoFisher Scientific). The $A_{260}/A_{280}$ ratio was not determined since the presence of glycogen in the library preparation interfered with absorbance at these wavelengths. The quality of the 3C libraries, and effectively, their reliability for 3C-qPCR, was determined with an inhibition testing experiment akin to the Purity Assessment[98]. Details of the yield of 3C libraries and inhibition testing experiments are provided in Supplementary Data 1E, and in the file labelled Purity Assessment in Supplementary Data 5 and 6.

The 3C libraries were diluted to a final concentration of 0.2 ng/µL with 10 mM Tris, pH 8.0. The libraries were individually pipetted into a 96-well plate for 3C-qPCR. 3C libraries were stored at 4 °C and placed at room temperature during reaction set-up to avoid freeze-thaw cycles. Library stocks were stored at -20 °C.

 

## 3C-qPCR[98,99]

**Primer and TaqMan probe (hydrolysis probe) design.** Primers and TaqMan probes for 3C-qPCR experiments of the *proVWX* operon were designed using the *Escherichia coli* K-12 MG1655 sequence (Accession number: NC_000913.3 – https://www.ncbi.nlm.nih.gov/nuccore/556503834) on SnapGene® Viewer 5.2. The program was also used for in silico specificity screening. For the fragments of interest for 3C-qPCR test reactions and cross-linking controls, primers and TaqMan probes were designed on opposite DNA strands within 100 bp of the downstream NlaIII site, with TaqMan probes positioned closer to the restriction site than the primers. All primers were designed to anneal to the genomic DNA with the same directionality to ensure that the signal observed during 3C-qPCR assays arises only from re-ligation between the pair of fragments being tested and not due to incomplete restriction digestion. Furthermore, the hybridisation of the TaqMan probe to the non-primed strand of a fragment ensures that the hydrolysis of the probe, and hence, the detection of a fluorescence signal in qPCR only occurs if amplification crosses the ligation junction being tested.

HPLC-purified, double-quenched TaqMan probes with a 5′ 6-FAM fluorophore, 3′ Iowa Black™ Fluorescence Quencher, and an internal ZEN quencher positioned 9 bases from the fluorophore were ordered as dried pellets from Integrated DNA Technologies, Inc. Primers (desalted, dry) were ordered from Integrated DNA Technologies, Inc. or Sigma-Aldrich. The oligonucleotides were dissolved in 1X TE pH 8.0 (Catalogue number: 574793, Sigma-Aldrich) to a final concentration of 100 µM and stored at 4 °C. The lists of primers and probes used for 3C-qPCR, their positions within the *proU* operon and their annealing sites on the chromosome are provided in Supplementary Data 1A and 1F.

**Control library preparation.** Three sets of control libraries were prepared for 3C-qPCR: digested and randomly re-ligated NT331 genomic DNA, digested and randomly re-ligated NT644 gDNA, and a synthetic control template where chimeric fragments of interest were separately prepared by PCR and pooled in equimolar ratios. The control libraries were serially diluted to prepare the standard samples for 3C-qPCR.

**Reaction set-up.** 3C-qPCR[98,99] was performed using the PrimeTime™ Gene Expression Master Mix Kit (Catalogue number: 10557710, Integrated DNA Technologies, Inc.) using 1 ng of the 3C libraries, corresponding to ~2 × 10^5 genome equivalents, as the template. A detailed reaction composition is provided in Supplementary Table 14. Experiments were set-up manually in Hard-Shell® 96-well PCR plates (Catalogue number HSP9635, Bio-Rad) and run on a C1000 Touch™ thermal cycler heating block (Catalogue number: 1841100, Bio-Rad) with a CFX96™ Real-time system (Catalogue number: 1845097, Bio-Rad) using the program outlined in Supplementary Table 15. Standard, Test, and No Template Control samples were processed in the same manner.

## qPCR data analysis

RT- and 3C-qPCR data was processed using the Bio-Rad CFX Manager™ Version 3.1 program (Bio-Rad) and analysed with Microsoft Excel (Microsoft 365). All the raw data files, and the exported.csv files are available as Supplementary material (Supplementary Data 3 and 4 for RT-qPCR, and Supplementary Data 5, 6 and 7 for 3C-qPCR). For all RT-qPCR experiments, the cycle of quantification (Cq) was determined using the Single Threshold option available on Bio-Rad CFX Manager™ Version 3.1. For 3C-qPCR, Cq was determined using the Regression function. Reactions with unreliable Cq values due to pipetting errors owing to the manual set-up of the experiment, or evaporation from improperly sealed wells, were eliminated from analysis. For the intercalator-based RT-qPCR experiment, melt curves were used to check the specificity of amplification, and hence, the validity of the experiment. Eliminated reactions and the justifications for each are provided in Supplementary Data 1C and 1D for RT-qPCR experiments, and Supplementary Data 1G and 1H for 3C-qPCR.

The Cq values of the standard samples were used to plot a standard curve of Cq against the logarithm of the starting quantity of the template. Standard samples with pipetting errors were eliminated from the plot. To verify that quantification was performed in the linear dynamic range, it was ensured that the standard curve was linear ($R^2 > 0.95$) and that the range of Cq values spanned by the standard curve encompassed the Cq values of the test reactions as much as possible. The slope, y-intercept, PCR efficiency, and $R^2$ value were extracted from the standard plot.

The Cq values of the technical replicates for each biological sample were used to determine the average Cq value ($Cq_{avg}$) and standard deviation of Cq ($\sigma Cq$) for each target. $Cq_{avg}$ was used to determine the relative transcript level of the amplicons in RT-qPCR experiments and the relative interaction frequency in 3C-qPCR. $\sigma Cq$ was used to determine the negative and positive errors for the biological samples. The calculations were done using the formula: $10\wedge((Cq-Intercept)/Slope)$. This processing was applied for all Test amplicons and Internal control candidates (Supplementary Data 1C, 1D, 1G, and 1H).

The relative transcript levels and relative interaction frequencies of the Test amplicons were normalised to an Internal control amplicon to allow comparisons to be drawn between biological samples. This was done by assigning an arbitrary value of 100 to the relative transcript level or interaction frequency of the selected Internal control. The values corresponding to the Test amplicons for all biological samples were re-calculated accordingly. Since all amplicons were quantified from standard curves plotted from the same standard samples, inter-assay comparisons were made possible.

## Genomic DNA preparation

An isolated *Escherichia coli* colony was cultured to stationary phase in LB medium at 37 °C with shaking at 200 rpm. Cells in 1.5 mL of the culture were collected by centrifugation at 10000 x *g* at room temperature and resuspended in 400 µL of TES buffer (50 mM Tris-HCl, 10 mM NaCl, 10 mM EDTA, pH 7.5). The cell suspension was incubated with 1.0% Sarkosyl, 100 µg/mL RNase A (Catalogue number: 19107, Qiagen), and 100 µg/mL Proteinase K (Catalogue number: 19157, Qiagen) at 65 °C until the solution cleared. 400 µL of 4 M NH₄OAc was added to the lysate, and the solution was extracted twice with 25:24:1 phenol:chloroform:isoamyl alcohol (Catalogue number: P3803, Sigma-Aldrich) and once with chloroform (Catalogue number: 319988, Sigma-Aldrich). Genomic DNA was precipitated with an equal volume of isopropanol (Catalogue number: 33539, Sigma-Aldrich) for 10 minutes at room temperature and pelleted by centrifuged for 20 minutes at 25000 x *g* at 4 °C. The pellet was dissolved in 400 µL of 0.1 M NaOAc (Catalogue number: S2889, Sigma-Aldrich) pH 6.0, and re-precipitated with 800 µL of cold 100% ethanol for 15 minutes at room temperature. The precipitated DNA was collected by centrifugation at 25000 x *g* for 15 minutes at 4 °C. The pellet was washed with cold 70% ethanol and air dried. Genomic DNA was dissolved in 1X TE pH 8.0 (Catalogue number: 574793, Sigma-Aldrich).

## Term-seq

*E. coli* MG1655 was grown in LB medium at 37 °C with shaking until exponential phase ($OD_{600}$ = ~0.5). Cells were harvested by centrifugation and flash frozen in liquid nitrogen. The cells were lysed using the RNAsnap protocol[130] and total RNA was purified using the Qiagen RNeasy Plus Mini kit (Catalogue number: 74134, Qiagen). Term-seq library preparation and sequencing was performed by Vertis Biotechnologie AG. Briefly, the 5′ TruSeq Illumina adapter was ligated to 3′ hydroxyl ends of rRNA-depleted RNA followed by first-strand cDNA synthesis using M-MLV reverse transcriptase. First strand cDNA was fragmented and the 3′ TruSeq Illumina adapter was ligated to the 3′ ends of single-strand cDNA. cDNA was amplified with a high-fidelity DNA polymerase and purified with Agencourt AMPure XP beads

(Catalogue number: A63881, Beckman Coulter Genomics). Libraries were analysed by capillary electrophoresis, and sequencing was performed on an Illumina NextSeq 500 platform using $1 \times 75$ bp read length. Term-seq experiments were performed in duplicate.

Due to the method of library preparation, raw Term-seq reads are in the anti-sense orientation. The reads were reverse complemented and aligned to the *E. coli* reference genome U00096.3 (https://www.ebi.ac.uk/ena/browser/view/U00096) using Bowtie2 to give binary alignment map (BAM) files[131]. To obtain read-depth data for aggregate plots, BAM files were converted to wiggle plots with bam2wig.py[132,133].

## H-NS purification

*Escherichia coli* BL21 Δ*hns* pLysE pRD18 was cultured in LB medium (1.0% bactotryptone (BD), 0.5% yeast extract (Alfa Aesar), 170 mM NaCl, pH 7.5) at 37 °C to an $OD_{600}$ of 0.4. Expression of H-NS from pRD18 was induced with a final concentration of 120 µg/mL IPTG for 3 hours. The cells were collected by centrifugation at 7000 x *g* for 15 minutes at 25 °C. Cell pellets were resuspended in 20 mL low salt H-NS buffer (130 mM NaCl, 20 mM Tris, 10% glycerol, 8 mM β-mercaptoethanol, and 3 mM benzamidine, pH 7.2) containing 100 mM $NH_4Cl$ with 1 µg/mL DNase, 100 µg/mL lysozyme, and 1 mM PMSF. The cells were lysed by sonication and the soluble fraction was loaded onto a P11 column. H-NS was eluted in low salt H-NS buffer with $NH_4Cl$ using a gradient of 100 mM to 1.0 M $NH_4Cl$ in 100 mL at a rate of 1 mL/min. Fractions with high concentrations of H-NS were pooled and dialysed overnight at 4 °C into low salt H-NS buffer. The dialysed sample was loaded onto a 1 mL HisTrap™ HP column (Catalogue number: 29051021or 17524701, Cytiva) and eluted in H-NS buffer (20 mM Tris, 10% glycerol, 8 mM β-mercaptoethanol, and 3 mM benzamidine, pH 7.2) using a gradient of 130 mM to 1.0 M NaCl in 20 mL at a rate of 1 mL/min. The fractions with high H-NS concentrations were pooled and dialysed at 4 °C overnight into low salt H-NS buffer. The H-NS was loaded onto a 1 mL Resource-Q column (Catalogue number: 17117701, Cytiva) and eluted into a highly concentrated and pure peak with a block elution in H-NS buffer from 200 mM to 500 mM NaCl. The purified H-NS was dialysed at 4 °C overnight into H-NS buffer with 300 mM KCl. H-NS was separated into 50.0 µL aliquots, flash-frozen in liquid nitrogen, and stored at -80 °C until use.

## Electrophoretic mobility shift assay

Purified H-NS was serially diluted in low salt binding buffer (130 mM NaCl, 20 mM Tris, 10% glycerol, 8 mM β-mercaptoethanol, and 3 mM benzamidine, pH 7.2). The dilutions were mixed with an equivalent volume of 400 nM $DRE^{mut}$ or $DRE^{wt}$ (Supplementary Fig. 4; Supplementary Data 1A) in nuclease-free water (Catalogue number: AM9932,ThermoFisher Scientific) and incubated for 20 minutes at 25 °C followed by 10 minutes at 4 °C. The samples were resolved on a polyacrylamide gel (Mini-Protean® TGX™ precast gels, 4-15%, Catalogue number: 4561086, Bio-Rad) at 30 V at 4 °C. Experiments were performed in triplicate. The polyacrylamide gels were stained for 45 minutes in a solution of 10X GelRed (Catalogue number: 41003, Biotium) and thereafter imaged with GelDoc™ XR+ (Catalogue number: 10000076955, Bio-Rad) using Bio-Rad's ImageLab software.

## Reporting summary

Further information on research design is available in the Nature Portfolio Reporting Summary linked to this article.

## Data availability

All data generated in this study have been deposited in the 4TU Repository (https://doi.org/10.4121/21065275). Hi-C data are also available from the NCBI GEO repository under accession number GSE214511. The RT-qPCR and 3C-qPCR data generated in this study are also provided in the Supplementary Information files. Source data are provided with this paper. E. coli reference genomes U00096.2 (https://www.ebi.ac.uk/ena/browser/view/U00096) and NC_000913.3 (https://www.ncbi.nlm.nih.gov/nuccore/556503834) were used in this study. Source data are provided with this paper.

## Code availability

SnapGene Viewer (RRID:SCR_015053) is a freely available program developed by Dotmatics (Boston, Massachusetts, U.S.A.) for viewing and annotating sequence files. Bio-Rad CFX Manager™ (RRID:SCR_017251) Version 3.1 is an experiment setup and data analysis software for all CFX Real-Time PCR Detection Systems (Bio-Rad Laboratories, Inc., Hercules, California, U.S.A.).

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

## Acknowledgements

This research was supported by grants from the Netherlands Organisation for Scientific Research [VICI 016.160.613/533 and OCENW.G-ROOT.2019.012](RTD) and the Human Frontier Science Program (HFSP; RGP0014/2014) (RTD, DWH, and DCG). We thank James Haycocks for his help in setting up the Chromosome Conformation Capture technique in

the very early stages of the project, Amin Allahyar for his help with data processing and visualisation, and Taku Oshima for providing the ChIP data for Supplementary Fig. 3. We acknowledge the Utrecht Sequencing Facility (USEQ) for providing sequencing service and data. USEQ is subsidised by the University Medical Center Utrecht and The Netherlands X-omics Initiative (NWO project 184.034.019).

## Author contributions

R.T.D. conceived and supervised the project. F.M.R. and F.G.E.C. optimised the chromosome conformation capture-based studies and performed Hi-C experiments. A.H. analysed Hi-C data. D.W.H. supervised Hi-C data analysis. F.M.R. engineered the strains, performed RT-qPCR and 3C-qPCR experiments, and analysed the data. D.F. performed Term-seq experiments. D.F. and D.C.G. analysed Term-seq data. F.M.R. wrote the manuscript. R.T.D. edited the manuscript.

## Competing interests

The authors declare no competing interests.

## Additional information

[1]Macromolecular Biochemistry, Leiden Institute of Chemistry, Leiden University, Leiden 2333CC, The Netherlands. [2]Centre for Microbial Cell Biology, Leiden University, Leiden 2333CC, The Netherlands. [3]Centre for Interdisciplinary Genome Research, Leiden University, Leiden 2333CC, The Netherlands. [4]Laboratoire Infection et Inflammation, INSERM, UVSQ, Université Paris-Saclay, Versailles 78180, France. [5]Statistical Physics and Theoretical Biophysics, Heidelberg University, Heidelberg D-69120, Germany. [6]School of Biosciences, University of Birmingham, Edgbaston B15 2TT, UK. ✉e-mail: rtdame@chem.leidenuniv.nl

