## [Peer Review File · Nature Communications]

The environmentally-regulated interplay between local three-dimensional chromatin organisation and transcription of proVWX in *E. coli*Reviewers' comments:

Reviewer #1 (Remarks to the Author):

The manuscript submitted by Rashid et al. addresses the long-standing questions of the interplay of chromatin organization and transcription in E.coli with focus on the nucleoid-associated protein H-NS and the osmo-stress induced proVWX operon, encoding a transport system for compatible solutes. The authors present a conclusive model on osmo-induction of proVWX and the role of H-NS in osmo-regulation. The model states that a compact local H-NS-DNA bridged chromatin complex is loosened in response to osmo-stress, as DNA-bridging by H-NS is K⁺ sensitive. This de-stabilization of the H-NS nucleoprotein complex in combination with an osmo-responsive proVWX promoter leads to a further transcription induced de-compaction and concomitant upregulation of proVWX expression. The model is based on sophisticated 3C and qRT-PCR analyses of remarkable quality. The relevance of this work is broad.

Comments

(237) I suggest to change "lower expression level" to "lower transcript level" here and throughout the manuscript

(241 to 263) please consider that a TSS was mapped within proW chr:2,806,877 see <https://www.nichd.nih.gov/research/atNICHD/Investigators/storz/data-protocols>
https://genome.ucsc.edu/cgi-bin/hgTracks?hubUrl=https://hpc.nih.gov/~NICHD-core0/storz/trackhubs/ecoli_dRNAseq/hub.hub.txt&hgS_loadUrlName=https://hpc.nih.gov/~NICHD-core0/storz/trackhubs/ecoli_dRNAseq/session.txt&hgS_doLoadUrl=submit

(288-289, 293-295) as to "provokes the hypothesis", this is known for other operons, e.g. gal, glmUS. Further, in E.coli proV, proW, proX are not regulated independently, as their expression depends on P2; though, their transcript levels are different indicative of additional independent levels of regulation (e.g. RNA processing and degradation).

(382-383, 688-690) The conclusion is unclear; assuming that upon adaption to HS the K⁺ level drops (due increase of compatible solutes such as glycine betaine, trehalose etc), repression by H-NS could be re-established at HS. It might help if the authors briefly referred to the results obtained with the rnc mutant at HS, which suggest that processing by RNase III contributes to downregulation of proU transcript levels at HS.(688-690) Apparently, refers back to 382-383. Could RNaseIII mediated processing and subsequent RNA-degradation contribute?

(454 to 467) Is it possible that the variations of transcript levels of P1 and the flanking genes (nrdF and ygaJ) are all attributable to effects of delta-stpA on the qRT-PCR standards rpoS and hcaT, respectively, and thus just relative changes?

(492 to 525) It is proposed that RNaseIII degrades the RNA from the 3' end. It is possible that an additional RNase III processing site is present at the 3' end of the RNA. However, degradation of the RNaseIII processed RNA is more likely a secondary event and involves other RNases, such as RNase E. Such a model would correspond better to the current view of RNA degradation. In brief, RNase III processing generates a mono-phosphorylated 5' end and abrogates translation of the RNA downstream of the processing site. This might trigger degradation by RNase E. I suggest to modify the conclusion.

(735-760) The role of IHF does not become clear

(769) I recommend to differentiate between repression of transcription initiation and transcription elongation, as only H-NS bridges repress elongation, while initiation is presumably repressed by both forms (bridges and filaments).

(774-775) Possibly rephrase to "The disruption of the DRE-H-NS-URE bridge in combination with the osmo-responsive P2 promoter...."

(Figure 9) Could "dismantling of the K⁺ weakened H-NS bridge by transcription" be included in the model?

Data presentation

(Figures 2 to 5) the order of the graphs and the way fold-changes and comparisons are presented vary from figure to figure. I suggest to streamline the figures.

(Figures 2 and S5) the figures are identical

(Figures 3 and S9) the figures are identical

(Figures 4 and S10) the figures are identical

(Figures 5 and S11) the figures are identical

(Figure 6) The graphs are very small and the lettering is unreadable. Could just one graph of sufficient size be shown that includes all three curves?

(Figure 7) An overlap of ABC would be helpful, the single graphs A, B, and C could be omitted. All graphs and the lettering is too small.

(Figure 8) Omit A, B, and C as an overlap is shown in J. K could be omitted as it is redundant to J (344-345) please add a figure or table in the supplement in which the sequence changes of NT644 as compared to the wt are compiled.

Editorial comments

(22 to 42, and elsewhere): The statements on the role of NAPs in bacterial chromosome compaction and organization is very strong, possibly too strong. Other factors such as DNA supercoiling, SMC-type proteins (NAPs?), fluidity of the cytoplasm, molecular crowding, possibly co-transcriptional translation have been shown and/or been discussed to contribute as well. Therefore, the words "driving its ..." and "determine" might be toned down.

(46) possibly change to "...binds to the minor groove of AT-rich DNA with an AT-hook like motif..."

(53 to 58) Rephrase to clarify that H-NS filaments may impair binding of RNA polymerase to promoters as well as H-NS – DNA – H-NS bridged complexes. Clarify as well that bridged complexes in addition impair transcription elongation, while filaments do not.

(109-111) The sentence "Cellular influx of K⁺ upon a switch to a higher osmolarity may drive a K⁺-mediated reorganisation of the H-NS—DNA complex into a transcriptionally conducive unbridged structure to relieve proVWX repression 20,26,50,69" should be modified as this has not shown before, but is an hypothesis in agreement with 20,26,50,69. In addition, rephrase „transcriptionally conducive unbridged structure" as this has been shown for transcription elongation, but not transcription initiation.

(89 to 148) is an amazing summary of publications on proVWX. It is rather long though, and I do not know how it could be simplified. Possibly a simplified schematic figure depicting the proVWX regulatory elements could be added and part of the details shifted to the figure legend?

(228 to 230) modify to "...of the proU operon involved transcriptional and translational lacZ fusions ..." add references.

(324 to 332) refer to the figure

(355 to 362) The paragraph is complex, possibly re-phrase? E.g. Expression analyses using lacZ reporter fusions as in 48 may be less suitable, as they mask down-shifts of expression at HS, because beta-galactosidase is a stable protein?

(473-475) Omit sentence "The report in 118 was limited..." as the regulatory role of RNaseIII on proVWX was studied both at the native, chromosomal locus and using plasmids.

(References) several references are listed twice.

Reviewer #2 (Remarks to the Author):

Studies on osmotic regulation of transcription of the proVWX (also called proU) operon in *Escherichia coli* and its close relative *Salmonella typhimurium*, since its discovery in the mid-1980s, have indicated that such regulation is not mediated by a classical or canonical osmo-specific activator or repressor protein. Instead the studies have implicated various factors (not necessarily mutually exclusive) such as potassium glutamate; DNA supercoiling; RpoS; nucleoid-associated proteins (NAPs) including H-NS, StpA, HU and IHF; as well as multiple cis mechanisms, in this regulation. However, a complete understanding of the process has remained elusive. In this manuscript, Rashid et al report data from Hi-C, RT-qPCR, and 3C-PCR experiments to conclude that H-NS orchestrates the 3-D architecture at proU, changes in which are responsible for its transcriptional regulation by growth medium osmolarity.

Comments:

1. There are two kinds of difficulties with the experiments reported in this manuscript as elaborated below, one related to the RT-qPCR experiments and the other to the Hi-C and 3C-PCR experiments.

2. The authors have performed RT-qPCR to determine transcriptional regulation for different discrete regions of proVWX and argue that this approach has advantages over that of using transcriptional and translational reporter gene fusions as had been done by other groups previously. However, their arguments are not convincing, nor consequently their attempts to offer meaningful biological interpretations for the differences in RNA abundance that they observe between different regions of the operon. This is especially so because the earlier studies with reporter fusions have demonstrated identical transcriptional regulation across the proV, proW, and proX genes (PMID 2649478, see also refs. 31 and 33). It is possible, indeed likely, that differences in the RT-qPCR data for the different regions within the operon are because of (a) differences in primer-binding efficiencies during RT-qPCR and / or (b) differences in physical degradation of different mRNA regions by ribonucleases within the cell (since the RT-qPCR method measures RNA abundance and not rate of mRNA synthesis alone).

3. Therefore the authors claims for (a) an additional internal promoter for proX (page 8, line 243); (b) a terminator between proV and proW (page 8, line 241); (c) RNAP recycling for increased proX expression (page 8, line 249); and (d) regulation by G4 quadruplex within proW, are not sustainable in the absence of support from additional independent lines of evidence.

4. Likewise on the basis of RT-qPCR data, the authors state that loss of StpA in a strain proficient for H-NS is associated with derepression of proVWX expression. This finding is questionable, however, since there is abundant evidence (ref. 81; PMID 9473058, PMID 11278074) that StpA is a molecular back-up to H-NS and that its absence alone has little effect in *E. coli*. Furthermore, PMID 9473058 has clearly shown that proVWX regulation is unaffected in a stpA single mutant.

5. By using LB with 0.08 M NaCl as their low-osmolarity medium, the authors have missed out on sampling the full range and magnitude of proVWX osmotic regulation, which is at least 200-fold. This is so since there is another order of magnitude of repression between 0.08 M NaCl and 0.01 M NaCl (refs. 31, 33, 57). A mere extrapolation of their model to cover the expanded range may not necessarily be appropriate here, since the full magnitude of regulation appears to be achieved through

a combination of mechanisms.

6. The authors indicate that because of an adaptation response, proVWX expression during steady-state growth at high osmolarity is only 1.3- to 2-fold higher than that during growth at low osmolarity, but the values reported by them are once again contrary to published data from many groups on proVWX expression measurements during steady state growth at different osmolarities (refs. 29, 31). The latter indicate that up-regulation of proVWX even during steady-state growth at high osmolarity is at least 100-fold, with a sigmoidal pattern of elevated expression observed with progressive increase of growth medium osmolarity.

7. With regard to the Hi-C and 3C-qPCR experiments, the authors have primarily catalogued the osmolarity-associated alterations at the proVWX locus and then attempted to build mechanistic models of regulation from these data. However, the fundamental questions that remain unaddressed are (a) to what extent are those alterations specific to proVWX in comparison to other control loci, and (b) are these alterations causally relevant to its transcriptional control (and not consequential, or even incidental, to the changes in transcription)? In other words, how do changes in growth medium osmolarity affect other control chromosomal loci, and what are the changes that occur when transcription is activated at other control operons by their cognate environmental cues? In the absence of such information, it may not be possible to consider as logical or meaningful the interpretations / conclusions sought to be drawn from the proVWX-related changes.

8. The authors' suggestion in the Abstract and at several places in the text, that YgaG participates in osmotic regulation of proVWX transcription, flies in the face of a wealth of published data that a 1.2-kb region, spanning the NRE and including the promoter(s), is sufficient to fully reconstitute proVWX osmotic regulation (refs. 49, 56, 57, PMID 7968545). Such regulation is observed even if this region is shifted to other chromosomal locations, or is cloned on a plasmid.

9. Page 19, para 2 is confusing. The authors employ mutations that "are expected to weaken IHF binding affinity", but conclude from the data obtained that it "highlights that the interaction is mediated by H-NS rather than IHF".

10. By combining Results and Discussion into a single section in the manuscript, the authors appear to have camouflaged their indulgence in extensive speculation from a relatively limited set of experimental data. For example, in each of Figures 2, 3, 4 and 5, only three panels represent original experimental findings whereas an additional five to eight panels have been added that comprise derivative information. Furthermore, the model upon which much of their speculation is founded, of instantaneous potassium influx upon an osmo-shock driving the relief from H-NS mediated repression of proVWX transcription, is itself not universally accepted (see PMID 7929004).

11. The authors use the term "osmosensitive" to refer to changes in proVWX transcription in response to changes in growth medium osmolarity, but this is confusing since the word would normally be expected to refer to a growth-sensitivity phenotype.

12. There are many errors in the list of References. Ref. 21 cites a Cell 2015 preprint. Of greater concern is the unconscionably large number of duplicate entries, of which I counted at least 20 pairs: 3-91; 94-102; 31-43; 32-44; 52-53; 45-88; 46-54; 47-55; 51-69; 70-74; 71-76; 84-87; 82-90; 80-113; 60-120; 37-130; 133-135; 144-145; 119-147; 57-118.

Reviewer #3 (Remarks to the Author):

In this manuscript, Fatema-Zahra et al. used RT-qPCR and 3C-qPCR in different cell lines (wild-type and mutants) to investigate the relationship between transcription regulation and local folding of the

E. coli chromosome. They focused on the transcriptional properties of the ProU operon, which is a well-known case of regulation by nucleoid-associated proteins (NAPs), particularly H-NS. From their study, they concluded that repression of ProU at low salt concentration occurs through the formation of two bridges involving H-NS between the downstream regulatory elements (DRE) and the upstream regulatory elements (URE) and the *ygaY* gene, respectively.

In my opinion, this work provides an interesting review of our knowledge of the action of H-NS in the structuring of DNA *in vitro*. It also provides interesting novel results about the effect of different mutants on the transcriptional properties of the ProU operon. However, I think that the authors' conclusions are not supported by their observations and therefore leave open the question of the precise mechanisms involved *in vivo* for the regulation of ProU and, more generally, for the regulation phenomena involving H-NS. Thus, I think that the results obtained are more suitable for a more specific journal.

More specifically, the new results on which the conclusions of this work are based are the results of 3C-qPCR in Figures 6 and 7, which need to be compared with the results in Figures 2 and 3. There are three major problems in this regard: firstly, the results in Figure 6A do not demonstrate that there is a loop between DRE and *ygaY*. The authors only hypothesize this loop based on the observation of a very slight decrease in contact frequencies (less than 10% at first sight). Secondly, the size of the fragments induced by NlaIII is too large to conclude that DRE forms a loop with URE. Finally, if there is a loop between DRE and URE, there must necessarily be a loop between DRE and *proU13* located between *nrdE* and *nrdF*, given the values of the frequencies obtained. In addition, during hyperosmotic shock, the entire region between *nrdE* and DRE is affected, as can be seen in Figure 6F, which goes against the idea that the observed variations concern only a single loop (between DRE and URE). In fact, there is at least one control missing in this work to conclude anything about the presence of loops between URE and other elements: a 3C-qPCR profile centered not on URE but, for example, on *proU6*. In particular, variations in 3C interactions can be related to variations in DNA-binding proteins because the 3C signal reflects the crosslinking between proteins and DNA and between proteins themselves. A control like *proU6* should therefore either confirm or refute this hypothesis.

Continuing on the theme of contact data between chromosomal loci, I find the choice to start with a general Hi-C analysis for this article not relevant and rather confusing. In particular, the analyses and conclusions discussed based on Figure 1 do not seem relevant compared to the rest of the discussion. In fact, the conclusions about the different relevant distances of the system seem incorrect: it is not reasonable to identify interaction distances based solely on a discussion of when the colors of a matrix change. These colors vary widely depending on the visualization choices used to represent the data and therefore cannot be used as a robust indicator. Furthermore, matrix comparisons are usually made using the log-ratio of the matrices, which in this case can actually lead to defining relevant distances for the action of NAPs (see e.g. [93]). In summary, the results in Figure 1 are very qualitative and do not provide a comprehensive explanation of the properties observed throughout the matrix.

More generally, I found that the results were often presented as a discussion, with hypotheses often justified from an *in vitro* standpoint, but whose relevance *in vivo* is not clear.

Please find an additional list of suggestions for the authors:

- * Line 78 and 79: the terms P1R and P2R are unnecessary since they are not used elsewhere in the article
- * Lines 95-96: Is the statement "Heterologous promoters..." associated with an H-NS phenomenon? If so, this should be explained. If not, this sentence should instead appear in the previous paragraph.
- * Lines 118-119: The sentence "The effect of *ihf* deletion..." is not clear (i.e., which effect?)
- * The sentence "HU regulates expression..." at lines 120-121 is problematic.
- * Lines 222 and 223: "One of the regions..." Many regions actually seem to be affected, with one, for

instance, in the NSR. A more systematic analysis would be required here.

* Line 452: "This may, however,..." The authors could elaborate.

* Fig. S7: It would be useful to show the ChIP-seq signal of H-NS over the entire segment discussed by the authors (from *nrdE* to *ygaZ*).

* Fig. S7: It would be useful to show the ChIP signal of StpA discussed by the authors.

Reviewers' comments:

> Please note that line references for the revised manuscript are provided according to the Revised_Manuscript.pdf file

Reviewer #1 (Remarks to the Author):

The manuscript submitted by Rashid et al. addresses the long-standing questions of the interplay of chromatin organization and transcription in E.coli with focus on the nucleoid-associated protein H-NS and the osmo-stress induced proVWX operon, encoding a transport system for compatible solutes. The authors present a conclusive model on osmo-induction of proVWX and the role of H-NS in osmo-regulation. The model states that a compact local H-NS-DNA bridged chromatin complex is loosened in response to osmo-stress, as DNA-bridging by H-NS is K⁺ sensitive. This de-stabilization of the H-NS nucleoprotein complex in combination with an osmo-responsive proVWX promoter leads to a further transcription induced de-compaction and concomitant upregulation of proVWX expression. The model is based on sophisticated 3C and qRT-PCR analyses of remarkable quality. The relevance of this work is broad.

> We thank the reviewer for the very positive evaluation of the quality of our analysis and the relevance of our studies. We found the comments very constructive and enjoyed addressing them. Comments for lines (382-383, 688-690) and (454 to 467) were particularly interesting to us. Addressing these comments much improved that part of our manuscript.

Comments

(237) I suggest to change “lower expression level” to “lower transcript level” here and throughout the manuscript

> ‘Transcript level’ is indeed more representative of the data that we show than ‘expression level’. We have made this change throughout the manuscript. We have also changed all our graphs accordingly.

(241 to 263) please consider that a TSS was mapped within proW chr:2,806,877 see <https://www.nichd.nih.gov/research/atNICHD/Investigators/storz/data-protocols>
https://genome.ucsc.edu/cgi-bin/hgTracks?hubUrl=https://hpc.nih.gov/~NICHD-core0/storz/trackhubs/ecoli_dRNAseq/hub.hub.txt&hgS_loadUrlName=https://hpc.nih.gov/~NICHD-core0/storz/trackhubs/ecoli_dRNAseq/session.txt&hgS_doLoadUrl=submit

> We thank the reviewer for referring us to these data, which were previously unknown to us. Including these observations strengthens and refines our model of proVWX regulation.

Lines 247-252 of the original manuscript (lines 186-197 of the of the revised manuscript) now read, ‘Indeed, with Term-seq – a technique used to identify transcription termination sites⁸¹ – the presence of a termination site downstream of *proW1* at chr:2806395-2806398 was detected in our study and at chr:2806397-2806401 in an earlier report⁸² (Accession number: NC_000913.3) (Figure S2, SI 1A). Despite its positioning downstream of *proW1* (SI 1A), the terminator may have had an impact on the *proW1* RT-qPCR signal. Our Term-seq studies also identify a termination site between *proW* and *proX* downstream of a putative hairpin structure (Figure S2, SI 1A), however, this site was not detected in Term-Seq studies performed by others⁸². TSS mapping using differential RNA sequencing⁸³ identifies a TSS within *proW* at chr:2806878 (NC_000913.3)⁸⁴ positioned 23 basepairs downstream of the *proW2* amplicon. Transcription initiating from this site may account for the increase in the transcript levels of *proX*. The promoter regulating expression from this site is likely a non-canonical promoter rather than a classical H-NS-repressed spurious promoter since H-NS ChIP shows no H-NS signal at *proW* (Figure S3)⁸⁵, and spurious transcripts have not been detected from *proW* in a Δhns background⁸⁶.’

Note that the sentence, ‘An A-rich tract also occurs immediately downstream of this site and may promote the expression of *proX* by RNAP recycling – transcription re-initiation by post-transcriptional RNAP diffusing

one-dimensionally on DNA ⁹⁶.' has been deleted since the hypothesis is not corroborated by differential RNA-Seq data (see comment 3 of reviewer #2).

We examined the tracks that were provided, and identified chr:2806878 as the +1 position. We have annotated this as transcription start site mapped within *proW* in Supplementary file 1.

(288-289, 293-295) as to “provokes the hypothesis”, this is known for other operons, e.g. *gal*, *glmUS*. Further, in *E. coli* *proV*, *proW*, *proX* are not regulated independently, as their expression depends on P2; though, their transcript levels are different indicative of additional independent levels of regulation (e.g. RNA processing and degradation).

> We fully agree with this comment of the reviewer and admit that we did not phrase our statement regarding the additional levels of independent gene regulation operating on *proVWX* correctly. The sentences (lines 253-256) now read, ‘The results show that the *proV*, *proW*, and *proX* genes that occur within the same operon and are co-regulated by the mechanisms operating on P2 exhibit a degree of independent regulation. The presence of additional regulatory mechanisms operating on individual genes of an operon has been documented previously, for instance, at the *gal*⁸⁹ and *glmUS*⁹⁰ operons.’

(382-383, 688-690) The conclusion is unclear; assuming that upon adaption to HS the K⁺ level drops (due increase of compatible solutes such as glycine betaine, trehalose etc), repression by H-NS could be re-established at HS. It might help if the authors briefly referred to the results obtained with the *rnc* mutant at HS, which suggest that processing by RNase III contributes to downregulation of *proU* transcript levels at HS. (688-690) Apparently, refers back to 382-383. Could RNaseIII mediated processing and subsequent RNA-degradation contribute?

> This is a very insightful suggestion. Indeed, RNaseIII still operates in cells adapted to high extracellular osmolarity, but has minimal impact in cells growing at low osmolarity. We have now included this hypothesis in our manuscript.

The following changes have been made:

- The sentence in lines 383-384 of the original manuscript (lines 318-321 of the revised manuscript) now reads, ‘This suggests that in *E. coli* cells adapted to higher osmolarities, the repression of *proU* may be mediated by factors other than H-NS occupancy at the *proU* promoter, for instance, RNaseIII-mediated processing and subsequent degradation (see below)’
- The following sentence has been added to the paragraph at lines 483-486 of the revised manuscript: ‘The function of RNaseIII upon adaptation to high osmolarity compared to growth at low osmolarity may explain why the *proVWX* transcript levels in NT644 during exponential growth at 0.3 M NaCl are as low as up to ~25% that at 0.08 M NaCl (Figures 2g, 2h, S6g, and S6h).’
- The sentence in line 688-690 of the original manuscript (lines 679-682 of the revised manuscript) now reads ‘The decline in expression occurs despite the operon remaining locked in its H-NS deficient structure, supporting our earlier hypothesis that in NT644 at high osmolarity, a factor other than H-NS represses *proVWX*, such as RNaseIII-mediated transcript processing and subsequent RNA degradation.’

(454 to 467) Is it possible that the variations of transcript levels of P1 and the flanking genes (*nrdF* and *ygaJ*) are all attributable to effects of Δ -*stpA* on the qRT-PCR standards *rpoS* and *hcaT*, respectively, and thus just relative changes?

> This is unlikely. To check whether Δ *stpA* may affect *rpoD* and *hcaT*, we reviewed the ChIP profiles of StpA-3xFLAG and H-NS-3xFLAG of *Escherichia coli* K-12 W3110 reported by Uyar et al., 2009, J Bacteriol, StpA does not have a strong binding signal at *rpoD* or *hcaT*.

Moreover, an effect of ΔstpA on the internal controls should equally affect all the amplicons instead of a subset of amplicons as in our data.

(492 to 525) It is proposed that RNaseIII degrades the RNA from the 3' end. It is possible that an additional RNase III processing site is present at the 3' end of the RNA. However, degradation of the RNaseIII processed RNA is more likely a secondary event and involves other RNases, such as RNase E. Such a model would correspond better to the current view of RNA degradation. In brief, RNase III processing generates a mono-phosphorylated 5' end and abrogates translation of the RNA downstream of the processing site. This might trigger degradation by RNase E. I suggest to modify the conclusion.

> We thank the reviewer for the clear insight in the current views on the role of RNaseIII. We have modified our conclusions as follows:

- The sentence 'This indicates an increased stability in the absence of RNaseIII and suggests that in addition to the processing of the conserved hairpin located at +203 to +293 of the *proU* transcript, RNaseIII downregulates the expression of *proU* by degrading *proV* transcripts from the 3' end.' in the paragraph at lines 492-525 of the original manuscript (lines 440-446 of the reviewed manuscript) now reads 'This indicates an increased stability of the 3' region of *proV* transcripts in the absence of *rnc*, suggesting that RNaseIII contributes to processing of the 3' end of the transcript. This can be a secondary event involving other RNases such as RNaseE. Processing of the conserved hairpin located at +203 to +293 of the *proU* transcript generates a monophosphorylated 5' end. Such termini can be recognised and bound by RNaseE, promoting degradation⁹⁷. However, it is also possible that an RNaseIII processing site may be present proximal to the 3' end of *proV* transcripts.'
- The phrase '...reinforcing that RNaseIII-mediated post-transcriptional regulation of *proU* expression also involves processing from the 3' ends of *proV* transcripts.' now reads, '...reinforcing the proposal that RNaseIII-mediated post-transcriptional regulation of *proU* expression may involve processing at a site proximal to the 3' ends of *proV* transcripts.' (lines 474-476 of the revised manuscript)

(735-760) The role of IHF does not become clear

> In this section we have written about the role IHF may play in dismantling the URE-H-NS-DRE bridge. We have now modified the section at lines 745-752 in the original manuscript to 'IHF is a Y-shaped NAP consisting of an α -helical body and a pair of β -sheet arms extending outwards. Proline residues positioned at the tip of the β -arms intercalate between bases generating sharp bends in the DNA¹⁰⁰. In its fully-bound state, the proline residues of IHF insert at two sites in the DNA, 9 bp apart, producing a $>160^\circ$ DNA bend with flanking DNA folding onto the body of IHF¹⁰⁰. At P2 such a bend can fold the URE onto the DRE, facilitating the formation of an H-NS-mediated bridge between the two elements. DNA bends formed by IHF are flexible and fluctuate between the fully-bound (bend angle $157^\circ \pm 31^\circ$) and partially-bound (bend angle $115^\circ \pm 30^\circ$) conformations^{101,105}. In the partially-bound state, the DNA on one side of the IHF-mediated bend is not folded onto the body of the protein^{101,105}. *In vitro* fluorescence lifetime measurements of the conformations sampled by IHF bound to its high affinity H' site show that the formation of a partially bent IHF-H' complex increases from 22% to 32% upon an increase in the concentration of KCl from 100 mM to 200 mM¹⁰⁵. Thus, the intracellular influx of K^+ upon a hyperosmotic shock may shift the equilibrium of IHF binding to favour the formation of a partially-bent IHF-P2 conformation, promoting the disassembly of the DRE—H-NS—URE bridge that is already weakened by K^+ .' (lines 724-736 of the revised manuscript)

(769) I recommend to differentiate between repression of transcription initiation and transcription elongation, as only H-NS bridges repress elongation, while initiation is presumably repressed by both forms (bridges and filaments).

> We thank the reviewer for this suggestion. We have now included the following sentences in our model for *proVWX* regulation (lines 752-758 of the revised manuscript) 'Conditions of low osmolarity favour an

'open' conformational state of H-NS in which both the DNA binding domains of an H-NS dimer are available for interacting with DNA. Under these conditions, an H-NS multimer preferentially bridges DNA loci. At *proVWX*, H-NS bridges the DRE to the URE, and to *ygaY*. The structures interfere with transcription initiation by occluding RNAP from the promoter and preventing promoter escape for promoter-bound RNAP. The bridges also behave as supercoil diffusion barriers and trap positive supercoils generated by elongating RNAP on the downstream template, hence, stimulating RNAP pausing and repressing *proVWX*.'

(774-775) Possibly rephrase to "The disruption of the DRE-H-NS-URE bridge in combination with the osmo-responsive P2 promoter...."

> Due to the textual changes we made and detail that we added to our model of the regulation of *proVWX*, we were not able to use the phrasing suggested by the reviewer. However, the comment indicates to us that we should mention that the osmo-responsive *proVWX* P2 promoter also contributes to the activation of the operon. In line with that, we have added the following to our model (lines 782-787 of the revised manuscript), 'The influx of K⁺ directly activates the osmo-responsive P2 promoter of the *proVWX* operon that encodes a transport system for inert, non-toxic osmoprotectants. The influx of K⁺ also shifts the equilibrium of the conformational states assumed by H-NS from the 'open' to the 'half-sequestered' state where one DNA binding domain per dimer folds on to the body of the protein and becomes unavailable for interacting with DNA. Due to the stochasticity of which DNA binding domain per dimer is sequestered, DNA—H-NS—DNA bridges may either be disrupted or weakened.'

(Figure 9) Could "dismantling of the K⁺ weakened H-NS bridge by transcription" be included in the model?

> In panels III, IV, and V of figure 9B the dismantling of the K⁺ weakened H-NS bridge by transcription is shown.

In our figure the H-NS multimer is shown in red and blue with every red-blue unit representing a dimer. The circles in this drawing are the DNA binding domains of H-NS. Panel I shows each of these DNA binding domains extended outwards. The weakening of the H-NS bridge by the sequestration of the H-NS DNA binding domains on to the body of the protein are represented in subsequent panels as the circles folding onto the body of the protein. The difference between DNA-bound and sequestered H-NS DNA binding domains is subtle in our figure, and it was not explained in the figure legend. We have now included the following information in our figure legend, 'In the figure, this...' [referring to sequestration of the H-NS DNA binding domain] '...is represented as some of the circular DNA binding domains of H-NS interacting with the oval-shaped H-NS body.' (lines 772-773 of the revised manuscript)

Data presentation

(Figures 2 to 5) the order of the graphs and the way fold-changes and comparisons are presented vary from figure to figure. I suggest to streamline the figures.

> The reviewer is correct. We have re-made the figures 1-4 (figures 2-5 of the original manuscript) to make them internally consistent. The figures are now also properly arranged from a-h/j/k from top left to bottom right, and they are organised in the order in which they are referred to in the text. The fold-changes have also been represented uniformly with black dots connected with a line.

(Figures 2 and S5) the figures are identical

> Figures 1 and S1 (figures 2 and S5 or the original manuscript) are not identical. These are graphs that have been plotted using a different internal control to show the reproducibility of the data. Presenting both data sets allows us to comply with 'Data analysis: Justification of number and choice of reference genes' of the MIQE checklist (Bustin et al., 2009, Clin Chem)

(Figures 3 and S9) the figures are identical

> Figures 2 and S6 (figures 3 and S9 or the original manuscript) are not identical. These are graphs that have been plotted using a different internal control to show the reproducibility of the data. Presenting both data sets allows us to comply with 'Data analysis: Justification of number and choice of reference genes' of the MIQE checklist (Bustin et al., 2009, Clin Chem)

(Figures 4 and S10) the figures are identical

> Figures 3 and S7 (figures 4 and S10 or the original manuscript) are not identical. These are graphs that have been plotted using a different internal control to show the reproducibility of the data. Presenting both data sets allows us to comply with 'Data analysis: Justification of number and choice of reference genes' of the MIQE checklist (Bustin et al., 2009, Clin Chem)

(Figures 5 and S11) the figures are identical

> Figures 4 and S8 (figures 5 and S11 or the original manuscript) are not identical. These are graphs that have been plotted using a different internal control to show the reproducibility of the data. Presenting both data sets allows us to comply with 'Data analysis: Justification of number and choice of reference genes' of the MIQE checklist (Bustin et al., 2009, Clin Chem)

(Figure 6) The graphs are very small and the lettering is unreadable. Could just one graph of sufficient size be shown that includes all three curves?

> This is a very helpful suggestion. Figures 5, 6, and 7 (figures 6, 7, and 8 of the original manuscript) have been re-sized so that the letters are readable.

(Figure 7) An overlap of ABC would be helpful, the single graphs A, B, and C could be omitted. All graphs and the lettering is too small.

> We respectfully disagree with this suggestion. We had initially considered making a single graph with all three curves, but decided against this since each single curve also includes the data points for the four biological replicates tested. This doubles as a clearer representation of the original data and error in our measurements.

(Figure 8) Omit A, B, and C as an overlap is shown in J. K could be omitted as it is redundant to J

> Since representing the single curves is important due to the individual data points that are represented in these curves, we have maintained figures 7a, 7b, and 7c (figures 8a-8c of the original manuscript). We have omitted figure 8j of the original manuscript.

(344-345) please add a figure or table in the supplement in which the sequence changes of NT644 as compared to the wt are compiled.

> This is a good suggestion. The sequence changes between the wild-type strain and NT644 have now also been visually represented in supplementary figure S4.

Editorial comments

(22 to 42, and elsewhere): The statements on the role of NAPs in bacterial chromosome compaction and organization is very strong, possibly too strong. Other factors such as DNA supercoiling, SMC-type proteins (NAPs?), fluidity of the cytoplasm, molecular crowding, possibly co-transcriptional translation have been shown and/or been discussed to contribute as well. Therefore, the words "driving its ..." and "determine" might be toned down.

> We thank the reviewer for this insightful comment. In accordance with this comment, we have edited the manuscript to change statements that place too much emphasis on NAPs as chromosome organisers by using terminologies/phrases such as 'maintain' and 'contribute to' instead of 'driving' and 'determine'.

(46) possibly change to "...binds to the minor groove of AT-rich DNA with an AT-hook like motif..."

> We thank the reviewer for this suggestion. We have made this change. The sentence (lines 46-47 of the revised manuscript) now reads, 'The dimer binds to the minor groove of AT-rich DNA with an AT-hook like motif within its C-terminal DNA binding domain ¹⁸.'

(53 to 58) Rephrase to clarify that H-NS filaments may impair binding of RNA polymerase to promoters as well as H-NS – DNA – H-NS bridged complexes. Clarify as well that bridged complexes in addition impair transcription elongation, while filaments do not.

> It is indeed useful to spell out at which stages of the transcription process which mode of binding exhibits a regulatory role. The following clarification has been added (lines 50-56 of the revised manuscript), 'H-NS—DNA filaments can repress transcription by occluding the binding of RNA polymerase to promoter and promoter-like elements trapped within the structures ^{21–23}, but the filaments do not exert a detectable roadblocking effect on elongating RNA polymerase *in vitro* ²⁴. The presence of two DNA binding domains per dimer allows the H-NS multimer to recruit a second DNA molecule to form a DNA—H-NS—DNA bridge ^{2,25–27}. DNA—H-NS—DNA bridges occlude the binding of RNA polymerase (RNAP) to promoters ²⁸, block transcription initiation at the promoter clearance step by trapping RNAP in a loop ²⁹, and impede transcription elongation *in vitro* ²⁴.'

(109-111) The sentence "Cellular influx of K⁺ upon a switch to a higher osmolarity may drive a K⁺-mediated reorganisation of the H-NS—DNA complex into a transcriptionally conducive unbridged structure to relieve proVWX repression ^{20,26,50,69}" should be modified as this has not shown before, but is an hypothesis in agreement with ^{20,26,50,69}. In addition, rephrase „transcriptionally conducive unbridged structure“ as this has been shown for transcription elongation, but not transcription initiation.

> The sentence that the reviewer refers to has now been merged with the sentence before it and modified to clarify that it is a hypothesis in agreement with *in vitro* studies. The phrase 'transcriptionally conducive unbridged structure' has also been edited to 'unbridged structure that is conducive to transcription elongation'.

The sentence. (lines 115-120 of the revised manuscript) now reads, '*In vitro* studies support a hypothesis where at low osmolarity (i.e. low intracellular K⁺), the H-NS—DNA complex organises into a transcriptionally-repressive bridged conformation to silence *proVWX* ^{24,27,49,55,56}, reminiscent of the role of H-NS at the *rrnB* P1, *hdeAB*, and *bgl* promoters ^{29,63,64}, and that the cellular influx of K⁺ upon a switch to a higher osmolarity may drive a K⁺-mediated reorganisation of the H-NS—DNA complex to relieve *proVWX* repression ^{24,27,39,51}'

(89 to 148) is an amazing summary of publications on proVWX. It is rather long though, and I do not know how it could be simplified. Possibly a simplified schematic figure depicting the proVWX regulatory elements could be added and part of the details shifted to the figure legend?

> We thank the reviewer for this suggestion. Generally, we would like to avoid providing new information in the figure legends. We prefer to leave the information in the introduction to be able to provide a complete overview of previous knowledge on *proVWX*.

(228 to 230) modify to "...of the proU operon involved transcriptional and translational lacZ fusions ..." add references.

> We have included references and changed the sentence (lines 169-172 of the revised manuscript) to 'Early studies examining the osmosensitivity of the *proU* operon involved transcriptional and translational fusions of *lacZ* to truncates of *proV*, and the subsequent detection of the specific activity of β -galactosidase (*lacZ*) in a variety of genetic backgrounds and osmolarity conditions as a measure of *proU* expression ^{34,37,39,47,49,50,53–55,67,68,78,79.}'

(324 to 332) refer to the figure

> It is not entirely clear to us what the reviewer means since figures 2H and S5H were referred to in the original manuscript. These correspond to figures 1h and S1h in the revised manuscript (lines 273-280 of the revised manuscript).

(355 to 362) The paragraph is complex, possibly re-phrase? E.g. Expression analyses using *lacZ* reporter fusions as in 48 may be less suitable, as they mask down-shifts of expression at HS, because beta-galactosidase is a stable protein?

> We edited the paragraph as suggested by the reviewer (lines 303-312 of the revised manuscript). It now reads 'The relative transcript level of *proVWX* in NT644 at low osmolarity (0.08 M NaCl) is comparable to the relative transcript level in the wild-type background (NT331) after a hyper-osmotic shock (from 0.08 M to 0.3 M NaCl) (Figures 2c and S6c; Tables S1-S4) and not NT331 at high osmolarity (Figures 2d and S6d; Tables S1-S4). This appears to contradict the *proVWX* osmoregulation studies reported earlier ⁴⁹ which show that the specific activity of β -galactosidase of a *proV-lacZ* fusion in a Δ *hns* background at low osmolarity (0.05 M, and 0.1 M NaCl) is similar to the specific activity of *proV*- β -galactosidase in a wild-type background at high osmolarity (growing exponentially at 0.3 M NaCl). The differences between these two sets of results may be accounted for by the post-translational evaluation of osmoresponse in REF⁴⁹. β -galactosidase is a stable protein. Hence, expression analyses using *lacZ* reporter fusions mask down-shifts of expression upon adaptation to high osmolarity.'

(473-475) Omit sentence "The report in 118 was limited..." as the regulatory role of RNaseIII on *proVWX* was studied both at the native, chromosomal locus and using plasmids.

> Kavalchuk et al., 2012 RNA Biol (reference 118 in the original manuscript; reference 54 in the revised manuscript) indeed studied the regulation of *proU* in the chromosome and using plasmids. However, the chromosomal studies were not performed at the native *proU* locus. The studies involved cloning the -315 to +1260 region centred around the *proU* P2 TSS at the ectopic *attB* locus in a Δ *proVWX* strain. We changed our sentence to clarify this. The statement (lines 430-435 of the revised manuscript) now reads, 'The regulatory role of RNaseIII on *proVWX* expression was evaluated using a construct carrying the -315 to +1260 region centred around the *proU* P2 TSS – a region was expected to contain all essential regulatory elements of *proU*⁵⁴. The construct was cloned into a plasmid or inserted at an ectopic *attB* locus on the chromosome in a Δ *proVWX* background for study ⁵⁴. We built on this report and determined the effect of a Δ *rnc* mutation on the regulation of the entire *proVWX* operon in its native chromosome context.'

(References) several references are listed twice.

> We sincerely apologise for this. During our final check just before starting the submission of our manuscript, we had to update our reference manager from Mendeley Desktop to Mendeley Cite. Due to the incompatibility of Mendeley Cite with the security of our work computers, the references were updated as a last step on a personal computer. Since the references had already been critically checked before the update, the manuscript was submitted immediately. We have made the corrections.

Reviewer #2 (Remarks to the Author):

Studies on osmotic regulation of transcription of the *proVWX* (also called *proU*) operon in *Escherichia coli* and its close relative *Salmonella typhimurium*, since its discovery in the mid-1980s, have indicated that such regulation is not mediated by a classical or canonical osmo-specific activator or repressor protein. Instead the studies have implicated various factors (not necessarily mutually exclusive) such as potassium glutamate; DNA supercoiling; RpoS; nucleoid-associated proteins (NAPs) including H-NS, StpA, HU and IHF; as well as multiple cis mechanisms, in this regulation. However, a complete understanding of the process has remained elusive. In this manuscript, Rashid et al report data from Hi-C, RT-qPCR, and 3C-PCR experiments to conclude that H-NS orchestrates the 3-D architecture at *proU*, changes in which are responsible for its transcriptional regulation by growth medium osmolarity.

Comments:

1. There are two kinds of difficulties with the experiments reported in this manuscript as elaborated below, one related to the RT-qPCR experiments and the other to the Hi-C and 3C-PCR experiments.

> We thank the reviewer for their evaluation of our manuscript.

2. The authors have performed RT-qPCR to determine transcriptional regulation for different discrete regions of *proVWX* and argue that this approach has advantages over that of using transcriptional and translational reporter gene fusions as had been done by other groups previously. However, their arguments are not convincing, nor consequently their attempts to offer meaningful biological interpretations for the differences in RNA abundance that they observe between different regions of the operon. This is especially so because the earlier studies with reporter fusions have demonstrated identical transcriptional regulation across the *proV*, *proW*, and *proX* genes (PMID 2649478, see also refs. 31 and 33). It is possible, indeed likely, that differences in the RT-qPCR data for the different regions within the operon are because of (a) differences in primer-binding efficiencies during RT-qPCR and / or (b) differences in physical degradation of different mRNA regions by ribonucleases within the cell (since the RT-qPCR method measures RNA abundance and not rate of mRNA synthesis alone).

> The referee raises multiple points in this comment. We have separated this comment into smaller sections and address each independently.

'The authors have performed RT-qPCR to determine transcriptional regulation for different discrete regions of *proVWX* and argue that this approach has advantages over that of using transcriptional and translational reporter gene fusions as had been done by other groups previously. However, their arguments are not convincing...'

> This comment is unexpected as the advantages of this approach are – in our view – common knowledge. In lines 230-232 of the original manuscript we explained that 'The technique...' [referring to reporter gene fusions] '...is limited in that it relies on the post-translational detection of gene/operon activity, and it remains relatively ineffective in evaluating the transcriptional profile across an operon. We used RT-qPCR to study osmosensitivity of *proU* at a transcriptional level.'

Post-translational detection of gene expression/regulation requires transcription of the reporter gene construct and translation of the mRNA. RT-qPCR directly quantifies transcript levels. In other words, transcriptional and translational reporter gene fusions are reflective of intracellular protein levels while RT-qPCR reflects transcription. Therefore, for our study of the interplay between local chromatin organisation and gene expression, RT-qPCR is a more suitable technique than reporter gene fusions.

We have now explained our choice of RT-qPCR over transcriptional and translational reporter gene fusions in more detail. Our explanation (lines 169-176 of the revised manuscript) reads, 'Early studies examining the osmosensitivity of the *proU* operon involved transcriptional and translational fusions of *lacZ* to truncates of *proV*, and the subsequent detection of the specific activity of β -galactosidase (*lacZ*) in a variety of genetic backgrounds and osmolarity conditions as a measure of *proU* expression ^{34,37,39,47,49,50,53–55,67,68,78,79}. The

technique is limited in that it relies on post-translational detection. It requires transcription of the reporter gene construct and translation of the mRNA for a read-out. As such, it is more reflective of intracellular protein levels than transcription. RT-qPCR (reverse transcriptase quantitative PCR) directly measures transcript levels. Therefore, we used RT-qPCR to study the osmolarity dependent response of *proVWX*.'

'...nor consequently their attempts to offer meaningful biological interpretations for the differences in RNA abundance that they observe between different regions of the operon.'

> This comment appears to be unsubstantiated. The reviewer does not provide additional explanation for this, so it is unclear to us what he/she is referring to. Our interpretations of the RT-qPCR results consistently refer to 'expression level' in the original manuscript which, on the advice of Reviewer #1 we have changed to 'transcript level' in our revised manuscript. Since RT-qPCR directly quantifies transcript levels, our interpretations are justified.

'This is especially so because the earlier studies with reporter fusions have demonstrated identical transcriptional regulation across the *proV*, *proW*, and *proX* genes (PMID 2649478, see also refs. 31 and 33).'

> We respectfully disagree with this conclusion. Dattananda and Gowrishankar, 1989 J. Bacteriol., Cairney et al., 1985 J. Bacteriol., and Sutherland et al., 1986 J. Bacteriol. are post-translational measurements of *proV*, *proW*, and *proX*, that bypass transcript levels. Our studies with RT-qPCR directly measure transcript levels and are, therefore, more appropriate for studying transcription regulation.

In Table 2 of Dattananda and Gowrishankar, 1989, J. Bacteriol. the data show that the specific activity of β -galactosidase is higher for *proX-lacZ* fusions than *proW-lacZ* fusions, which is in turn higher than *proV-lacZ*. If we do not account for the technical differences in post-translational detection assays and RNA measurement assays, this is in apparent contrast with the RT-qPCR results presented by us and published RNA-Seq data of *E. coli* that show that *proV* and *proX* have a higher transcript level than *proW*. While we have referred to Peters et al., 2012, Genes Dev in our manuscript, this is a general trend observed in *E. coli* RNA-Seq data (Oberto et al., 2009, PLoS One; Prieto et al., 2012, Nucleic Acids Res). Regulatory processes acting at the translational level may account for the differences.

Our RT-qPCR studies complement earlier studies with reporter fusions; they do not negate them. Throughout the manuscript, we compare our RT-qPCR data with published reports and explicitly mention where our data match earlier findings, and where they do not. We provide justifiable arguments for the latter with relevant citations, and in cases where there were no scientific reports to base our explanations on, we state that these are our hypotheses. Our biological interpretations and models of *proVWX* regulation take the literature focused on the regulation of *proU* into account to provide a meaningful, unbiased and integrated model.

'It is possible, indeed likely, that differences in the RT-qPCR data for the different regions within the operon are because of (a) differences in primer-binding efficiencies during RT-qPCR and / or (b) differences in physical degradation of different mRNA regions by ribonucleases within the cell (since the RT-qPCR method measures RNA abundance and not rate of mRNA synthesis alone).'

> It is true that the cycle of quantitation (Cq) values measured for every individual amplicon depend on primer binding efficiencies. Cq values are also affected by replication efficiency of the amplicon (affected by amplicon sequence), and the fluorescence signal detected at the end of each cycle (affected by amplicon length, size, quality of the polymerase and reagents used, and the detection platform). To account for all of these in an unbiased manner, for every amplicon/target that we measured by RT-qPCR (and 3C-qPCR) we generated a standard curve of Cq value against concentration for that specific amplicon/target in the same multi-well plate and using the same Master mix as our test reactions. We used the standard curve to quantify the amplicon/target. The template used to generate the standard curve was identical across all RT-

qPCR (or 3C-qPCR) reactions. These routine technical considerations comply with the MIQE guidelines (Bustin et al., 2009, Clin Chem). Therefore, our RT-qPCR data accurately reflect transcript levels and are not obscured by primer binding efficiencies.

The reviewer is correct in pointing out that differences in relative transcript levels measured by RT-qPCR may be caused by differences in the physical degradation of different mRNA regions by ribonucleases in the cell since RT-qPCR does not measure mRNA synthesis. In fact, we explicitly address this with our experiments using the *rnc* mutant. To us, this comment concerns our finding that *proW* has a lower expression level than the flanking *proV* and *proX* genes, and our interpretations that this indicates termination between *proV* and *proW* and the presence of a promoter between *proW* and *proX*. Term-Seq experiments performed by us (Figure S2), and others (Adams et al., 2021, eLife – suggested by reviewer #1) show that transcription termination does occur between *proV* and *proW*. Moreover, using a differential RNA sequencing approach, Thomason et al., 2015, J. Bacteriol. detect the presence of a transcription start site between *proW* and *proX*.

3. Therefore the authors claims for (a) an additional internal promoter for *proX* (page 8, line 243); (b) a terminator between *proV* and *proW* (page 8, line 241); (c) RNAP recycling for increased *proX* expression (page 8, line 249); and (d) regulation by G4 quadruplex within *proW*, are not sustainable in the absence of support from additional independent lines of evidence.

> This seems to be an incorrect representation of our writing. In the version of the manuscript that we submitted for review, we did not present (a), (c), and (d) as claims. In the absence of supporting experimental evidence, we used the terms ‘predict’ and ‘hypothesise’ for these statements. To highlight that these were data-driven hypotheses, we cited relevant literature. Statement (b) was made based on Term-Seq data that we presented in the supplementary information of our manuscript that shows termination between *proV* and *proW*.

It should be noted that the references suggested by reviewer #1 (Adams et al., 2021, eLife; Thomason et al., 2015, J Bacteriol), added at the revision stage, provide evidence for hypothesis (a), and additional supporting evidence for statement (b). Adams et al., 2021, eLife and Thomason et al., 2015, J Bacteriol do not support hypothesis (c), so we have removed it. (d) still remains a hypothesis.

Our revised manuscript contains these changes in lines 183-197. ‘Mechanistically, the decline in relative expression between *proV* and *proW* may be accounted for by transcription termination at the junction of the two genes, while the sharp increase in expression between *proW* and *proX* may arise because of the presence of an internal promoter between the two genes. Indeed, with Term-seq – a technique used to identify transcription termination sites⁸¹ – the presence of a termination site downstream of *proW1* at chr:2806395-2806398 was detected in our study and at chr:2806397-2806401 in an earlier report⁸² (Accession number: NC_000913.3) (Figure S2, SI 1A). Despite its positioning downstream of *proW1* (SI 1A), the terminator may have had an impact on the *proW1* RT-qPCR signal. Our Term-seq studies also identify a termination site between *proW* and *proX* downstream of a putative hairpin structure (Figure S2, SI 1A), however, this site was not detected in Term-Seq studies performed by others⁸². TSS mapping using differential RNA sequencing⁸³ identifies a TSS within *proW* at chr:2806878 (NC_000913.3)⁸⁴ positioned 23 basepairs downstream of the *proW2* amplicon. Transcription initiating from this site may account for the increase in the transcript levels of *proX*. The promoter regulating expression from this site is likely a non-canonical promoter rather than a classical H-NS-repressed spurious promoter since H-NS ChIP shows no H-NS signal at *proW* (Figure S3)⁸⁵, and spurious transcripts have not been detected from *proW* in a Δhns background⁸⁶.’

4. Likewise on the basis of RT-qPCR data, the authors state that loss of StpA in a strain proficient for H-NS is associated with derepression of *proVWX* expression. This finding is questionable, however, since there is abundant evidence (ref. 81; PMID 9473058, PMID 11278074) that StpA is a molecular back-up to H-NS

and that its absence alone has little effect in *E. coli*. Furthermore, PMID 9473058 has clearly shown that *proVWX* regulation is unaffected in a *stpA* single mutant.

> The reviewer correctly confirms that *StpA* is widely considered to be a molecular back-up of H-NS (lines 132-134 of our original manuscript). As such, we had expected to observe no changes in the transcript levels of *proVWX* in a Δ *stpA* strain. But we did detect such changes. According to the in vitro transcription assays reported in Boudreau et al., 2018, *Nucleic Acids Res*, detecting changes in transcription of genes co-regulated by H-NS and *StpA* is an expected observation.

We politely disagree with the reviewer that PMID 9473058 has clearly shown that *proVWX* regulation is unaffected in a *stpA* single mutant. PMID 9473058 relies on the *proU* P2 promoter system as a read-out to study the role of *StpA* as a molecular adapter of H-NS. The study shows that the expression of the *proV-lacZ* reporter cloned downstream of the *proU* P2 promoter is unaffected in a Δ *stpA* strain at low osmolarity. Our experiments show a ~4-fold increase in *proV* expression. The templates/constructs used to probe the role of *StpA* differ between the two studies. PMID 9473058 uses only *proU* P2 and the part of the negative regulatory element (NRE) occurring on the *proV* truncate to report on the role of *StpA* at *proU*. In our experiments, we study the effect of *stpA* deletion on the expression of *proVWX* at its native locus in the chromosome in the presence of all the known regulatory elements of the operon and those that are yet to be identified.

5. By using LB with 0.08 M NaCl as their low-osmolarity medium, the authors have missed out on sampling the full range and magnitude of *proVWX* osmotic regulation, which is at least 200-fold. This is so since there is another order of magnitude of repression between 0.08 M NaCl and 0.01 M NaCl (refs. 31, 33, 57). A mere extrapolation of their model to cover the expanded range may not necessarily be appropriate here, since the full magnitude of regulation appears to be achieved through a combination of mechanisms.

> We agree with the reviewer. We have now included the statement 'Notably, the full range of *proVWX* osmotic regulation has not been sampled in our report, for instance, there is at least an order of magnitude of repression of *proVWX* between 0.08 M NaCl and 0.01 M NaCl^{34,37,49,54}. Other regulatory mechanisms may function in these ranges, adding to the complexity of *proVWX* osmoregulation.' at lines 800-803 of the revised manuscript.

6. The authors indicate that because of an adaptation response, *proVWX* expression during steady-state growth at high osmolarity is only 1.3- to 2-fold higher than that during growth at low osmolarity, but the values reported by them are once again contrary to published data from many groups on *proVWX* expression measurements during steady state growth at different osmolarities (refs. 29, 31). The latter indicate that up-regulation of *proVWX* even during steady-state growth at high osmolarity is at least 100-fold, with a sigmoidal pattern of elevated expression observed with progressive increase of growth medium osmolarity.

> The reviewer correctly points out that Gowrishankar, 1985, *J Bacteriol* (ref. 29 of the original manuscript; ref. 33 of the revised manuscript), Cairney et al., 1985, *J Bacteriol* (ref. 31 of the original manuscript; ref. 37 of the revised manuscript), and even Nagarajavel et al., 2007, *Journal of Biological Chemistry* (ref. 49) show the sigmoidal pattern of *proU* up-regulation with increasing osmolarity. However, these studies post-translationally detect the expression of *proU* using reporter gene fusions to a truncate of *proVWX*. Our experiments use RT-qPCR to directly determine transcript levels of the *proVWX* operon.

Our data do not negate these studies of *proU* regulation, where regulation was detected post-translationally. They complement them by providing information of *proVWX* regulation at the transcript level. (See our response to comment 2)

7. With regard to the Hi-C and 3C-qPCR experiments, the authors have primarily catalogued the osmolarity-associated alterations at the *proVWX* locus and then attempted to build mechanistic models of regulation

from these data. However, the fundamental questions that remain unaddressed are (a) to what extent are those alterations specific to *proVWX* in comparison to other control loci, and (b) are these alterations causally relevant to its transcriptional control (and not consequential, or even incidental, to the changes in transcription)? In other words, how do changes in growth medium osmolarity affect other control chromosomal loci, and what are the changes that occur when transcription is activated at other control operons by their cognate environmental cues? In the absence of such information, it may not be possible to consider as logical or meaningful the interpretations / conclusions sought to be drawn from the *proVWX*-related changes.

> The referee raises multiple points in this comment. We have separated the comment into smaller sections and responded to each.

‘With regard to the Hi-C and 3C-qPCR experiments, the authors have primarily catalogued the osmolarity-associated alterations at the *proVWX* locus and then attempted to build mechanistic models of regulation from these data. However, the fundamental questions that remain unaddressed are (a) to what extent are those alterations specific to *proVWX* in comparison to other control loci,...

> We report on the interplay between local chromatin organisation and gene expression in the *proVWX* operon of *Escherichia coli*. We motivate our choice with the results of Hi-C experiments that we show in the Supplementary information (Figure S9) where we observe that one of the regions that stands out with regards to chromosome remodelling in response to osmotic stress is the chromatin around *proVWX*. We cannot and should not extrapolate the molecular details of our model – for instance, the positions of the DNA–H-NS–DNA bridges – to any other loci, not even *proVWX* of *Salmonella* sp. since all genes/operons differ in the affinity and distribution of NAP binding sites. However, the mechanistic features of our model, such as the consequences of the formation and disruption of DNA–H-NS–DNA bridges on transcription, can be extrapolated. These features of the model are supported by in vitro and in silico studies reported earlier (van der Valk et al., 2017, eLife; Qin et al., 2020, Nucleic Acids Res; Boudreau et al., 2018 Nucleic Acids Res).

‘...and (b) are these alterations causally relevant to its transcriptional control (and not consequential, or even incidental, to the changes in transcription)?...’

> This is indeed an important point. We have answered this question in the version of the manuscript that was evaluated by the reviewer. We reported in lines 679 to 752 of the original manuscript (670-736 of the revised manuscript) that there is an interplay between the local three-dimensional structure of *proVWX* and transcription. Briefly, a comparison of the 3C-qPCR profiles of the wild-type strain and of a strain where the H-NS DNA binding affinity of the *proVWX* downstream regulatory element was reduced show that the decrease in the relative interaction frequency between promoter P2 of *proVWX* and *ygaY* favours the activation of *proVWX*. The 3C-qPCR profiles of wild-type cells treated with rifampicin indicate that active transcription is necessary to remodel the interaction between promoter P2 of *proVWX* and *ygaY*. There is an interplay between local chromosome structure and gene expression rather than a unidirectional effect.

‘...In other words, how do changes in growth medium osmolarity affect other control chromosomal loci, and what are the changes that occur when transcription is activated at other control operons by their cognate environmental cues?...’

> These are truly exciting questions. It is a new and open field, and here, we have made the first steps towards answering these questions. We hope that our research will inspire other scientists in the bacterial chromatin community to engage in similar studies to reveal the full extent of interplay between (dynamic) chromatin structure and transcription.

‘...In the absence of such information, it may not be possible to consider as logical or meaningful the interpretations / conclusions sought to be drawn from the *proVWX*-related changes.’

> We respectfully disagree with this opinion of the reviewer. Our data show that there is an interplay between local three-dimensional chromatin organisation and gene expression using *proVWX* of *Escherichia coli* as a model system. Our data show that the nucleoid associated protein H-NS is implicated in structural regulation. This is direct evidence in favour of a long-standing hypothesis that H-NS is a sensor-effector. It functions as a transcription factor by re-modelling local chromatin in response to environmental cues. The absence of information about the structural remodelling of other loci in response to their cognate environmental cues does not minimise the significance of this finding.

8. The authors' suggestion in the Abstract and at several places in the text, that YgaG participates in osmotic regulation of *proVWX* transcription, flies in the face of a wealth of published data that a 1.2-kb region, spanning the NRE and including the promoter(s), is sufficient to fully reconstitute *proVWX* osmotic regulation (refs. 49, 56, 57, PMID 7968545). Such regulation is observed even if this region is shifted to other chromosomal locations, or is cloned on a plasmid.

> We do not agree with the reviewer. Experiments of *proU* osmotic regulation have focused on truncates of the *proVWX* operon outside its chromosomal context. The presence of regulatory elements in flanking regions has not been investigated. Hence, the reference data to be able to conclude whether the 1.2 kb region spanning the NRE and including promoters is sufficient to fully reconstitute the osmotic regulation of *proVWX* is unavailable.

The references that the reviewer has cited to oppose our results do in fact not support his/her opposition.

- Pavitt and Higgins, 1993, Mol Microbiol (PMID 7968545) use the *proU* promoter cloned upstream of the bioluminescent luciferase (*luxAB*) reporter genes of *Vibrio fischeri* as a supercoiling sensitive probe that can be tuned with osmolarity. The *proU-luxAB* probe was recombined at multiple locations in the *Salmonella typhimurium* chromosome to assess supercoiling levels in different chromosomal domains. The article does not report on whether the 1.2 kb region spanning the NRE and including the promoter(s), is sufficient to fully reconstitute *proVWX* osmotic regulation.
- Dattananda et al., 1991, J Bacteriol (ref. 49 of the original manuscript; ref. 50 of the revised manuscript) study the osmotic inducibility of the *proU* operon and demonstrate the osmoresponse of promoter P1, promoter P2, and the negative regulatory element in *E. coli*. These experiments were performed using truncates of the *proVWX* operon cloned into a plasmid. The authors do not study whether *ygaY* is involved in the regulation of *proVWX* since it is not included in the experimental design. The authors also do not draw comparisons between their results and the regulation of *proVWX* in its chromosomal context since the latter information is unavailable. Hence, this article does not permit drawing conclusions of whether P1, P2, and the NRE are sufficient to fully reconstitute *proVWX* osmotic regulation.
- Similar to Dattananda et al., 1991, J Bacteriol, Overdier and Csonka, 1992, PNAS (ref. 56 of the original manuscript; ref. 53 of the revised manuscript), report on the regulatory region of *proVWX* positioned at the 5' region of the operon using truncates of *proVWX* cloned on plasmids in *Salmonella typhimurium*.
- Kavalchuk et al., 2012, RNA Biol (ref. 57 of the original manuscript; ref. 54 of the revised manuscript) studied the osmoregulation of *proU* using constructs containing the -315 to +1260 region centred around the *proU* P2 transcription start site. Experiments were performed with the constructs cloned into plasmids, or as chromosomal insertions at an ectopic locus (*attB*) in a Δ *proVWX* background. The study does not produce data that can be used to conclude that the 1.2 kb region comprising P1, P2, and the NRE is sufficient to reconstitute *proVWX* osmoregulation. In contrast, it shows that post-transcriptional processing by RNaseIII would be required to reconstitute *proVWX* osmoregulation.

9. Page 19, para 2 is confusing. The authors employ mutations that "are expected to weaken IHF binding affinity", but conclude from the data obtained that it "highlights that the interaction is mediated by H-NS rather than IHF".

> We thank the reviewer for his/her valuable suggestion to rewrite this section more clearly. To study the role of H-NS in the interplay between local three-dimensional chromatin architecture and expression of the *proVWX* operon, we reduced the H-NS-binding affinity of the *proVWX* DRE by introducing point mutations. Mutation of the H-NS binding site located at -7 to +15 around P2 (Lucht et al., 1994, J Biol Chem), involved disruption of the IHF binding site located at the same position. Despite the point mutations impacting both H-NS and IHF binding, we chose to introduce them. This is because with the exception of introducing mutations to the high affinity H-NS binding sites of the DRE (Bouffartigues et al., 2007, Nat Struct Mol Biol), all other potential mutation sites would involve disruption of the -35 promoter element, -10 promoter element, or Shine-Dalgarno sequence (figure S4).

We have added the following details to the revised manuscript (lines 614-621) since the earlier explanation was insufficient: 'Both IHF and H-NS can form DNA—NAP—DNA bridges ^{25,101}. But, while IHF forms a bridge between its consensus binding site and a locus that is recruited non-specifically ¹⁰¹, H-NS preferentially bridges AT-rich loci ¹⁰². The 3C-qPCR profiles of *proVWX* in NT644 show that point mutations to the DRE disrupt the interaction between fragments proU3_NlaIII and proU11_NlaIII (arrow I, Figures 6a-6c), rather than causing non-specific decreases in relative interaction frequency. This observation indicates that the interaction is mediated by H-NS rather than IHF. The osmosensitivity of this interaction observed in the wild-type strain (Figure 5d) further strengthens this conclusion.'

10. By combining Results and Discussion into a single section in the manuscript, the authors appear to have camouflaged their indulgence in extensive speculation from a relatively limited set of experimental data. For example, in each of Figures 2, 3, 4 and 5, only three panels represent original experimental findings whereas an additional five to eight panels have been added that comprise derivative information. Furthermore, the model upon which much of their speculation is founded, of instantaneous potassium influx upon an osmo-shock driving the relief from H-NS mediated repression of *proVWX* transcription, is itself not universally accepted (see PMID 7929004).

> The reviewer raises multiple points in this comment. We have addressed them separately below.

'By combining Results and Discussion into a single section in the manuscript, the authors appear to have camouflaged their indulgence in extensive speculation from a relatively limited set of experimental data. For example, in each of Figures 2, 3, 4 and 5, only three panels represent original experimental findings whereas an additional five to eight panels have been added that comprise derivative information...'

> We do not agree with the observations and interpretations of the reviewer. We have presented an extensive dataset in this manuscript. Our findings (independently) demonstrate the osmoregulation of *proVWX* and osmoregulation by H-NS *in vivo*. They add to a molecular model of osmoregulation of *proVWX* by H-NS that has been developing over several years based on *in vitro* and *in silico* studies. Our manuscript integrates years of *proVWX* and H-NS research, and it was written with a combined results and discussion section to ensure that readers have a clear description of our observations, proper references for each of our inferences and hypotheses, and a step-wise view of the larger model that we present. We have not extensively speculated nor have we camouflaged our speculation where we have made speculations – in fact, we have explicitly stated it. This way, we have highlighted where our work is limited and where it can be built upon.

We specifically separated the original experimental findings from the derivative information to provide readers with a clear view of what our dataset looks like, what specific information we have derived from the dataset, and visually show the reliability (or not) of this derived information. We draw conclusions from this derived information and want to ensure that we are visually open with our analysis.

The derived information that we extract from our data and draw conclusions from are only fold changes in transcript levels that were calculated by dividing the transcript levels of sample A with the transcript levels

of sample B. This is a standard analysis performed for RT-qPCR, RNA-Seq, and reporter gene fusion assays. All studies of the regulation of *proVWX* cited in our manuscript and those referred to by the reviewer extract such information from experiments and use it to draw conclusions.

‘...Furthermore, the model upon which much of their speculation is founded, of instantaneous potassium influx upon an osmo-shock driving the relief from H-NS mediated repression of *proVWX* transcription, is itself not universally accepted (see PMID 7929004).’

> We welcome the reviewer’s argument and would like to present a discussion. Csonka et al., 1994, J Bacteriol (PMID 7929004) show that in *Salmonella typhimurium* the intracellular glutamate pool that (is predicted to) reflect the intracellular K⁺ level does not correlate with levels of *proU* induction. In this article, the intracellular glutamate pools and the β-galactosidase activity of a *proU-lacZ* fusion in wild-type and *gltB* mutant backgrounds are measured. The authors find that if the strain backgrounds and culture conditions are not considered, cells that have the same intracellular glutamate level do not have the same level of β-galactosidase activity. But, in the same background, an increase in intracellular glutamate level (triggered by an increase in extracellular osmolarity) is associated with an increase in β-galactosidase activity. This finding shows the link between intracellular glutamate levels (a proxy for the influx of K⁺) and expression from *proVWX*. Note that *gltB* mutants are defective in glutamate synthesis, and that the authors of PMID 7929004 acknowledge that ‘it will be of interest to see whether the *gltB/D* mutants can elevate their K⁺ pool...’

It is worth noting that since Csonka et al., measured intracellular glutamate pools, and acknowledge that glutamate is assumed to reflect intracellular K⁺ levels, they conclude that glutamate is not an obligatory component of the transcription regulation of *proU*. They say that they ‘cannot exclude the possibility that K⁺ by itself is the regulatory signal and that glutamate [is] merely one acceptable compatible anion.’ Nevertheless, the report also shows that in *in vitro* transcription-translation assays involving the use of the whole cell lysate for *in vitro* studies, an increase in K-glutamate induces expression from the osmoresponsive *proU* promoter with the silencer, the *proU* promoter without the silencer, the non-osmoresponsive *glnA* promoter, and the constitutive *lacUV5* promoter.

However, Sutherland et al., 1986, J Bacteriol find that in *E. coli* the β-galactosidase activity of *proU-lacZ* is reduced in mutants of *kdp*, the high affinity potassium transport system, compared to a wild-type background during osmotic stress (See table 3 of Sutherland et al., 1986, Journal of Bacteriology). The authors also show that during osmotic stress at extracellular K⁺ concentrations of 100 μM, 350 μM, and 1 mM, the intracellular K⁺ concentration is lower in *kdp* mutants (See table 4 of Sutherland et al., 1986, Journal of Bacteriology). At extracellular K⁺ concentrations of 10 mM and 50 mM, the intracellular K⁺ concentration is comparable between *kdp::Tn10* and wild-type backgrounds. By directly measuring intracellular K⁺ concentrations, Sutherland et al. show that the increase in intracellular K⁺ induces *proU*.

We present a model of how local chromatin re-organization mediated by the K⁺-sensitive H-NS regulates *proVWX* expression in *E. coli*. It is more suitable to base our model on a study of *proU* regulation in *E. coli* that directly measures intracellular K⁺ concentration rather than on the study investigating *proU* regulation in *Salmonella typhimurium* in response to intracellular glutamate levels that may or may not reflect intracellular K⁺ levels.

11. The authors use the term "osmosensitive" to refer to changes in *proVWX* transcription in response to changes in growth medium osmolarity, but this is confusing since the word would normally be expected to refer to a growth-sensitivity phenotype.

> We thank the reviewer for highlighting to us the confusion caused our choice of terminology. We have changed the terminology that we use to ‘osmoresponsive’.

12. There are many errors in the list of References. Ref. 21 cites a Cell 2015 preprint. Of greater concern

is the unconscionably large number of duplicate entries, of which I counted at least 20 pairs: 3-91; 94-102; 31-43; 32-44; 52-53; 45-88; 46-54; 47-55; 51-69; 70-74; 71-76; 84-87; 82-90; 80-113; 60-120; 37-130; 133-135; 144-145; 119-147; 57-118.

> We sincerely apologise for this. During our final check just before starting the submission of our manuscript, we had to update our reference manager from Mendeley Desktop to Mendeley Cite. Due to the incompatibility of Mendeley Cite with the security of our work computers, the references were updated as a last step on a personal computer. Since the references had already been critically checked before the update, the manuscript was submitted immediately. We have made the corrections.

Reviewer #3 (Remarks to the Author):

In this manuscript, Fatema-Zahra et al. used RT-qPCR and 3C-qPCR in different cell lines (wild-type and mutants) to investigate the relationship between transcription regulation and local folding of the E. coli chromosome. They focused on the transcriptional properties of the ProU operon, which is a well-known case of regulation by nucleoid-associated proteins (NAPs), particularly H-NS. From their study, they concluded that repression of ProU at low salt concentration occurs through the formation of two bridges involving H-NS between the downstream regulatory elements (DRE) and the upstream regulatory elements (URE) and the ygaY gene, respectively.

In my opinion, this work provides an interesting review of our knowledge of the action of H-NS in the structuring of DNA in vitro. It also provides interesting novel results about the effect of different mutants on the transcriptional properties of the ProU operon. However, I think that the authors' conclusions are not supported by their observations and therefore leave open the question of the precise mechanisms involved in vivo for the regulation of ProU and, more generally, for the regulation phenomena involving H-NS. Thus, I think that the results obtained are more suitable for a more specific journal.

> We thank the reviewer for the constructive evaluation of our manuscript.

More specifically, the new results on which the conclusions of this work are based are the results of 3C-qPCR in Figures 6 and 7, which need to be compared with the results in Figures 2 and 3. There are three major problems in this regard: firstly, the results in Figure 6A do not demonstrate that there is a loop between DRE and ygaY. The authors only hypothesize this loop based on the observation of a very slight decrease in contact frequencies (less than 10% at first sight). Secondly, the size of the fragments induced by NlaIII is too large to conclude that DRE forms a loop with URE. Finally, if there is a loop between DRE and URE, there must necessarily be a loop between DRE and proU13 located between nrdE and nrdF, given the values of the frequencies obtained. In addition, during hyperosmotic shock, the entire region between nrdE and DRE is affected, as can be seen in Figure 6F, which goes against the idea that the observed variations concern only a single loop (between DRE and URE). In fact, there is at least one control missing in this work to conclude anything about the presence of loops between URE and other elements: a 3C-qPCR profile centered not on URE but, for example, on proU6. In particular, variations in 3C interactions can be related to variations in DNA-binding proteins because the 3C signal reflects the crosslinking between proteins and DNA and between proteins themselves. A control like proU6 should therefore either confirm or refute this hypothesis.

> The referee raises multiple points in this comment. We have separated the comment into smaller sections and responded to each.

More specifically, the new results on which the conclusions of this work are based are the results of 3C-qPCR in Figures 6 and 7, which need to be compared with the results in Figures 2 and 3. There are three major problems in this regard: firstly, the results in Figure 6A do not demonstrate that there is a loop between DRE and ygaY. The authors only hypothesize this loop based on the observation of a very slight decrease in contact frequencies (less than 10% at first sight).

> It is true that the data shows a <10% decrease in the relative interaction frequency between the DRE and *ygaY* upon a hyperosmotic shock. We also agree that, at first sight, the result might not appear notable. The decrease stood out to us due to the features of the 3C-qPCR technique, and the sharp decrease in relative interaction frequency between DRE and *ygaY* in NT644.

3C-qPCR measures the relative interaction frequency between a pair of loci by quantifying hybrid ligation junctions formed between them. Ligation between a pair of fragments removes these from the DNA pool. Therefore, capturing an interaction between the DRE and *ygaY* with the DRE as an anchor fragment makes the DRE unavailable for interaction with another fragment. Hence, even small decreases in relative interaction frequency may reflect large changes in local chromosome organization. The relative interaction frequency between loci that are sequentially proximal on the primary genome sequence is higher than those that are further apart. This means that small changes in the relative interaction frequency between loci further apart – in this case the DRE and *ygaY* – may reflect large changes in chromatin organization. Due to the absence of a systematic study that quantifies these effects, we have avoided placing a strong significance on these explanations in our manuscript. Nevertheless, they prompted us to hypothesize that should this interaction be important in the regulation of *proVWX*, mutating the DRE to reduce its NAP binding efficiency should affect the DRE-*ygaY* interaction. Indeed, in the DRE mutant strain NT644, we detected a decrease in the interaction between the DRE and *ygaY*. This provided us with experimental proof of the DRE-*ygaY* interaction. Since, as we explain in the manuscript, this interaction is mediated by H-NS, and H-NS forms H-NS:DNA filaments and DNA:H-NS:DNA bridges, we conclude that a loop forms between the DRE and *ygaY*.

Secondly, the size of the fragments induced by *NlaIII* is too large to conclude that DRE forms a loop with URE. Finally, if there is a loop between DRE and URE, there must necessarily be a loop between DRE and *proU13* located between *nrdE* and *nrdF*, given the values of the frequencies obtained. In addition, during hyperosmotic shock, the entire region between *nrdE* and DRE is affected, as can be seen in Figure 6F, which goes against the idea that the observed variations concern only a single loop (between DRE and URE).

> We do not disagree with the reviewer. We have also mentioned this in our manuscript (lines 623-629 of the revised manuscript), ‘A comparison of the *proVWX* 3C-qPCR profile of NT331 at 0.08 M NaCl where H-NS-mediated interactions of *proU3_NlaIII* are maintained with that of NT644 where H-NS-mediated interactions of the DRE are disrupted, shows a detectable decrease in the relative interaction frequency between the DRE (*proU3_NlaIII*) and the URE (*proU2_NlaIII*) (arrow II, Figures 6d), and a general decrease in the relative interaction frequency between *proU3_NlaIII* and the *nrd* genes that lie upstream of *proVWX* (shaded area, Figure 6d). This indicates that in addition to the DRE—H-NS—*ygaY* bridge, the repressive complex formed by H-NS involves an interaction between the URE and the DRE, and the compaction of local chromatin upstream of the *proVWX* promoter.’ But, since this is the first published report where 3C-qPCR is used as a major technique to study local changes in chromosome structure, we are very conservative with our conclusions. We placed some focus on the interaction between the URE and the DRE since earlier reports have extensively implicated cooperativity between the two elements in the regulation of *proVWX*. The interactions between the DRE and its upstream region were not interpreted beyond ‘remodelled chromatin’ to avoid building a complex model based on a limited set of data and literature.

We have now also explained our conclusion that a loop forms between the DRE and the URE by stating that (lines 551-553 of the revised manuscript), ‘... physical interaction between the high affinity H-NS binding site of the DRE contained in fragment *proU3_NlaIII* and the URE contained in *proU2_NlaIII* that also exhibits an H-NS binding preference.’

In fact, there is at least one control missing in this work to conclude anything about the presence of loops between URE and other elements: a 3C-qPCR profile centered not on URE but, for example, on *proU6*. In

particular, variations in 3C interactions can be related to variations in DNA-binding proteins because the 3C signal reflects the crosslinking between proteins and DNA and between proteins themselves. A control like proU6 should therefore either confirm or refute this hypothesis.

> Detecting the presence and role of loops between the URE and loci within and around the *proVWX* operon is an obvious next step. In our screening studies, we detected potential regulatory interactions between the URE and loci downstream of the DRE-*ygaY* interaction. Since the experiments were performed with only two biological replicates, and validated with fewer controls, we have chosen not to report this in this manuscript or in its supplementary information.

ProU6 is sufficiently stable with regards to relative interaction frequency when considering the proU3-proU6 interaction. However, this may not be true for the interaction of proU6 with other fragments. Therefore, generating the 3C-qPCR profile of proU6 would not fulfil the role of a control, but would be a full experiment and a new study. It is also a very interesting target to look at given its position just upstream of the *proW* gene. This locus is in close proximity to the termination site between the *proV* and *proW* genes, and the transcription start site of *proW*. It may reveal additional 'layers' of regulatory elements and features of *proVWX*.

We understand the need for proper controls. We have made sure to follow all the controls defined for 3C-qPCR (Hagège et al., 2007, Nat Protoc) and have performed all the controls that are required by the MIQE guidelines for publication of qPCR experiments (Bustin et al., 2009, Clin Chem).

Continuing on the theme of contact data between chromosomal loci, I find the choice to start with a general Hi-C analysis for this article not relevant and rather confusing. In particular, the analyses and conclusions discussed based on Figure 1 do not seem relevant compared to the rest of the discussion. In fact, the conclusions about the different relevant distances of the system seem incorrect: it is not reasonable to identify interaction distances based solely on a discussion of when the colors of a matrix change. These colors vary widely depending on the visualization choices used to represent the data and therefore cannot be used as a robust indicator. Furthermore, matrix comparisons are usually made using the log-ratio of the matrices, which in this case can actually lead to defining relevant distances for the action of NAPs (see e.g. [93]). In summary, the results in Figure 1 are very qualitative and do not provide a comprehensive explanation of the properties observed throughout the matrix.

> We thank the reviewer for raising this point and have made changes accordingly. We have moved this section from the main text to the supplementary information, and according to the explanation given by the reviewer, we have rectified what we have presented (Figure S9).

More generally, I found that the results were often presented as a discussion, with hypotheses often justified from an in vitro standpoint, but whose relevance in vivo is not clear.

> We apologize that it has come across in this manner. Since we present an extensive dataset on the *proVWX* model system that has been explored since the 1980s, and on the H-NS protein whose mode of function has been a topic of extensive research, we presented our work in a combined results and discussion section. In that manner we could build a step-wise model of the regulation of the *proVWX* operon in context of the in vitro, in silico, and in vivo work that has been done. This was also meant to make the results, our interpretation of them, and the model we build on the basis of these easier and clear to follow even for a reader less familiar with the topic.

In vivo work is, at present, limited in providing a mechanistic insight. Therefore, we have used the in vivo data that we present to provide a molecular model, and the published in vitro models to propose a mechanistic model.

We have made changes to our results section to distinguish the molecular and mechanistic models. This was particularly encouraged by reviewer #1 (See comments and responses referring to lines 735-760, 769, 774-775, 53-58, and 109-114 of the original manuscript). We have also made textual changes to improve the distinction between our results and our interpretations of them.

Please find an additional list of suggestions for the authors:

* Line 78 and 79: the terms P1R and P2R are unnecessary since they are not used elsewhere in the article

> We thank the reviewer for noticing. We have removed these abbreviations.

* Lines 95-96: Is the statement "Heterologous promoters..." associated with an H-NS phenomenon? If so, this should be explained. If not, this sentence should instead appear in the previous paragraph.

> Heterologous was not the correct word to use. We have changed it to 'Other promoters...' (lines 104-105 of the revised manuscript)

* Lines 118-119: The sentence "The effect of *ihf* deletion..." is not clear (i.e., which effect?)

The deletion of *ihf* did not affect the expression of *proVWX* according to genome-wide studies. If we have understood the comment correctly, it sounds like the sentence seems to present a 'defined' effect of *ihf* deletion. So, we have changed 'The effect...' to 'An effect...' (line 127 of the revised manuscript)

* The sentence "HU regulates expression..." at lines 120-121 is problematic.

> Indeed the sentence construction was odd. It now reads 'HU regulates expression from the *proU* P2 promoter and its osmoinducibility⁶⁸.' (lines 129-130 of the revised manuscript)

* Lines 222 and 223: "One of the regions..." Many regions actually seem to be affected, with one, for instance, in the NSR. A more systematic analysis would be required here.

> We have moved this section to the supplementary information. Since we have not performed a systematic analysis nor do we have the experience to perform that reliably, we have removed such mentions.

* Line 452: "This may, however,..." The authors could elaborate.

> We have now included an explanation. The sentence reads, 'This may be accounted for by the inherent osmosensitivity of P1 – expression from P1 increases with increasing osmolarity *in vitro*^{39,50}.' (lines 393-395 of the revised manuscript)

* Fig. S7: It would be useful to show the ChIP-seq signal of H-NS over the entire segment discussed by the authors (from *nrdE* to *ygaZ*).

> We have now shown this in Figure S3.

* Fig. S7: It would be useful to show the ChIP signal of StpA discussed by the authors.

> We have now shown this in Figure S3.

REVIEWERS' COMMENTS

Reviewer #1 (Remarks to the Author):

The revised manuscript addresses all points raised in my review and the manuscript is excellent. I agree with the authors that fold changes in transcript levels determined by qRT-PCR, a direct read-out method, do not necessarily replicate fold-changes determined by promoter lacZ fusions, as published before and this is accurately discussed in the manuscript. The Hi-C data are remarkable in my opinion and possible pitfalls are mentioned and discussed.

As a editorial comment I suggest to change

179 The transcriptional profile of proU in NT331

to: The transcriptional profile of proU in E. coli MG1655 strain NT331

hns^{wt} to hns^{+}

Reviewer #2 (Remarks to the Author):

The rebuttals offered by the authors and the revisions made to the manuscript in response to the reviewers' comments are reasonable and satisfactory. The work has employed novel approaches to offer a new model for osmoresponsivity of the proVWX locus of E. coli, that can be tested in future studies.

Reviewer #3 (Remarks to the Author):

I appreciated the response from the authors and the changes made to the text. However, I still believe that the abstract contains overstatements regarding the findings reported by the authors. The authors indeed state:

"We show that activation of proVWX in response to a hyperosmotic shock involves the destabilization of H-NS-mediated bridges anchored between the proVWX downstream and upstream regulatory elements (DRE and URE), and between the DRE and ygaY that lies immediately downstream of proVWX."

Nevertheless, this statement is not entirely accurate. As mentioned in my initial report and acknowledged by the authors, the conclusion about a specific loop between DRE and URE cannot be drawn solely from this work due to the limitations posed by the length of the 3C-qPCR fragments. The authors clarify in their response (and in the updated version) that this conclusion can only be supported when considering the context of previous in vitro studies. Therefore, the abstract should be revised to reflect this nuance.

Additionally, I believe it would be worthwhile to mention that the experimental results indicate a more complex scenario than previously expected, with the compaction of local chromatin upstream of the proVWX promoter, as discussed in lines 623-629.

REVIEWERS' COMMENTS

Reviewer #1 (Remarks to the Author):

The revised manuscript addresses all points raised in my review and the manuscript is excellent. I agree with the authors that fold changes in transcript levels determined by qRT-PCR, a direct read-out method, do not necessarily replicate fold-changes determined by promoter lacZ fusions, as published before and this is accurately discussed in the manuscript. The Hi-C data are remarkable in my opinion and possible pitfalls are mentioned and discussed.

We thank the reviewer for their positive opinion of our manuscript.

As a editorial comment I suggest to change 179 The transcriptional profile of proU in NT331 to: The transcriptional profile of proU in E. coli MG1655 strain NT331 hnswt to hns+

We agree that since NT331 is a strain that was developed by us, it is indeed better to include a description of it with standardized terminology in headings and subheadings. We have changed the title in line 179 (line 169 in the revised manuscript) from “The transcriptional profile of proU in NT331” to “The transcriptional profile of the proVWX operon in E. coli K-12 strain MG1655 ΔendA (NT331)”.

We have also changed hnsWT in line 114 (of the revised manuscript) to hns+. Thank you for pointing out the error to us.

Reviewer #2 (Remarks to the Author):

The rebuttals offered by the authors and the revisions made to the manuscript in response to the reviewers' comments are reasonable and satisfactory. The work has employed novel approaches to offer a new model for osmoresponsivity of the proVWX locus of E. coli, that can be tested in future studies.

We thank the reviewer for their positive evaluation of our manuscript.

Reviewer #3 (Remarks to the Author):

I appreciated the response from the authors and the changes made to the text. However, I still believe that the abstract contains overstatements regarding the findings reported by the authors. The authors indeed state:

“We show that activation of proVWX in response to a hyperosmotic shock involves the destabilization of H-NS-mediated bridges anchored between the proVWX downstream and upstream regulatory elements (DRE and URE), and between the DRE and ygaY that lies immediately downstream of proVWX.”

Nevertheless, this statement is not entirely accurate. As mentioned in my initial report and acknowledged by the authors, the conclusion about a specific loop between DRE and URE cannot be drawn solely from this work due to the limitations posed by the length of the 3C-

qPCR fragments. The authors clarify in their response (and in the updated version) that this conclusion can only be supported when considering the context of previous in vitro studies. Therefore, the abstract should be revised to reflect this nuance.

Yes, indeed, this nuance is missing from the abstract. In accordance with the reviewers suggestion, we have changed the statement quoted by the reviewer to, "By consolidating our in vivo investigations with earlier in vitro and in silico studies that provide mechanistic details of how H-NS re-models DNA in response to osmolarity, we report that activation of proVWX in response to a hyperosmotic shock involves the destabilization of H-NS-mediated bridges anchored between the proVWX downstream and upstream regulatory elements (DRE and URE), and between the DRE and ygaY that lies immediately downstream of proVWX."

Additionally, I believe it would be worthwhile to mention that the experimental results indicate a more complex scenario than previously expected, with the compaction of local chromatin upstream of the proVWX promoter, as discussed in lines 623-629.

Thank you for this advice. We have now included the following statement in our abstract, "Our results also reveal additional structural features associated with changes in proVWX transcript levels such as the decompaction of local chromatin upstream of the operon, highlighting that further complexity underlies the regulation of this model operon."